# Unconventional human CD61 pairing with CD103 promotes TCR signaling and antigen-specific T cell cytotoxicity

Megat H. B. A. Hamid [1,14], Pablo F. Cespedes [1,2,14], Chen Jin[1,3,14], Ji-Li Chen[1,4,14], Uzi Gileadi [4,14], Elie Antoun[1,5,14], Zhu Liang [1,6,14], Fei Gao [1], Renuka Teague[1,7], Nikita Manoharan[1], David Maldonado-Perez[1,7], Nasullah Khalid-Alham[8,9], Lucia Cerundolo[7], Raul Ciaoca[1], Svenja S. Hester[6], Adán Pinto-Fernández[1,6], Simeon D. Draganov[1,6], Iolanda Vendrell[1,6], Guihai Liu[1], Xuan Yao[1], Audun Kvalvaag[2,10], Delaney C. C. Dominey-Foy [1], Charunya Nanayakkara [1], Nikolaos Kanellakis [1,9,11,12], Yi-Ling Chen [1,4], Craig Waugh[13], Sally-Ann Clark [13], Kevin Clark[13], Paul Sopp[13], Najib M. Rahman [1,9,11,12], Clare Verrill[1,7,9], Benedikt M. Kessler [1,6], Graham Ogg[1,4], Ricardo A. Fernandes [1], Roman Fisher [1,6,15], Yanchun Peng[1,4,15], Michael L. Dustin [1,2,15] & Tao Dong [1,4,15] ✉

Cancer remains one of the leading causes of mortality worldwide, leading to increased interest in utilizing immunotherapy strategies for better cancer treatments. In the past decade, CD103⁺ T cells have been associated with better clinical prognosis in patients with cancer. However, the specific immune mechanisms contributing toward CD103-mediated protective immunity remain unclear. Here, we show an unexpected and transient CD61 expression, which is paired with CD103 at the synaptic microclusters of T cells. CD61 colocalization with the T cell antigen receptor further modulates downstream T cell antigen receptor signaling, improving antitumor cytotoxicity and promoting physiological control of tumor growth. Clinically, the presence of CD61⁺ tumor-infiltrating T lymphocytes is associated with improved clinical outcomes, mediated through enhanced effector functions and phenotype with limited evidence of cellular exhaustion. In conclusion, this study identified an unconventional and transient CD61 expression and pairing with CD103 on human immune cells, which potentiates a new target for immune-based cellular therapies.

Integrins are large, heterodimeric transmembrane glycoproteins that facilitate adhesion between cells, and with the extracellular matrix[1]. They require the pairing between an α and β subunit to exit the endoplasmic reticulum and reach the cell surface to become functionally active[2-5]. The pairing and functions of integrin β3 (gene *ITGB3* encoding CD61) with its known *cis*-integrin partners, integrin αV (gene *ITGAV* encoding

CD51) and integrin αIIb (gene *ITGA2B* encoding CD41) have been well documented on nonlymphocytic cells such as megakaryocytes, platelets and macrophages, as well as on endothelial cells[2,4,6-8]. While there are pieces of evidence of CD61 pairing with CD41 or CD51 on murine T cells[9,10], the expression and functional implications of CD61 on antigen-specific T cell immunity in human diseases, including cancer, remains unclear.

**Fig. 1 | CD61 is expressed on CD103⁺ CD8⁺ T cells. a**, Network plot showing clustering of enriched proteins in cancer-specific CD103⁺ T cell clones (*n* = 2 patients' paired T cell clones). Highlighted circles indicative of proteins likely associated with immunity. **b**, Heat map showing selected proteins enriched in both CD103⁺ NY-ESO-1-specific and SSX-2-specific T cell clones but downregulated in both CD103⁻ T cell clones. Cytotoxic T cell (CTL) CD103⁺ T cell clone is shown in teal and CD103⁻ T cell clone is shown in orange. Antigen (Ag) specificity: NY-ESO-1-specific T cell clones (pink), SSX-2-specific T cell clones (green). Expression level by log₂ fold-change (FC) values, with a gradient of red to blue. **c**, Graph showing the frequency of CD61⁺ cells (of total CD103⁺ TILs) by flow cytometry (right *y* axis), and the CD61⁺CD103⁺ co-located TILs by

IHC (left *y* axis), of each patient with NSCLC. n₁ IHC = 31 patients; n₂ flow cytometry = 19 patients. Diamonds represent the area of CD61⁺CD103⁺ co-located TILs by IHC. **d,e**, Percentage of CD61⁺CD103⁺ and CD61⁻CD103⁺ T cells of paired peripheral blood, paratumor tissue and tumor tissue by flow cytometry plots and a line plot. *n* = 19 patients. \*\*\**P* < 0.001, one-way analysis of variance (ANOVA) with Tukey's multiple-comparison test. *P* value (tumor versus paratumor): 0.0009, *P* value (tumor versus peripheral blood): 0.0003. **f**, Histograms showing CD61 expression on paired CD103⁺ and CD103⁻ cancer-specific T cell lines (*n* = 7 patients). Gray represents CD103⁺ T cell lines, and light red represents CD103⁻ T cell lines. *n* = 3 independent experiments with consistent results.

On the other hand, it is well established that integrin αE (gene *ITGAE* encoding CD103) pairs with integrin β7 (gene *ITGB7*) on murine and human immune cells. CD103 is considered a key phenotypic marker of resident memory T ($T_{RM}$) cells in a variety of tissues including tumors. In cancer, CD103[+] $T_{RM}$ tumor-infiltrating T lymphocytes (TILs) are known to be immunophenotypically diverse, ranging from terminally exhausted (layilin[+]), $T_{RM}$ precursors (granzyme H[+]) and $T_{RM}$ transitional-phase (XCL1[+]) cells[11]. Certain CD103[+] TIL subtypes, such as the CD103[+]CD39[+] cells are tumor-reactive TILs, with clonal expansion observed in different cancer types[12–14]. Our recent study further demonstrated CD103[+]TGF-β1[+] cytotoxic T cells were efficient killers of antigenic cancer[15].

Clinically, the enriched presence of CD103[+] T cells in patients with cancer and in patients with pathogenically disease has been associated with improved outcomes[16–19]. This positive clinical attribute makes CD103[+] T cells an important target for immunotherapy strategies. However, the immune-associated proteins and the mechanisms that are utilized to promote effective cellular immune activities, responses and protection remain poorly defined. Therefore, this study evaluates the mechanistic contributors of protective immunity on human antigen-specific cytotoxic CD8[+] T cells, using cancer as a disease model.

## Results

### CD61 is expressed on CD103[+]CD8[+] T cell clones and $T_{RM}$ TILs

The presence and enrichment of CD103[+] TILs are considered a good indicator of desirable clinical prognosis and outcome[12,14,16–18]. To investigate how these T cells can promote protective immunity, we previously generated HLA-A*02:01-restricted CD103[+] and CD103[−] CD8[+] T cell clones, from two separate patients[15]. The paired CD103[+] and CD103[−] T cell clones isolated from a patient with gastric cancer are characterized by the same T cell antigen receptor (TCR): TRAV8-6 TRAJ30, TRBV6-1 TRBJ2-7, recognizing the SSX-2 tumor antigen. In contrast, the second pair of T cell clones were isolated from a patient with melanoma with a distinct TCR: TRAV12-2 TRAJ31, TRBV12-4 TRBJ1-2, recognizing the tumor antigen, NY-ESO-1. In this study, the sourcing of biological samples from different patients with cancer will help to identify common immune markers of CD103[+] T cells that can modulate protective antitumor immunity beyond the interaction between the TCR and a peptide in the major histocompatibility complex (pMHC).

As part of a discovery approach to identify CD103 immune-related proteins, we first compared the proteomic profiles between the cancer-specific CD103[+] and CD103[−] T cell clones from these two patients with cancer. The differential expression analysis revealed 103 proteins enriched in the two CD103[+] T cell clones, compared to their respective paired CD103[−] T cell clones (Extended Data Fig. 1a). Among these proteins, 70.8% (92 proteins) were associated with cellular processes,

10% (13 proteins) with metabolism and 16.9% (22 proteins) with protein synthesis and trafficking (Extended Data Fig. 1b). Several of these protein subgroups were identified to be linked to immune activities, including immune effectors/cytokines, integrins, metabolic-related, TGF-β1-related and epigenetic-related groups (Fig. 1a).

Due to being described as an integrin of nonimmune cells such as platelets, endothelial cells and megakaryocytes in humans[2,4,6–8], the enrichment of CD61 (Fig. 1b) in the CD103[+] clones was therefore unexpected. While this proteomics approach was underpowered, these data provided an exploratory observation, and we chose to pursue CD61 from this preliminary dataset by validation through multiple orthogonal approaches.

To validate this exploratory in vitro proteomics approach, we investigated the existence of CD61[+] T cells within the total CD8[+] TILs in a larger cohort of patients ex vivo. We performed dual immunohistochemistry (IHC) and flow cytometric CD61 protein analyses on samples from a cohort of patients with non-small cell lung cancer (NSCLC) obtained from the Oxford Radcliffe Biobank (ORB; Extended Data Figs. 2 and 3a).

As evidenced by the dual IHC and flow cytometry analyses, we confirmed the presence of CD61[+] T cells in the CD103[+] TILs population, with variable frequencies across the patients with NSCLC, ranging from 2% to 77% (Fig. 1c and Extended Data Fig. 3b). The variability in the frequency of the TIL subpopulation correlated with the NSCLC tumor stages (Extended Data Fig. 3c), but not with other clinical parameters evaluated such as NSCLC pathology type (including between adenocarcinoma and squamous cell carcinoma), gender and age (Extended Data Fig. 3d).

Importantly, regardless of the frequency variability between patients, the CD61[+] TIL subset was significantly enriched in the lung tumor tissue of these patients compared to the paired paratumor tissue and peripheral blood (Fig. 1d,e).

To further confirm the existence of CD61 on human antigen-specific CD8[+] T cells, we evaluated its expression on seven pairs of cancer-specific CD103[+] and CD103[−] T cell lines, from seven different patients with cancer. Indeed, we found positive CD61 expression on the seven CD103[+] T cell lines, compared to their paired CD103[−] T cell lines (Fig. 1f). Taken altogether, these discovery-to-validation findings highlight the unconventional presence and expression of CD61 on human antigen-specific CD8[+] T lymphocytes, especially in human cancer.

### CD61 transiently colocalizes with CD103 at the synapse

To evaluate the kinetic expression of CD61 on human T cells, we next measured the expression following different activation regimens. The cancer-specific CD103[+] T cell lines, sourced from seven different patients with cancer demonstrated positive surface expression of CD61

**Fig. 2 | CD61 transiently colocalizes with CD103. a**, Horizontal bar graph showing the average CD61 median fluorescence intensity (MFI) on CD103[+] T cell lines following either activation by αCD3/CD28, or co-culture with antigenic cancer cells, or no activation by flow cytometry. ***$P < 0.001$, one-way ANOVA with Tukey's multiple-comparison test. $n = 7$ patients, examined over three independent experiments. ***$P$ values: (patient 1: 0.00098, patient 2: 0.00089, patient 3: 0.00001, patient 4: 0.000003, patient 5: 0.000, patient 6: 0.00002, patient 7: 0.00001). **b**, Representative synapse images of integrin β7, CD103, CD61 and merged, of a cancer-specific CD103[+] TCR-T cell at 10 min after synaptic formation. Enlarged green box shows zoomed-in synaptic microcluster images of CD103 and CD61 colocalization, but not integrin β7. **c,d**, Representative dot plot showing CD103 and CD61 colocalization by PCC ($P = 0.9435$), or CD103 and integrin β7 negative PCC colocalization ($P = 0.1973$) at 5, 10 and 15 min after synaptic formation; each dot represents the average PCC per synapse.
**e**, Representative synapse images showing internal reflection microscopy (IRM; denoting the area of synapse), integrin β7, CD49d and merged, of a cancer-specific CD103[+] TCR-T cell at 5 min after synaptic formation. Enlarged green box shows zoomed-in images of colocalized integrin β7 and CD49d.
**f**, Representative Co-IP immunoblot images of CD103 and CD61-flag on anti-flag

IP pulldown lysate and whole-cell lysate, of CD103[−]CD61-flag[+], CD103[+]CD61-flag[−] and CD103[+]CD61[−]flag[+] T cell lines. Molecular weight (MW) of CD103: ~150 kDa, of CD61-flag: ~100 kDa. $n = 3$ lines examined over three independent experiments, with consistent results. **g**, Volcano plot showing enriched proteins on lysates of CD103[+]CD61[+] T cells in comparison to CD103[+]CD61[−] T cells. $n = 2$ lines examined over one experiment. Dot represents one protein. One-way ANOVA with Tukey's multiple-comparison test, converted to $-\log_{10} P$ values for each data point. Raw fold-change values were normalized using $\log_2$. **h**, Representative flow cytometry plots of intracellular and surface staining of CD61-flag with CD103-HA following initial transduction of primary T cells with CD103 (left), followed by secondary transduction with CD61 (right). $n = 3$ independent experiments, with consistent results. **a,c,d**, Data are presented as the median ± s.e.m., **c,d**, Two-way ANOVA with Tukey's multiple-comparison test. **b,e**, $n = 150$ cells examined over three independent experiments with consistent observations per **c** and **d**; each dot represents the average PCC per synapse. **c**, time of 5 min: 72 synapses, time of 10 min: 102 synapses, time of 15 min: 141 synapses. **d**, time of 5 min: 53 synapses, time of 10 min: 81 synapses, time of 15 min: 40 synapses. **b,e**, Microscopy images: big scale bar, 5 μm; small scale bar, 1 μm. Ag, antigen. NS, not significant.

following activation either by agonistic αCD3/αCD28 antibody or following co-culture with antigenic cancer cells (Fig. 2a and Extended Data Fig. 4a). In contrast, resting T cells showed undetectable levels

of CD61 expression, which was consistently observed across all seven patients with cancer (Fig. 2a). We further found that upregulation of CD61 peaked within the first 2 h after T cell activation, before gradually

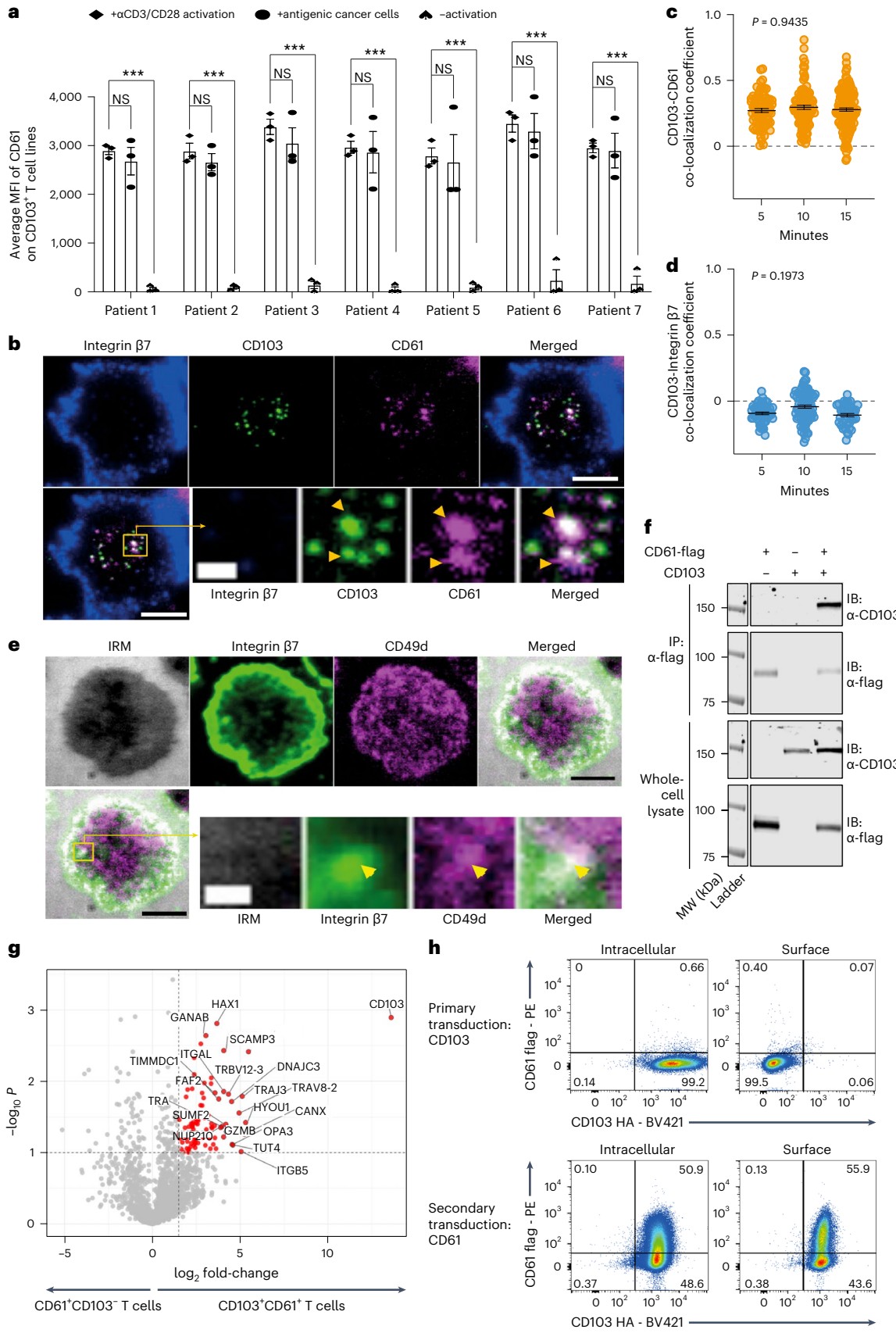

decreasing over time (Extended Data Fig. 4b), suggesting that CD61 expression in human T cells is transient.

Considering the transient upregulation of CD61 on the T cells, we sought to evaluate its potential involvement in the temporal scales of cell-to-cell contacts. We used total internal reflection fluorescence microscopy (TIRFM) to analyze the recruitment and distribution of CD61 within the synaptic contacts formed between the NY-ESO-1-specific CD61+ T cell line and supported lipid bilayers (SLBs)-containing physiological densities of antigenic pMHC, ICAM-1, CD58 and E-cadherin (SLB protein densities are as informed in the Methods). We found a time-dependent increase of CD61, as well as of CD103, at the points of contact between the T cell membrane and the bilayer, within the central supramolecular activation cluster (cSMAC) region, defined by the area of accumulation of antigen and TCRαβ (Extended Data Fig. 4c,d).

Interestingly, the CD61 colocalized with CD103 at the points of contact (Fig. 2b). The Pearson's correlation coefficient (PCC) of the colocalization between CD61 and CD103 was consistently positive throughout the 15 min of contact (Fig. 2c). In contrast, integrin β7, the known canonical integrin partner of CD103, was located in the outermost synaptic compartment, also known as the distal supramolecular activation cluster (dSMAC; Fig. 2b). The PCC for CD103 and integrin β7 was consistently below zero, indicating they were inversely correlated at the synapse (Fig. 2d). The dSMAC localization of integrin β7 is likely due to its interaction with its alternative *cis*-integrin partner, the integrin α4 (*ITGA4*, CD49d), rather than with CD103. We observed colocalization between CD49d and integrin β7 at the dSMAC microclusters (Fig. 2e), which may reflect an early CD49d-integrin β7 pairing and positioning, leading to segregation of integrin β7 from CD103.

With the central clustering of CD61 at the cSMAC, we next evaluated whether CD61's conventional *cis*-interacting integrin partners, CD41 and CD51, were relied upon for CD61 integrin heterodimerization on the T cell surface. Unexpectedly, both CD41 and CD51 were not expressed on the cell surface of the NY-ESO-1-specific CD61+ T cell line (Extended Data Fig. 4e). These data suggest that the absence of CD41 and CD51 on these T cells enables the unconventional pairing with CD103, as per Fig. 2b.

To further investigate the possible interaction between CD61 and CD103 on human T cells, we performed co-immunoprecipitation (Co-IP) analysis on CD61 pulldown lysates of primary T cells overexpressing both integrins. CD61 was found to be co-immunoprecipitated with CD103 when pulled down from the lysate of CD61-flag+CD103+ T cell line, but not on the lysate controls: CD61-flag−CD103+ T cell line lysate and the CD61-flag+CD103− T cell line lysate (Fig. 2f).

To further strengthen this observation, we carried out an analysis of the overall proteins that may be co-immunoprecipitated with the CD61-flag protein. Apart from proteins that are commonly known to associate with integrin heterodimer complex formation, we confirmed the presence of co-precipitated CD103 on the CD61-flag+CD103+ T cell line (Fig. 2g), further demonstrating the existence of a CD61–CD103 integrin complex. Additionally, no enrichment of integrin β7 was detected on the CD61-flag+CD103+ T cells (Extended Data Fig. 4f), suggesting the exclusion of integrin β7 from the CD61–CD103 complex.

As it is widely recognized that interaction between an α and β integrin subunit is required for the cell surface expression of integrins, we evaluated whether CD61 secondary transduction can promote CD103 cell surface expression (Extended Data Fig. 4f). Following the primary transduction of CD103-HA on primary T cells, we did not observe any surface expression of the CD103 (Fig. 2h). However, a secondary transduction of CD61 on the same CD103+ T cell line was able to rescue surface expression of CD103 on the T cells (Fig. 2h). Taken altogether, our findings using the multifaceted approaches above have demonstrated the potential interaction between CD61 and CD103 on human T cells.

## CD61 enhances TCR signaling

We next sought to evaluate the possible role of CD61 on the T cell signalosome. In parallel to CD61 and CD103 colocalization at the cSMAC, we further observed colocalization between CD61 with TCRαβ, as well as between CD103 and TCRαβ (Fig. 3a), suggesting that CD61 may modulate TCR signaling activity.

To assess the functional significance of CD61 toward proximal TCR signaling, we first generated CD61 knock-down T cells using short interfering RNA (siRNA), and CD61 CRISPR knock-out (KO) T cells, from the wild-type (WT) CD61+ T cell clone of cancer patient 1 (Extended Data Fig. 5a) before functional evaluation. Interestingly, we found Zap70 and PLCγ1 phosphorylation levels gradually decreased following serial CD61 siRNA treatments (Fig. 3b and Extended Data Fig. 5b). Additionally, the CD61KO T cell clone demonstrated impaired phosphorylation of both these proteins, to levels comparable with those seen on WT CD61− T cell clones (Fig. 3c and Extended Data Fig. 5b). To further verify the importance of CD61 in regulating Zap70 phosphorylation, we then evaluated the Zap70 (pY292) expression on WT CD61+ T cell lines from seven different patients with cancer. Indeed, we found that phosphorylated Zap70 levels were significantly impaired following treatment with αCD61 neutralizing antibody (Fig. 3d and Extended Data Fig. 5c).

Given that a cytosolic adaptor protein, the kinase Lck, is known to maintain and directly sustain the phosphorylation of Zap70 at the TCR complex's cytoplasmic domains[20,21], we next sought to evaluate the potential protein linker between CD61 and Zap70. We found the WT CD61+ T cell clone exhibited the highest Lck expression in comparison to the siRNA-treated and CD61KO T cell clones (Fig. 3e). In agreement, we demonstrated that treatment with αCD61 neutralizing antibody led to a significant downregulation of Lck expression on WT CD61+ T cell lines of seven different patients with cancer (Fig. 3f), suggesting that CD61 is a modulator of Lck-dependent Zap70 phosphorylation.

To further confirm the involvement of Lck in the CD61–Zap70 phosphorylation axis, we treated a WT CD61+ T cell clone with either the Lck-specific inhibitor A770041 or the broader tyrosine kinase inhibitor

**Fig. 3 | CD61 enhances TCR signaling. a**, Representative dot plot showing PCC of TCRαβ and CD61 colocalization (left) and TCRαβ and CD103 colocalization (right) at 5, 10 and 15 min after synaptic formation. *P* value (TCRαβ-CD61, 5 versus 15 min): 0.0009, *P* value (TCRαβ-CD103, 5 versus 10 min): 0.0075, *P* value (TCRαβ-CD103, 5 versus 15 min): 0.000782. Each dot represents the average PCC per synapse. *n* = 150 cells examined over three independent experiments; each dot represents the average PCC per synapse. Time of 5 min: 91 synapses; time of 10 min: 101 synapses; time of 15 min: 71 synapses. **b,c**, Average MFI of either Zap70 (pY292) or PLCγ1 (pY783) on (**b**) WT CD61+ T cell clone (from patient 1) following treatment with either 25, 50 or 100 nM CD61 siRNA, or no treatment (**b**), as well as on WT CD61+, CD61KO or WT CD61−T cell clones (from patient 1), by flow cytometry (**c**). **d**, Bar graph showing the average MFI of Zap70 (pY292) on CD61+ T cell lines, following treatment with αCD61 (neutralizing treatment), IgG isotype control treatment or no treatment, by flow cytometry. *P* values: (patient 1: 0.00076, patient 2: 0.0083, patient 3: 0.0096, patient 4: 0.00031, patient 5: 0.0087, patient 6: 0.037, patient 7: 0.036). **e**, Average MFI of Lck on WT CD61+, CD61KO or WT CD61+ T cell clones (from patient 1), including on WT CD61+ T cell clone treated with 25 nM, 50 nM or 100 nM CD61 siRNA, or no treatment, by flow cytometry. **f**, Bar graph showing the average MFI of Lck on CD61+ T cell lines, following treatment with αCD61 neutralizing treatment, IgG isotype control treatment or no treatment, by flow cytometry. *P* values: (patient 1: 0.047, patient 2: 0.0059, patient 3: 0.011, patient 4: 0.049, patient 5: 0.048, patient 6: 0.0053, patient 7: 0.038). **g**, Representative histogram of phosphorylated Zap70 (pY292) on WT CD61+ T cell clone (from patient 1) following activation with or without Lck inhibition when using A770041 (right), or with genistein as positive control of tyrosine kinases inhibition (left), **h**, Schematic of CD61 modulation of Zap70 phosphorylation via Lck activity under no inhibition (left), after CD61 knock-down (middle) and Lck inhibition (right). Created with BioRender.com. **d,f**, *n* = 7 patients examined, three independent experiments. **b,c,e,g**, *n* = 3 independent experiments. **a–f**, Data are presented as the median ± s.e.m., ***P < 0.001, **P < 0.01, *P < 0.05, one-way ANOVA with Tukey's multiple-comparison test.

amino genistein as a positive control. As expected, we observed a significant impairment of the Zap70 phosphorylation on the WT CD61⁺ T cell clone following the inhibition of Lck (Fig. 3g). Taken altogether, these observations suggest that CD61 may enhance TCR proximal signaling, including Zap70 phosphorylation in an Lck-dependent manner (Fig. 3h).

## CD61 improves T cell cytotoxicity and tumor control

To determine the importance of CD61 on T cell antitumor immunity, we performed in vitro T cell degranulation and cytotoxicity analyses. We found that the cancer-specific WT CD61⁺ T cell clone had significantly elevated expression of a cytolytic degranulation marker, CD107a, compared to the T cells treated with CD61 siRNA, CD61^KO and the WT

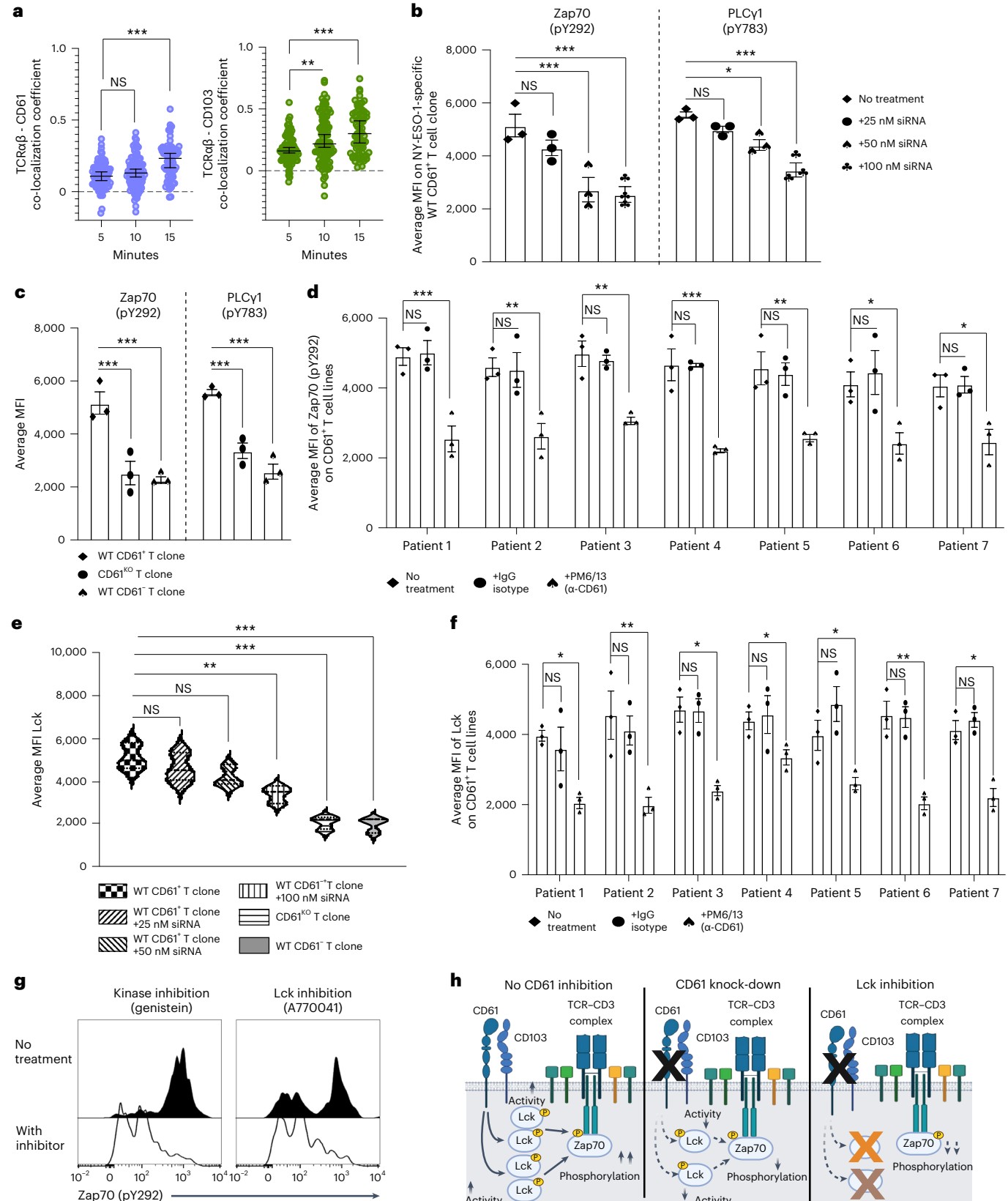

CD61⁻ T cell clones (Fig. 4a). Additionally, neutralizing CD61 with blocking antibody also limited the degranulation capacity of the CD61⁺ T cell, consistently observed across the CD61⁺ T cell lines from seven different patients with cancer (Fig. 4b).

Consistent with the increased degranulation activity, we observed a higher frequency of cancer cell death induced by the WT CD61⁺ T cell clone across multiple time points, with CD61 siRNA-treated and CD61^KO T cell clones exhibiting impaired T cell cytotoxicity (Fig. 4c). To validate the importance of CD61 toward T cell cytotoxicity, we next treated WT CD61⁺ T cell lines from seven different patients with cancer with anti-CD61 neutralizing antibody before the co-culture with antigenic cancer cells. We confirmed that neutralizing anti-CD61 antibody consistently limited the cytotoxic responses of these T cell lines (Fig. 4d).

We further assessed the in vivo physiological relevance of CD61⁺ T cells by evaluating the growth of xenografted antigenic tumors in NOD.SCID mice following adoptive transfer of either WT CD61⁺ or CD61⁻ T cell clones over time (Extended Data Fig. 5d). Tumor growth was significantly mitigated in mice injected with the CD61⁺ T cells compared to the CD61⁻ T cells (Fig. 4e). Importantly, the differences in tumor sizes were readily observed after the second adoptive transfer of T cells, with more substantial differences seen following the third T cell injection (Fig. 4f,g).

Since these in vitro and in vivo findings demonstrated the immune contribution of CD61⁺ cancer-specific T cells toward tumor control, we explored the possible clinical relevance of the CD61⁺ TILs on patients' overall survival (OS) probability, in a cohort of patients with lung cancer (LC) and a cohort of patients with skin cutaneous melanoma (SCM) from The Cancer Genome Atlas (TCGA) database[12,22–24]. We found that the CD61^hiCD103⁺CD8⁺CD3⁺ patients with SCM exhibited better OS prognosis compared to the CD61^loCD103⁺CD8⁺CD3⁺ patients (Fig. 4h). In validating this survival pattern, we further found that the CD61^hiCD103⁺CD8⁺CD3⁺ patients with LC also had improved OS prognosis compared to the CD61^lo CD103⁺CD8⁺CD3⁺ patients (Fig. 4h).

Taken altogether, the in vitro, in vivo and clinical findings indicated a unique role of CD61 in promoting T cell cytotoxicity, mitigating tumor growth and improving the OS in patients.

## CD61⁺ TILs have enhanced effector phenotypes

To further dissect the relevant clinical immunophenotype of the CD61⁺ T cells that may contribute toward enhanced cancer immunity and improved survival in patients with cancer, we performed multicolor flow cytometric profiling of tumors from 19 patients with NSCLC. We first stratified the CD61⁺ and CD61⁻ T_RM TILs according to the well-established tissue-resident memory TIL phenotype of CD103⁺CD69⁺CD49a⁺CD45RO⁺ (CD62L⁻CCR7⁻CD45RA⁻) CD8⁺ TILs (Extended Data Fig. 2).

Clinical immunophenotypic analyses showed significant upregulation of key antitumor effector cytokines, chemokines and cytolytic molecules (including granzyme M, granulysin, granzyme B, CD107a, CCL5, XCL2, TNF and IFN-γ) on CD61⁺ TILs, compared to the CD61⁻ TILs (Fig. 5a,b and Extended Data Fig. 6). Additionally, we confirmed that, in particular, granulysin and granzyme M expression on CD61⁺ TILs was dependent on CD61 activity, as treatment using an anti-CD61 blocking antibody demonstrated impaired expression of both cytokines (Fig. 5c).

The CD61⁺ TILs have significantly enriched combinatorial immune effector signatures compared to the CD61⁻ TILs (Fig. 5d). As upregulated immune effector signatures on TILs are highly indicative of tumor responsiveness, we hypothesized that CD61⁺ TILs could be more infiltrative of the NSCLC tumor bodies. Using an in situ IHC approach, we confirmed that the CD61⁺ TILs (identified by CD61⁺CD103⁺CD8⁺ colocalized cells) were significantly present at higher frequency within tumor islets compared to the CD61⁻ TILs (identified by CD61⁻CD103⁺CD8⁺ colocalized cells; Fig. 5e).

Previous studies on TILs have established tumor-reactive TILs as marked by dual-positive CD103⁺CD39⁺ expression[12–14]. In the analysis of our NSCLC patient cohort, we found significant enrichment of these combinatorial markers on the CD61⁺ TILs compared to the CD61⁻ TILs (Fig. 5f), therefore suggesting that the TILs subset is likely immune reactive within the tumor microenvironment.

## CD61⁺ TILs do not exhibit hallmarks of exhaustion

The tumor microenvironment is known to be immunosuppressive, and this is well established to contribute to chronic T cell exhaustion. The hallmarks of T cell exhaustion include (i) reduced antigen sensitivity, (ii) regression of effector responses, (iii) terminal stage of differentiation and, most importantly, (iv) coexpression of multiple immune inhibitory receptors[15,25–27].

As shown in Fig. 5, the CD61⁺ TILs do not have regression of effector responses but instead exhibited an enhanced immune effector phenotype. Therefore, we next evaluated the other hallmarks of cancer T cell exhaustion that may be exhibited by the TILs subset, namely the coexpression of multiple inhibitory receptors such as Tim-3, PD-1 and TIGIT, which we have shown previously to be the most prominent inhibitory receptors coexpressed on total CD8⁺ TILs in a variety of cancers including NSCLC[28]. Interestingly, the CD61⁺ TILs exhibited enriched PD-1 expression, but reduced expression of Tim-3 and TIGIT, when compared to the CD61⁻ TILs (Fig. 6a). This observation was confirmed by the limited frequency of combinatorial expression of PD-1⁺Tim-3⁺TIGIT⁺ by CD61⁺ TILs, compared to the CD61⁻ TILs (Fig. 6b). Instead, the CD61⁺ TILs were highly enriched for the PD-1⁺Tim-3⁻TIGIT⁻ population (Fig. 6c). As PD-1 is also well established to be a marker of activation on T cells[29,30], our current finding suggests that the PD-1⁺Tim-3⁻TIGIT⁻ T_RM TIL population could be less exhaustive, and therefore more active and responsive in lung cancers.

Consistent with its more active and responsive nature, we found that CD61⁺ TILs were not at the terminal stage of differentiation

**Fig. 4 | CD61 improves cytotoxicity and tumor control. a**, Bar plot of CD107a MFI between NY-ESO-1-specific WT CD61⁺, CD61^siRNA-treated, CD61^KO and WT CD61⁻ T cell clones (from patient 1) following activation with antigenic cancer cells, by flow cytometry. ◆, WT CD61⁺ T cell clone; •, WT CD61^siRNA-treated T cell clone, ♠, WT CD61^KO T cell clone; ♣, WT CD61⁻ T cell clone. **b**, Horizontal bar graph showing CD107a MFI on CD61⁺ T cell lines following αCD61 neutralizing antibody treatment, IgG isotype control treatment or no treatment (n = 7 patients), by flow cytometry. P values: (patient 1: 0.00078, patient 2: 0.00089, patient 3: 0.0099, patient 4: 0.0053, patient 5: 0.0057, patient 6: 0.033, patient 7: 0.021). **c**, Line plot showing the percentage of cancer cell death, following co-culture with NY-ESO-1-specific WT CD61⁺, CD61^siRNA-treated, CD61^KO or WT CD61⁻ T cell clones (from patient 1). **d**, Horizontal bar graph showing the percentage of cancer cell death, following co-culture with CD61⁺ T cell lines after αCD61 neutralizing antibody treatment, IgG isotype control treatment or no treatment (n = 7 patients). P values: (patient 1: 0.0078, patient 2: 0.0043, patient 3: 0.0097, patient 4: 0.047, patient 5: 0.049, patient 6: 0.04, patient 7: 0.007). **e**, Kinetic analysis of mouse tumor volume after adoptive transfer of either WT CD61⁺ or WT CD61⁻ T cell clones (patient 1). The arrows show the timepoints of T cell injections. **f,g**, Dot plots of mouse tumor volume after the 2nd (day 10, P value: 0.048) or 3rd (day 16, P value: 0.0089) T cell injection of either WT CD61⁺ (black box) or WT CD61⁻ (white box) T cell clones (from patient 1). **h**, Kaplan–Meier survival curves of patients with SCM and patients with stage 1 LC using TCGA dataset. Patient groups: (i) patients with CD61⁺CD103⁺CD8⁺CD3⁺ samples (G1, light red), or (ii) patients with CD61⁻CD103⁺CD8⁺CD3⁺ samples (G2, light blue). The starting number of patients with SCM analyzed: n_G1 = 16 patients, n_G2 = 122 patients. The starting number of patients with LC analyzed: n_G1 = 85 patients, n_G2 = 22 patients. One-way ANOVA, with Tukey's multiple-comparison test. **a,c**. n = 3 independent experiments. **b,d**. n = 7 patients examined over three independent experiments. **e–g**, n_+WT CD61⁺ T cells = 8 mice, n_+WT CD61⁻ T cells = 10 mice. **a–g**, Data are presented as the median ± s.e.m., denoted as \*\*\*P < 0.001, \*\*P < 0.01, \*P < 0.05, with either one-way ANOVA with Tukey's multiple-comparison tests (**a–e**) or two-tailed t-test with Wilcoxon adjustment (**f** and **g**).

(another hallmark of T cell exhaustion). Instead, they were enriched for early-differentiated cells (CD27⁺CD28⁺; Fig. 6d). In contrast, the CD61⁻ TILs were predominantly at the late stage of differentiation.

Being at the earlier stage of maturation, we further demonstrated that the CD61⁺ TILs were capable of undergoing more cellular divisions compared to the CD61⁻ TILs, with an increased frequency of

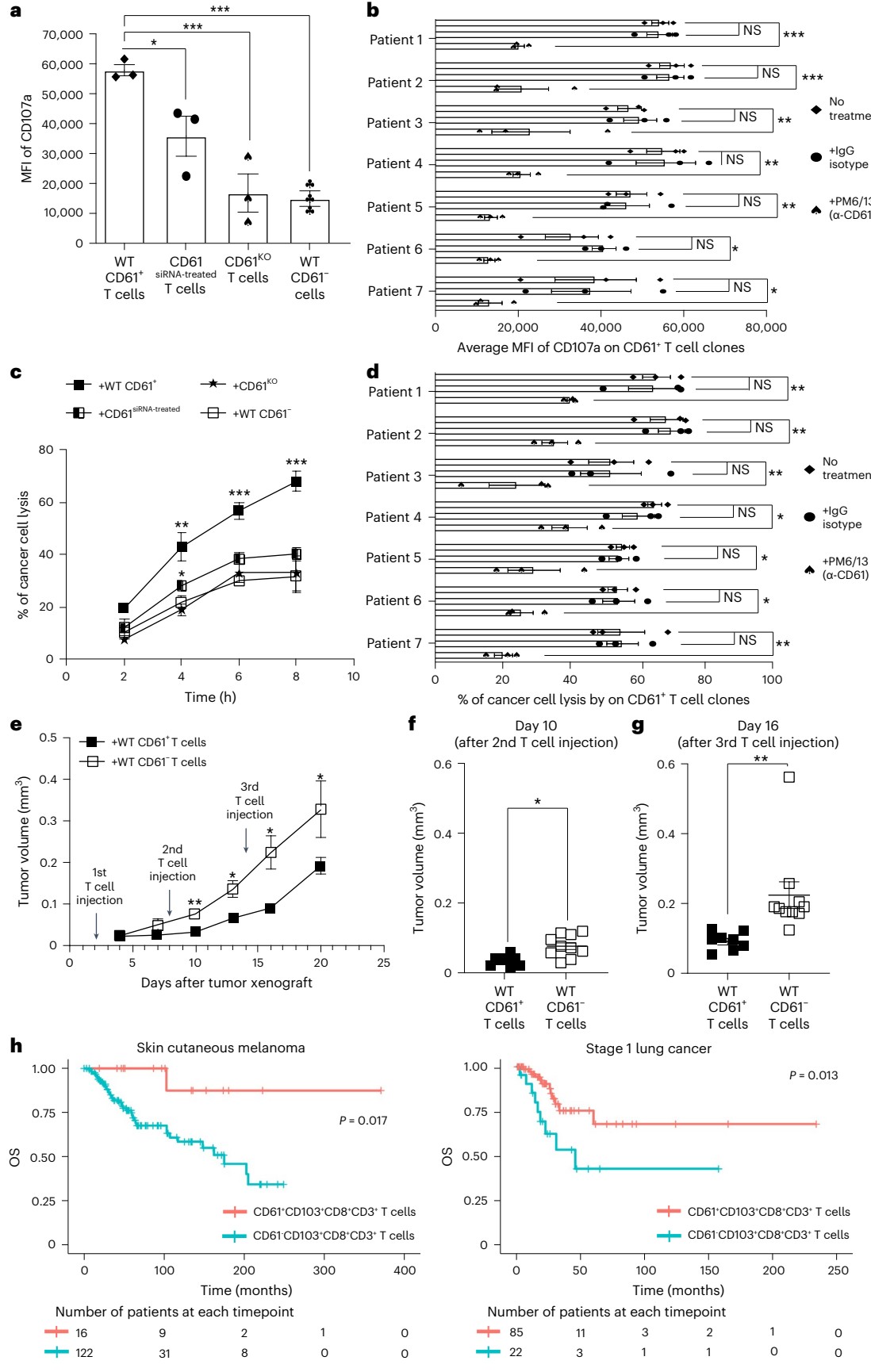

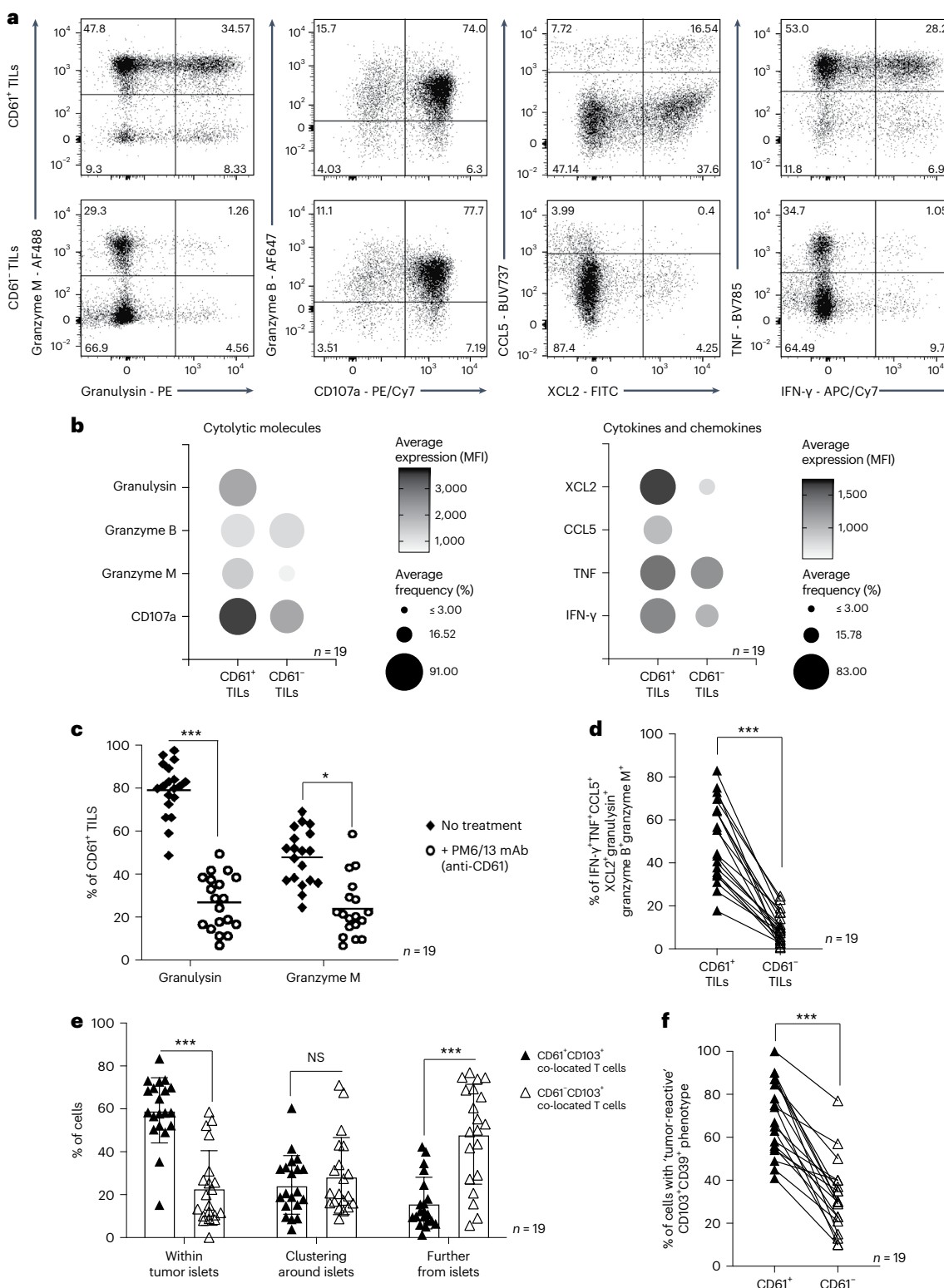

**Fig. 5 | CD61⁺ TILs have enhanced antitumor effector phenotypes in NSCLC.**
**a**, Expression of cytolytic molecules (granulysin, granzyme M, granzyme B), degranulation marker CD107a, chemokines (CCL5, XCL2) and cytokines (TNF, IFN-γ) between CD61⁺ and CD61⁻ TILs, by representative flow cytometry plots of 1 patient (patient 7). **b**, Dot plots of the average MFI and frequency of cytolytic molecules, cytokines and chemokines, by flow cytometry. **c**, Dot plot showing the percentage of CD61⁺ TILs expressing granulysin and granzyme M following ex vivo αCD61 neutralizing antibody treatment or no treatment ex vivo. *P* value granulysin: 0.00021, *P* value granzyme M: 0.045. **d**, Line plot on the frequency of combinatorial effector signatures positive (IFN-γ⁺TNF⁺CCL5⁺XC

L2⁺granzyme M⁺granzyme B⁺granulysin⁺) cells between CD61⁺ and CD61⁻ TILs. *P* value: 0.00087. **e**, The frequency of CD61⁺CD103⁺CD8⁺ co-located cells or CD61⁻CD103⁺CD8⁺ co-located cells present within the tumor body, clustering around the tumor body, or further from the tumor body, by IHC. Data are presented as the median ± s.e.m. *P* value (within tumor islets): 0.0009, *P* value (further from islets): 0.00078. **f**, Line plot on the frequency of cells with tumor-reactive combinatorial markers expression (CD39⁺CD103⁺) between CD61⁺ and CD61⁻ TILs. *P* value: 0.00074. **c**–**f**, ***P* < 0.001, **P* < 0.05; one-way ANOVA with Tukey's multiple-comparison test (**c** and **e**) or two-tailed *t*-test with Wilcoxon adjustment (**d** and **f**). **b**–**f**, *n* = 19 patients. mAb, monoclonal antibody.

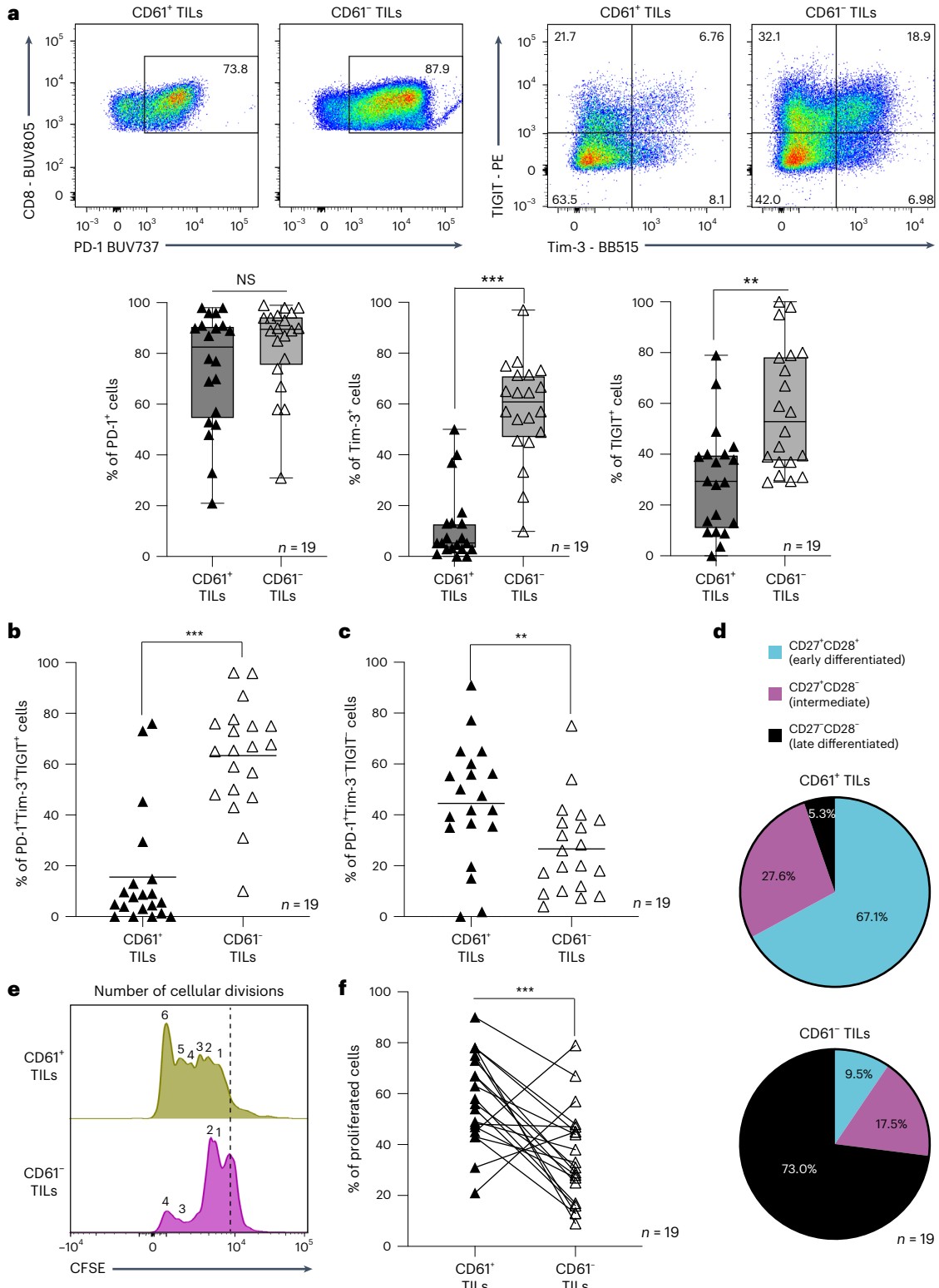

**Fig. 6 | CD61⁺ TILs do not exhibit hallmarks of exhaustion. a**, The expression of PD-1, TIGIT and Tim-3 by representative flow cytometry plot of patient 6 (top) and the frequency of PD-1⁺, Tim-3⁺ and TIGIT⁺ cells by box-and-whisker plots (bottom) between CD61⁺ TILs and CD61⁻ TILs. Data are presented as the median ± s.e.m., with center percentile. *P* values (Tim-3: 0.00045, TIGIT: 0.0034). **b,c**, Dot plots on the frequency of triple coexpressed PD-1⁺Tim-3⁺TIGIT⁺ cells and PD-1⁺Tim-3⁻TIGIT⁻ cells between CD61⁺ TILs and CD61⁻ TILs. **b**, *P* value: 0.00083. **c**, *P* value: 0.0089. **d**, Pie charts showing percentages of cells in the early-differentiated stage (CD27⁺CD28⁺), intermediate differentiated stage (CD27⁺CD28⁻) and late differentiated stage (CD27⁻CD28⁻) between CD61⁺ TILs and CD61⁻ TILs. **e**, Representative histogram plot (from patient 7) of cells undergoing cellular divisions (denoted by the number of CFSE peaks) between CD61⁺ TILs and CD61⁻ TILs. Peaks at the dashed line represent cells that are not proliferating. **f**, Line plot of proliferated cells between CD61⁺ TILs and CD61⁻ TILs. *P* value: 0.00097. ***P* < 0.001, **P* < 0.01, two-tailed *t*-test with Wilcoxon adjustment (**a**–**c** and **f**). **a**–**d** and **f**, *n* = 19 patients.

proliferating cells (Fig. 6e,f). Taken altogether, these findings high-lighted CD61[+] TILs as proactive, tumor-responsive T cells exhibiting enhanced antitumor effector and cytotoxic immune responses but lack the expression of multiple immune checkpoint receptors (a key hallmark of TIL exhaustion). These positive attributes, therefore, are likely to contribute toward T cells' capacity to mitigate tumor growth and improve survival.

## Discussion

In this study, we demonstrated an unexpected CD61 expression on human cytotoxic CD8[+] T cells in cancer. With the absence of its *cis* inte-grins partners CD41 and CD51, CD61 can colocalize and pair transiently with CD103 at the cell-to-cell contacts. On the contrary, integrin β7 was segregated apart from the CD103, toward the distal synaptic space. The discovery using in vitro proteomics model was validated and confirmed using multifaceted approaches with a larger cohort of in situ, in vitro and ex vivo patient samples. We further demonstrated CD61 colocaliza-tion with the TCR, which augmented the proximal TCR signaling and contributed toward elevating antitumor cytotoxicity. This allowed for mitigation of tumor growth, evidenced in the immunocompromised mouse model used in this study. Clinically, the presence of CD61[+] TILs was associated with enhanced effector functions and phenotypes and limited hallmarks of cellular exhaustion.

The pairing between an integrin α and β subunit only become functionally active following cellular activation[2–5]. CD103 was thought to be exclusively paired with integrin β7 in mediating cel-lular adhesion, primarily between intraepithelial CD8[+] T cells and the E-cadherin-expressing epithelial and endothelial cells[3,31]. The CD103–integrin β7 pair functions similarly to another integrin pair found on T cells, the integrin αL–β2, which promotes synapse assembly and stability, particularly when they are present at the dSMAC and periph-eral supramolecular activation cluster (pSMAC)[32,33]. In contrast to these studies, we showed minimal co-clustering between CD103 and integrin β7 at the synapse. Instead, CD103 colocalized together with CD61 at the cSMAC. These observations suggest that CD103 is likely a promiscuous and dynamic integrin, that is not restricted to a single integrin β partner as previously thought.

The pairing and functions of CD61 with its canonical *cis*-integrin partners, CD41 and CD51, have been well documented on nonlympho-cytic cells, such as megakaryocytes, platelets and macrophages and endothelial cells[2,4,6–8]. While there is evidence of CD61 expression on murine T cells[9,10], the expression and functional implications of CD61 on human T cells were not known. Most remarkably, we observed that CD61 expression on human cancer-specific CD8[+] T cells can occur in the absence of CD41 and CD51. The CD61 was enriched in its colocalization with CD103 in the synaptic microclusters, as well as evidenced by the enrichment of both proteins in the Co-IP lysates and flow cytometry approaches. Significant to the field, our study uncovered an exam-ple of CD61 potential pairing with an I-domain-containing integrin α subunit, CD103.

CD61 upregulation was only observed on the CD103[+] T cells, but not on the CD103[−] T cells. The link between CD61 and CD103 coex-pression is likely due to their upregulation by TGF-β1. Not only has our recent study shown that the CD103[+] T cell clones can specifically express TGF-β1 (ref. 15), but others have also demonstrated that TGF-β1 is required for sustaining phenotypic expression of CD103 on certain human and murine cells[34–36]. In parallel, a recent study further showed that TGF-β1 can also upregulate CD61 mRNA and protein expression in a dose-dependent and time-dependent manner[37].

The enrichment of CD61 on the cSMAC alongside the TCR suggests that CD61 may be involved in co-stimulatory signals. This is because the dynamic cSMAC is well established as a centripetally enriched zone for the TCR and its associated signaling molecules to induce net signaling outcomes and cytolytic activity[38–42]. In contrast, the pSMAC, where most integrins such as integrin αL–β2 are known to be usually clustered and maintained, is primarily responsible for the assembly and firm adhesion of the synaptic structure[38,42–47]. While CD61 can signal via Fyn kinase[48], we showed that CD61 is involved in modulat-ing TCR-dependent ZAP70 phosphorylation, importantly through the intermediary Lck protein, providing an example of the potential mechanism by which CD61 can operate on T cells.

The transient nature of CD61 expression implicates the high turnover rate of this protein and the dynamic pairing with an I-domain-containing α integrin subunit. A recent study on an unconven-tional integrin pairing between CD51 and CD29 has suggested that low intra-heterodimer integrin affinity can lead to better functional activ-ity[49]. For CD61, potentially low-affinity interactions with its partners, such as CD103, are likely needed depending on different situations and cell types. For example, CD61 interaction with a non-integrin partner, the heparan sulfate proteoglycans, can prompt vesicular endocytosis, leading to internalization and losing affinity interaction[50].

Our study shows that the immune potency and functions of CD61 on human T cells are spatially, temporally and TCR activation depend-ent. At least in this model, the TCR–pMHC interactions are necessary to trigger the temporal recruitment of CD61, via CD103, to the cell surface and the spatial reorganization and function within the synapse. This fine spatiotemporal regulation of CD61 and its kinetic association with the TCR and its signaling may explain why CD61[+] TILs in patients with NSCLC exhibited elevated antitumor activities and a proliferative immunophenotype but a limited cellular exhaustion phenotype. This may therefore contribute toward better tumor control. However, how CD61 associates with the T cell infiltrate merits further investigation.

The differential protective capacity of the cancer-specific CD61[+] and CD61[−] T cells may additionally be explained by the epigenetic reprogramming differences between both cells that make the CD61[−] T cells less functional. For example, our recent study demonstrated that certain cancer-specific (SSX-2-specific) T cell clones were defective in their antitumor effector cytokines production compared to other T cell clones that shared the same TCR repertoire, likely contributed by the accumulating CpG hypermethylation on the *IFNG* gene pro-moter[27]. However, the CD61 association with its epigenetic phenotype is not within the remit of this study but merits further investigation in the future.

The initial proteomics approach indicates several other pro-teins than CD61 that may contribute further to functions. However, the role of these proteins in regulating CD103[+] T cell functions and activities, beyond that of CD61 is not within the remit of this study and future investigations are warranted. However, our study has vali-dated the possibility of an unexpected integrin pairing, for example, between CD61 and CD103 not only on human immune cells, such as antigen-specific T cells. Therefore, this study is significant by broad-ening our understanding of the unexpected dynamics of integrins on immune cells, including in regulating tissue homeostasis, disease pathogenesis, immunity and cellular biophysical protein–protein interactions. Future immunotherapy strategies and translational works targeting these proteins may enlighten the possibility of uti-lizing CD61 and its unconventional immune roles, to promote and provide protective immunity.

## Online content

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

## Methods

### Human ethics approval

Patients with confirmed non-metastatic NSCLC were recruited from the John Radcliffe Hospital, Oxford, United Kingdom, between December 2020 and April 2021. Participants included both females and males who were between 63 and 80 years old. Ethics was approved by the NHS South Central – Oxford C Research Ethics Committee (REC no. 19/SC/0173) under the ORB Tissue Access Committee ethics reference numbers 18/A026 and 20/A081. All procedures were performed according to the Declaration of Helsinki guidelines. Clinical parameters of individual participants are described in Extended Data Fig. 3. Participants provided voluntary informed written consent before surgery, and no compensation was provided. Tumor resection volume was no more than 90 mm³ and paratumor resection weight was a maximum of 0.1 g. Tissues were stored in RPMI-1640 on ice and de-identified before tissue processing. Tumors were confirmed using immunohistology by the ORB. In total, 31 patients were used for immunohistochemistry analysis and fresh samples from 19 of the same patients were used for flow cytometry analysis.

### Peripheral blood and tissue processing

Cell suspensions were isolated from tissue using methods described previously[15]. Briefly, tissues were cut into small pieces before enzymatic dissociation using the human tumor dissociation kit (Miltenyi Biotech), following the protocol provided by the supplier. Cells were then filtered through a 100-µm strainer to remove indigestible parts, with dead cells and debris removed by centrifugation at 250g for 10 min. Cells were then resuspended in RPMI-1640 supplemented with 10% FCS (Sigma-Aldrich), 2 mM L-glutamine (Sigma-Aldrich) and 1% vol/vol (500 U ml⁻¹) penicillin–streptomycin (Sigma-Aldrich; R10). For peripheral blood, the peripheral blood mononuclear cells were isolated using Ficoll-Hypaque gradient isolation as described previously[20].

### Patient-derived cancer-specific CD8⁺ T cells

HLA-A*02:01-restricted NY-ESO-1$_{157-165}$-specific, SSX-2$_{41-49}$-specific, Tyrosinase$_{369-377*N370Dvariant}$-specific and melan-A/MART-1$_{26-35*A27L}$-specific T cells were generated from patients with gastric cancer and melanoma, as previously described[15,51]. Briefly, mononuclear cells were stimulated with tumor-associated antigens of the following: 10 µg ml⁻¹ SSX-2$_{41-49}$-specific KV9 peptide (KASEKIFYV; PeproTech), NY-ESO-1$_{157-165}$-specific SC9 peptide (SLLMQITQC; PeproTech), melan-A/MART-1$_{26-35*A27L}$-specific EV10 peptide (ELAGIGILTV; PeproTech) or Tyrosinase$_{369-377*N370Dvariant}$-specific YV9 peptide (YMDGTMSQV) (JPT) in RPMI-1640 media, supplemented with 10% vol/vol heat-inactivated human AB serum (National Blood Service), 2 mM L-glutamine (Sigma-Aldrich), 1% vol/vol (500 U ml⁻¹) penicillin–streptomycin (Sigma-Aldrich; H10), recombinant human IL-2 (200 U ml⁻¹; PeproTech) and recombinant human IL-15 (0.5 ng ml⁻¹; PeproTech) for 14 days at 37 °C. After 14 days, antigen-specific T cells were sorted using PE-conjugated HLA-A*02:01/cancer peptide tetramers and cultured in vitro for another 14 days. The purity of sorted populations was confirmed by tetramer staining and confirmed to have >90% tetramer purity. Validated cancer-specific T cells were stored in several batches for future assays, tested for *Mycoplasma* monthly and re-authenticated using the tetramer staining method before every assay.

### CD61 manipulation on T cells

For knocking-down CD61, siRNA targeting the *CD61* transcript was purchased commercially (Thermo Fisher, 4392420, assay IDs: s7580, s7581, s7582; ref. 52). Triple siRNA targeting *CD61* treatment was performed according to the Lipofectamine RNAiMax protocol (Thermo Fisher). Briefly, the WT CD61⁺ T cell clone (from patient 1) was seeded in a 96-well round-bottom plate at 2 M ml⁻¹ with H10 and recombinant human IL-2 (200 U ml⁻¹; PeproTech). A serial dilution of siRNA was prepared in 100 µl Opti-MEM (for a final concentration of 25 nM, 50 nM and 100 nM

for each siRNA). In parallel but separately, 3 µl of the Lipofectamine RNAiMax reagent was added to 100 µl Opti-MEM before merging in a 1:1 ratio and incubated at room temperature (RT) for 20 min to form transfection complexes. The solution was then added dropwise onto cells solution 3–7 days before co-culture with target cancer cells. Validation data are shown in Extended Data Fig. 5a. Surface staining was performed 5 days after siRNA treatment and before each T cell functional assay, to ensure consistent CD61 downregulation on the T cells.

For knocking-out CD61, a CRISPR–Cas9 approach was taken. We used the WT CD61⁺ T cell clone that is 100% positive for CD61 expression as the cell input for treatment (Extended Data Fig. 5a). Ablation of the gene of interest, *ITGB3*, was achieved by transfection with Cas9–gRNA ribonucleoprotein (RNP) complexes. *ITGB3* gRNA was commercially purchased from Integrated DNA Technologies (IDT; Hs.Cas9.ITGB3.1.AB, Hs.Cas9.ITGB3.1.AC and Hs.Cas9.ITGB3.1.AD). Before transfection, WT CD61⁺ T cell clone (from patient 1) was washed three times with ten volumes of prewarmed Opti-MEM-I medium (Thermo Fisher). Cells were resuspended to a final concentration of 3 × 10⁷ cells per milliliter. In parallel, RNP complexes were assembled in two steps. First, 200 pmol of Alt-R CRISPR–Cas9 tracrRNA (200 µM stock, IDT) was mixed with 200 pmol of Alt-R CRISPR–Cas9 predesigned ITGB3 crRNA (200 µM stock, IDT), and incubated at 95 °C for 5 min, with the resultant duplex guide RNA allowed to cool to RT. The duplex gRNAs were then mixed with 124 pmol of Alt-R *Streptococcus pyogenes* CRISPR–Cas9 Nuclease V3 (IDT) and incubated at 37 °C for 15 min. The resultant RNPs were allowed to cool to RT and then supplemented with 200 pmol of Alt-R Cas9 Electroporation Enhancer (200 µM stock, IDT). The input cell (WT CD61⁺ T cell clone) was mixed with the RNP solution and immediately transferred to a 2-mm cuvette (Bio-Rad), electroporated at 290 V for 2 ms using an ECM 830 Square Wave electroporator. Cells were then cultured with prewarmed, H10 supplemented with 100 U ml⁻¹ recombinant human IL-2 for 5 days. The 5-day culture after CRISPR–Cas9 addition was made to ensure full degradation of preexisting and synthesized CD61. Cell sorting was then performed using CD61 marker, to enable the selection of truly CD61 negative (CD61$^{KO}$) cells from the initial 100% positive WT CD61⁺ T cell input. On the foundational basis that a gene deletion leads to the absence of protein, we carried out flow cytometry staining on the sorted cells at four different passages. Based on Extended Data Fig. 5a, while the WT CD61⁺ T cell (input cell) was 100% positive for CD61, the sorted cell stained at each passage was absent of CD61, confirming the KO effect. Additionally, surface staining was performed regularly at the start of all functional assays to ensure consistent CD61 abrogation on the T cells (Extended Data Fig. 5a).

### Mice

Immunodeficient NOD SCID gamma (NSG) mice (strain NOD.Cq-Prkdc scid Il2rg$^{tm1Wjl/SzJ}$), both male and female, were bred locally at the Department of BioMedical Services (BMS), University of Oxford. Details of the experimental model and in vivo assay used are according to previous study[53]. All mice were housed in ventilated cages, maintained under specific pathogen-free conditions, with a 12-h dark–light cycle, ambient RT between 18 and 23 °C, 40–60% humidity and used at 8–10 weeks of age. The mouse diet was commercially sourced from Safe-Lab, Germany (A03 Safe Diet). All mouse experiments were performed following the Animals (Scientific Procedures) Act 1986 and according to the University of Oxford Animal Welfare and Ethical Review Body (AWERB) guidelines and operating under the UK Home Office PPL license PBA43A2E4. All tumor xenograft and tumor burden experiments complied with the abovementioned mouse background.

### NY-ESO-1-specific TCR-T cell generation

The NY-ESO-1 TCR sequence used is described in a previous paper[15]. DNA templates were designed in silico and synthesized by GeneArt (Thermo Fisher Scientific). The plasmids were used directly as the

repairing template. The TCR construct for CRISPR–Cas9-mediated HDR repair was designed with the following structure: 5′ homologous arm, P2A, TCR-β, T2A, TCR-α, bGHpA tail, 3′ homologous arm. To facilitate TCR expression, the TCR sequence was codon optimized and sequence confirmed by Sanger sequencing. Both 5′ and 3′ homologous arm sequences were used as previously described[54]. Peripheral blood mononuclear cells were isolated from the peripheral blood of a healthy human donor using Ficoll-Hypaque gradient isolation. Primary CD8+ T cells were then isolated using the CD8+ T cell isolation kit (Miltenyi Biotec) before being activated in vitro with 25 µl ml⁻¹ ImmunoCult Human CD3/CD28 T cell activator (StemCell Technologies, 10791) for 2 days. NY-ESO-1 transgenic T cells were generated by using an orthotopic TCR replacement system with modifications[54]. Briefly, exogenous NY-ESO-1 TCR was inserted into the primary T cell *Trac* gene locus, together with the blockage of *Trbc* gene expression. CRISPR gDNA sequences used were: 5′-AGAGTCTCTCAGCTGGTACA-3′ for *Trac* and 5′-GGAGAATGACGAGTGGACCC-3′ for *Trbc* (targeting both *Trbc1* and *Trbc2*). Two-day activated T cells were harvested and washed with PBS before resuspension in P3 Primary Cell Nucleofector Solution (Lonza). The CRISPR RNP complex was generated with sgRNA (IDT) and Alt-R *S. pyogenes* Cas9 Nuclease V3 protein (IDT) by incubation at RT for 15–20 min. Cells were then electroporated with CRISPR RNPs in the presence of DNA HDR repairing template using the 4D Nucleofector X unit (Lonza). After electroporation, cells were plated and incubated with prewarmed allogenic feeders. After a week, cells were sorted using the NY-ESO-1 tetramer on the BD LSR Fusion (BD Biosciences). The sorted TCR-T cells were confirmed for TCR antigen specificity using the tetramer approach and confirmed for purity >90%.

## Integrin lentiviral overexpression system

Briefly, LentiX cells were plated in six-well plates at 650,000 cells per well in DMEM supplemented with 10% FCS, 2 mM L-glutamine and 1% vol/vol (500 U ml⁻¹) penicillin–streptomycin and incubated for overnight at 37 °C. Cells were next co-transfected with the packaging plasmids pMD2G (0.26 µg per well; Addgene plasmid, 12259) and psPAX2 (0.5 µg per well; Addgene, plasmid 12260), as well as the relevant lentiviral expression vector plasmid (at 0.75 µg per well; pHR-SIN plasmid backbone (Addgene, plasmid 79121)) in Opti-MEM and FuGENE HD transfection reagent. The full-length gene sequence of *ITGAE, ITGB3* or *ITGB7* (with relevant tag) DNA fragments were custom-purchased from IDT and Thermo Fisher. The full-length protein sequences were obtained from Uniprot. DNA sequence integrity and identity were confirmed by Sanger sequencing. To improve the transfection efficacy, ViralBoost Reagent (Alstem) was added to the LentiX cell culture medium at the time of transfection. Lentiviruses were harvested 72 h after transfection and dead cells were removed by centrifugation at 1,500g for 5 min. Lentiviruses were placed on ice following harvesting. Then, 0.5 M of freshly isolated primary CD8+ T cells from a healthy donor were first activated overnight with 10 µg per well of OKT3 (BioLegend) or 10 µl per well of αCD3/CD28 (StemCell Technologies). The overnight-activated T cells were cultured with 3 ml of respective lentivirus in a T25 flask upright and incubated at 37 °C for a minimum of 2 h. Then, 2 ml H10 was added before further incubation for 5 days. Cells were then collected and stained with live/dead staining before cell surface staining (for the list of antibodies, see the Nature Portfolio Reporting Summary). Cells were then permeabilized with BD CytoFix/CytoPerm Solution for 20 min at 4 °C before intracellular staining and fixed with BioLegend's FLouriFix buffer. Samples were then acquired on an Attune Nxt flow cytometer v3.2.1 (Thermo Fisher) and analyzed on FlowJo v.10.5.3 (BD Biosciences).

## Liquid chromatography–mass spectrometry

Paired CD103+ and CD103− T cell clones were activated for 3–6 h with 10 µl αCD3/CD28 (StemCell Technologies) at 37 °C, with non-activated T cells as a normalization control. To retain proteins and prevent

secretion of molecules, 0.7 µg ml⁻¹ Monensin and 1 µg ml⁻¹ Brefeldin A (BD Biosciences) were added per sample before activation. Cells were then washed with PBS thoroughly three times before they were lysed with 1% NP-40 cell lysis buffer (Thermo Fisher), 1× protease inhibitor cocktail (Sigma-Aldrich) and 1 mM phenylmethylsulfonyl fluoride (Thermo Fisher) on ice for 1 h. The solution was vortexed in 10-min intervals during the ice incubation, before microcentrifugation at 8.0g for 10 min at 4 °C. Supernatants were transferred into new tubes and frozen on dry ice and stored at −80 °C. Samples were thawed and proteins were denatured in 8 M urea for 30 min. Protein reduction was performed with 10 mM tris(2-carboxyethyl(phosphine) for 30 min at RT before undergoing alkylation with 50 mM iodoacetamide for another 30 min at RT in the dark. Samples were then diluted with 1.5 mM urea and 50 mM triethylammonium bicarbonate (TEAB) before digestion with 1.5 µg trypsin and incubation overnight at 37 °C. Digested samples were cleaned on a SOLA HRP C18 and evaporated to dryness using a vacuum centrifuge. Samples were then reconstituted in 5% dimethylsulfoxide and 5% formic acid. Samples were analyzed using an Ultimate 3000 UHPLC (Thermo Fisher Scientific) connected to an Orbitrap Fusion Lumos Tribrid instrument control software v3.3 (Thermo Fisher Scientific). Peptides were loaded onto a trap column (PepMapC18; 300 µm × 5 mm, 5-µm particle size; Thermo Fisher) and separated on a 50-cm-long EasySpray column (ES803, Thermo Fisher) with a gradient of 2–35% acetonitrile in 5% dimethylsulfoxide and 0.1% formic acid at a flow rate of 250 nl min⁻¹ over 60 min. Eluted peptides were then analyzed on an Orbitrap Fusion Lumos Tribrid platform (instrument control software v3.3). Data were acquired in data-dependent mode, with the advanced peak detection mode enabled. Survey scans were acquired in the Orbitrap at a resolution of 120,000 over a $m/z$ range of 400–1500, AGC target of 4e5 and S-lens RF of 30. Fragment ion spectra (MS/MS) were obtained in the Ion trap (rapid scan mode) with a Quad isolation window of 1.6, 40% AGC target and a maximum injection time of 35 ms, with HCD activation and 28% collision energy. For CD61 interactomics, CD61 Co-IP lysates (described further in Co-IP section) were analyzed using S-trap (Protifi). Proteins were reduced with 10 mM dithiothreitol in $H_2O$, followed by alkylation with 20 mM iodoacetamide in $H_2O$ in the dark. Samples were acidified by addition of 12% phosphoric acid (to a final concentration of ~1.1%), diluted with 90% methanol in 100 mM TEAB (640 µl of methanol mixture per 100 µl of sample), and captured on S-TrapTM mini columns (ProtiFi). Columns were washed with 90% methanol in 100 mM TEAB followed by centrifugation at 4,000g (400 µl per column x3). Captured proteins were digested with trypsin (1:30 wt/wt) overnight at RT. Peptides were first eluted with 50 mM TEAB (80 µl, 4,000g for 1 min), followed by elution with 0.5% trifluoroacetic acid (TFA) in $H_2O$ (80 µl, 4,000g for 1 min) and finally eluted with a 50:50:0.5 acetonitrile:MilliQ:TFA mixture, and dried in a vacuum concentrator. Dried peptides were dissolved in buffer A (98% MilliQ-$H_2O$, 2% $CH_3CN$ and 0.1% TFA). Around 2.2% of the tryptic peptides were analyzed by liquid chromatography–tandem mass spectrometry (LC–MS/MS) using a U3000 HPLC connected to an Orbitrap Ascend tribrid instrument (Thermo Fisher), loaded onto a PepMacC18 trap column (300 µm × 5 mm, 5-µm particle size, Thermo Fisher) and separated on a 50-cm EasySpray column (ES803, Thermo Fisher) using a 60-min linear gradient from 2% to 35% acetonitrile, 0.1% formic acid and at a flow rate of 250 nl min⁻¹. MS data were acquired in data-independent mode (DIA) with minor changes from a previously described method[55,56]. Briefly, MS1 scans were acquired in the Orbitrap over the mass range of 350–1650 $m/z$, with a resolution of 45,000, maximum injection time of 91 ms, an AGC set to 125% and an RF lens at 30%. MS2 scans were then collected using the tMSn scan function, with 40 variable width DIA scan windows at an Orbitrap resolution of 30,000, normalized AGC target of 1,000%, maximum injection time set to auto and a 30% collision energy.

## Proteomics analysis

Raw mass spectrometry files were label-free quantified using DIA-NN (version 1.8) in library-free mode using the Uniprot proteome

UP000005640 (2022) as a FASTA file. Data were further processed in Perseus (version 1.6.2.3). The $\log_2$ fold-change values of each protein were calculated by normalizing with non-activation sample control. TGF-β1 fold-change values were used as the threshold to exclude any proteins with lower fold-change values than TGF-β1 stimulated cells. This method was used because we have previously identified TGF-β1 as a protein exclusively expressed by the CD103[+] T cell clones, but not by the CD103[−] T cell clones[15]. Therefore, it can be assumed that any proteins with values lower than that of TGF-β1 are least likely to be expressed by the CD103[+] T cell clones. We cross-referenced these proteins to the Gene Ontology NCBI annotation database (https://geneontology.org/), categorizing the proteins according to their known biological activities. We utilized the STRING interactions database (https://string-db.org/) to stratify the proteins into their specific functional protein subgroups. A heat map analysis of selected proteins was carried out using R. Combined network plots for 3 h versus 0 h and 6 h versus 0 h for the paired NY-ESO-1-specific T cell clones gene list were converted to Entrez IDs of protein (org.Hs.eg.db version 3.11.4). An upregulated proteins list was used as input for overrepresentation analysis (clusterProlifer version 3.18.0, ReactomePA version 1.32.0) to find REACTOME pathways with enriched proteins (with $P$-value cutoff of 0.01 and adjusted $P$-value cutoff of 0.05)[57–59]. The resulting output was used to create a concept network plot. Bar plots were constructed for selected proteins of a pathway using $\log_2$ fold-change values for specific proteins (ggplot2 version 3.3.2). Volcano plots were generated using the processed data and plotted using VOlcaNOseR v1.0.3 (https://huygens.science.uva.nl/VolcaNoseR/).

## Multicolor flow cytometry immunophenotyping and analysis

A total of 1 M cells of paratumor tissue, tumor and peripheral blood were first stained with Live/Dead Fixable Aqua Stain Kit (Thermo Fisher) for 20 min at 4 °C. For surface staining, cells were washed and then stained with dumping markers: BV510 anti-CD56 (BioLegend, clone: 5.1H11, 362534, titer: 1:33) and BV510 anti-CD11b (BioLegend, clone: ICRF44, 301334; titer: 1:50); T cells markers: BUV805 anti-CD8 (BD Biosciences, clone: SK1, 612889, titer: 1:50) and BV650 or APC/Cy7 anti-CD3 (BD Biosciences, clone: UCHT1, 563851, titer: 1:33); integrins: BUV395 anti-CD103 (BD Biosciences, clone: Ber-Act8, 564346, titer: 1:33), BV421 anti-CD61 (BioLegend, clone VI-PL2, 744381, titer: 1:25), AF647 anti-CD61 (BD Biosciences, clone VI-PL2, 336408, titer: 1:33), PerCP/Cy5.5 anti-CD41 (BioLegend, clone: HIP8, 303704, titer: 1:50), FITC anti-CD41 (BioLegend, clone: HIP8, 303719, titer: 1:33), APC anti-integrin β7 (BioLegend, clone: FIB504, 321208, titer: 1:50) and PE (BioLegend, clone: NKI-M9, 327910, titer: 1:33; FITC anti-CD51 BioLegend, clone: NKI-M9, 327908, titer: 1:33); tissue-resident T cell markers: PerCP/Cy5.5 anti-CD45RO (BioLegend, clone: UCHL1, 304222, titer: 1:33), PE (BD Biosciences, clone: T2/S7, 568716, titer: 1:33), BUV496 anti-CD49a (BD Biosciences, clone: T2/S7, 755215, titer: 1:50), PE/Cy7 or BV605 anti-CD69 (BioLegend, clone: FN50, 310938, titer: 1:50), BUV486 anti-CD62L (BD Biosciences, clone: DREG-56, 741155, titer: 1:50), PE/Cy7 anti-CCR7 (BioLegend, clone: G043H7, 353226, titer: 1:50); tumor-reactive TIL markers: APC/Cy7 anti-CD39 (BD Biosciences, clone: A1, 328226, titer: 1:25), BV785 anti-CD39 (BioLegend, clone: A1, 328240, titer: 1:25); T cell differentiation markers: PE/Cy7 anti-CD27 (BioLegend, clone: M-T271, 256411, titer: 1:33) and BUV496 anti-CD28 (BD Biosciences, clone: 28.2, 741168, titer: 1:33); inhibitory markers: BUV737 anti-PD-1 (BD Biosciences, clone: EH12.1, 612791, titer: 1:33), BV421 anti-Tim-3 (BD Biosciences, clone: 7D3, 565568, titer: 1:33), BB515 anti-Tim-3 (BD Biosciences, clone: 7D3, 565562, titer: 1:33) and PE anti-TIGIT (BD Biosciences, clone: TgMAB-2, 568672, titer: 1:33) for another 20 min at 4 °C. For intracellular cytokine staining, cells were then washed and permeabilized with BD CytoFix/CytoPerm Solution for 20 min at 4 °C, before staining with the following cytokines: APC/Cy7 anti-IFN-γ (BioLegend, clone: B27, 506524, titer: 1:33) and BV785 anti-TNF (BioLegend, clone: Mab11, 502948, titer: 1:33); cytolytic molecules: PE anti-granulysin (BioLegend, clone: Dh2, 348004, titer:

1:33), AF488 anti-granzyme M (Thermo Fisher, clone: 4B2G4, 53-9774-42, titer: 1:25), AF647 anti-granzyme B (BD Biosciences, clone: GB11, 560212, titer: 1:50); chemokines: BUV737 anti-CCL5 (BioLegend, clone: 2D7/CCR5, 612808, titer: 1:33) and FITC anti-XCL2 (Novus Biological, clone: 06, NBP3-06177F, titer: 1:25) for another 20 min at 4 °C. Following antibody staining, cells were fixed with 1× CellFix (BD Biosciences) and acquired on a BD LSR Symphony (BD Biosciences) using BD FACSDiva v9.0 software and and analyzed using FlowJo v.10.5.3 with Phenograph and $t$-SNE plugins installed (BD Biosciences).

## Ex vivo T cell proliferation assay

Tissue cell suspensions were stained with 0.5 μg ml$^{-1}$ CFSE before activation with 10 μl αCD3/CD28 (StemCell Technologies). The cells were incubated at 37 °C for 72 h. After, the cells were stained with Live/Dead Fixable Aqua Cell Stain Kit (Thermo Fisher) for 20 min at 4 °C before being stained with BV650 anti-CD3 (BD Biosciences), BUV805 anti-CD8 (BD Biosciences), BUV395 anti-CD103 (BD Biosciences), BV421 anti-CD61 (BD Biosciences), PerCP/Cy5.5 anti-CD45RO (BioLegend), PE anti-CD49a (BD Biosciences and BioLegend) and PE/Cy7 anti-CD69 (BioLegend; details mentioned above). Following antibody staining, cells were fixed with 1× CellFix (BD Biosciences) and acquired on a BD LSR Symphony (BD Biosciences) using BD FACSDiva v9.0 software and analyzed on FlowJo v.10.5.3 (TreeStar). Cells were considered proliferative based on the decrease in CFSE fluorescence, within the 1st downward peaks of CFSE onwards.

## Immunohistochemistry analysis

Lung tumor resections were cut at 5-μm thickness and adjacent slides were separately stained against CD103 (Leica, clone EP206, PA0374, titer: 1:1,000), CD8 (Leica, clone 4B11, PA0183, titer: 1:1,500), CD61 (Abcam, clone VI-PL2, AB_1086-711, titer: 1:250) and E-cadherin (Leica, clone: 36B5, PA0387, titer: 1:500). Slides were digitized using Phillips IntelliSite Pathology Solution and analyzed using the Visiopharm Integrator System (VIS) platform version 2020.09.0.8195. Analysis protocols were implemented as Analysis Protocol Packages (APPs) in VIS. The Tissuealign module was used to align five digitized serial sections. The alignment was performed both on a large scale and on a finer-detailed level, to get the best possible match of the five tissue slides. First auxiliary APPs run with threshold classification that identifies the tissue regions. Secondary auxiliary APPs run on the CD61 slide using DeepLabv3 network of the VIS AI module that identifies the CD61[+] regions. The regions of interest (ROIs) are then superimposed on the aligned CD8 slide to outline regions for analysis. A CD8 APP was run on the ROIs outlined by the CD61 app to find co-located cells between CD61 and CD8. A similar approach was used to co-locate cells between CD61, CD8 and CD103. A HDAB-DAB color deconvolution band was used to detect positively stained cells. Several pre-processing steps were included to enhance positive signal while suppressing the background variation. The thresholding classification method defines a threshold for a given feature and assigns one class to all pixels with a feature value above or equal to that value, and another class for the rest. The classification rule is as previously described[20]. A method for cell separation that is based on shape and size was used, cell areas that are too small were removed and, finally, unbiased counting frames to avoid cells intersecting with neighboring tiles were counted twice (or more). To determine tumor bodies, we used E-cadherin overexpression as a tumor marker because E-cadherin is known to be overexpressed on epithelial tumor cells. On a serial section, we identified E-cadherin-positive staining area using the threshold method on APP to separate tumor cells and normal epithelia. Cells present at 'within' areas were defined as cells located within the E-cadherin[overexpressed] stained regions; cells present at 'clustering' areas represent cells located within 1.5 cm ($\chi < 1.5$ cm) from the E-cadherin[overexpressed] regions; and cells present at 'distal' areas represent cells located over 1.5 cm ($\chi > 1.5$ cm) from the E-cadherin[overexpressed] regions. The width denoting 'clustering' is as shown on Extended Data

Fig. 7, in which the APP algorithm is used to set the 'clustering' as the half average of the distance between one E-cadherin[overexpressed] stained region with another.

## Preparation of glass-supported lipid bilayers

Acid-cleaned and plasma-cleaned (5 min) Coverslip Glass D 0.17 ± 0.005 mm (Schott Nexterion, 1472315) was attached to Sticky-Slide VI[0.4] chambers (Ibidi, 80608) to assemble six-well imaging chambers. To form glass-supported lipid bilayers, a liposome master mix containing 0.1875% vol:vol of 0.4 mM CapBio, 12.5% vol:vol 0.4 mM DGS-NTA(Ni) in a 0.4 mM DOPC matrix (to 100% vol:vol) was incubated on the glass for 30 min at RT to allow spreading. To remove excess liposomes, assembled SLBs were washed three times with HBS/HSA buffer and then blocked using a 5% BSA solution containing 5 µg ml$^{-1}$ of Streptavidin either unconjugated or conjugated with Dylight 405 (Thermo Fisher Scientific, 21831) and 100 µM NiSO$_4$ for 20 min at RT. After three washes, SLBs were reconstituted to form an antigen-presenting cell membrane by incubation with 200 molecules per µm$^2$ of ICAM-1, 100 molecules per µm$^2$ of CD58, 100 molecules per µm$^2$ of E-cadherin (SinoBiological) and 30 molecules per µm$^2$ of biotinylated antigenic HLA-A2 NY-ESO-1 peptide complex (HLA-A*02:01 loaded with NY-ESO-1$_{157-165}$-specific SC9 peptide (SLLMQITQC)). Designs and calibration of recombinant protein densities were performed on bead-supported lipid bilayers as described elsewhere[60,61]. E-cadherin was calibrated with flow cytometry on cells. After a 30-min incubation at RT, SLBs were washed three times and incubated with 0.75 M T cells per well for 5, 10 or 15 min at 37 °C. After stimulation, cells were immediately fixed for 10 min with prewarmed 4% paraformaldehyde in PBS containing 2 mM MgCl$_2$, washed three times with HBS/HSA buffer, and stained using BV421 anti-Integrin β7 (BD Biosciences, clone: FIB504, 564283, titer:1:50), AF488 anti-CD103 (Abcam, clone: EPR4166(2), 129202, titer:1:500) and AF647 anti-CD61 (BioLegend, clone: VI-PL2, 336408, titer:1:33), and washed three times before performing microscopy as below.

## TIRFM

Imaging of immune synapses was performed on an Olympus IX83 inverted microscope equipped with a four-line (405-nm, 488-nm, 561-nm and 640-nm laser) illumination system, fitted with an Olympus UApON 150 × 1.45 numerical aperture objective, and a Photometrics Evolve delta EMCCD camera to provide Nyquist sampling. Quantification of fluorescence intensity was performed with Fiji/ImageJ (LifeLine Java version 8; National Institutes of Health), as previously described[62]. For colocalization analyses, we used the EzColocalization plugin with a combination of manual and Costes' method-assisted thresholding to identify relevant pixel values, as previously described[63]. For analyzing the sectional distribution of each integrin within synapses, we used custom-written Fiji/ImageJ macros to segment cells based on either the Ag channel or the integrin β7 integrin channel. We then performed radial averaging on all the channels from the segmented micrographs by rotating them 1° for 359 times before averaging all the rotated copies from each channel. We then averaged all radial averages from each channel before we drew a diagonal line plot on the resulting micrographs to analyze the radial MFI of signal from each channel. When segmenting cells on the Ag, the radial averages were centered on the cSMAC. When segmenting on the integrin β7 channel, the radial averages were centered on the whole contact area, as this signal was mainly found in the periphery. To ensure flatness in the topography of the contact zone between membrane and the bilayer, we used the bilayer system established in our previous studies[61,64–66], and T cell membrane flatness was ensured using ICAM-1–LFA-1 interactions on the substrate, restricting formation of membrane protrusions.

## Co-IP of CD61, CD103 and integrin β7

Integrin-transduced primary CD8$^+$ T cells were washed twice with cold PBS and lysed in Pierce IP Lysis Buffer (25 mM Tris, pH 7.4, 150 mM NaCl, 1% Nonidet P-40, 1 mM EDTA, 5% glycerol containing PhosSTOP phosphatase inhibitors and cOmplete protease inhibitor cocktail). Whole-cell lysates were incubated on ice for 10 min and centrifuged at 13,000$g$ at 4 °C for 5 min to remove dead cells or cell debris. Pre-cleared lysates were incubated with Anti-FLAG or anti-cMyc beads (Sigma-Aldrich), at 4 °C overnight. The proteins bound were pulled down by magnetic beads according to the manufacturer's protocol. Briefly, the samples were incubated with beads overnight, followed by three washes with PBS. Pulldown samples were eluted by competition using the 3X FLAG peptide. The elution was carried out at RT via the incubation of beads in 0.1 mg ml$^{-1}$ 3X FLAG peptide for 30 min. Beads were then removed, and samples were subjected to immunoblotting analysis.

## Immunoblotting

Samples were loaded onto the 4–15% gradient Criterion TGX precast gels (Bio-Rad). Proteins were transferred onto the nitrocellulose membrane, and the membrane was then blocked in 5% skim milk in 0.1% TBS-Tween 20 (TBST) for 1–2 h. After blocking, the membrane was incubated overnight at 4 °C with primary antibodies (purified anti-β-actin antibody (Sigma-Aldrich, clone: AC-74,A5316, titer: 1:1,000), purified anti-CD103 antibody (clone: EPR4166(2), Abcam, 129202, titer:1:500), purified anti-cMyc antibody (Sigma-Aldrich, clone 9E10, M4439, titer: 1:500) or purified anti-FLAG antibody (Sigma-Aldrich), clone: M2, F3165, titer:1:1,000)). The membranes were then incubated with relevant IRDye secondary antibodies (Li-COR) in TBST containing 5% skim milk after washing 3–4 times in TBST. Gels were imaged on the Li-COR Odyssey DLx, using Li-COR Acquisition software v2.0.

## In vitro T cell functional assays

For cytotoxicity assays, cancer cells were stained with 0.5 µg ml$^{-1}$ CFSE (Thermo Fisher) before co-culture with T cells at an E:T ratio of 1:2 at 37 °C for 2–8 h. Cells were then stained with BV421 anti-E-cadherin (BD Biosciences, clone 67A4, 743712, titer: 1:33) and PE/Cy7 anti-CD8 (BD Biosciences, clone SK1, 344712, titer: 1:25) and 7-AAD dye (BD Biosciences). For CD107a staining, T cells were co-cultured with cancer cells at the same E:T ratio at 37 °C for 4 h, in the presence of PE/Cy7 anti-CD107a (BioLegend, clone H4A3, 328618, titer: 1:20) staining. For phosflow staining, T cells were co-cultured with cancer cells using the same E:T ratio at 37 °C for 15 min, 30 min or 2 h. Cells were then stained with 0.1 µl Live/Dead Fixable Aqua Dead Cell Stain Kit (Thermo Fisher) before being fixed with BD Fixation Buffer for 10 min at 37 °C. Cells were permeabilized using BD Phosflow Perm Buffer III for 30 min at 4 °C before stained with AF647 anti-ZAP70 (pY292) (BD Biosciences, clone: J34-602, 558515, titer:1:33), AF647 anti-PLCγ1 (pY783) (BD Biosciences, clone: 27/PLC, 557883, titer:1:25) and AF647 anti-Lck (pY505) (BD Biosciences, clone:4/LCK-Y505, 558552, titer:1:25). In certain cases, cells were treated with 10 nM genistein (Santa Cruz Biotechnology) or 7.5 nM A770041 (Lck inhibitor, Sigma-Aldrich) before T cell activation. In each assay, the CD61$^+$ T cell lines from seven patients with cancer were treated with anti-CD61 (10 µg ml$^{-1}$; clone: PM6/13, Novus Biotechnology, titer 1:10), in parallel to T cell activation. Cells in these assays were acquired immediately on the Attune Nxt flow cytometer v3.2.1 (Thermo Fisher) and analyzed with FlowJo v.10.5.3 (BD Biosciences).

## Mice xenograft and tumor growth kinetics assay

Immunodeficient NSG mice (strain NOD.Cq-Prkdc scid Il2rg$^{tm1Wjl/SzJ}$) were xenografted with NY-ESO-1$^+$ HCT116. A WT HCT116 is absent of NY-ESO-1 antigen as previously described[15] and, therefore, was transduced with lentivirus expressing the NY-ESO-1 protein with eGFP via an internal ribosomal entry site (IRES) link, as previously described[53]. Transduced cells were sorted based on eGFP-positive expression and cultured for three passages before confirmation with flow cytometry staining. In total, 1 M NY-ESO-1$^+$ HCT116 cells in PBS were injected subcutaneously in a 1:1 ratio with Matrigel Matrix solution (Corning). After

48 h, mice were randomized into groups ($n$ = 8–10), and 1 M T cells (WT CD61$^+$ or CD61$^-$ T cell clones (of patient 1)) were injected intravenously. No statistical methods were used to predetermine samples sizes, but our sample sizes are similar to those reported in previous publications[53,67]. T cell injection was randomized within each cage with either T cell clone (T cell clone initially pseudonymized by another individual) before injection by another individual. Additional intravenous injections with the same number of T cells were carried out on days 7 and 14 after tumor xenografts. Digital caliper measurements were taken on days 4, 7, 10, 13, 16 and 20 after tumor xenografts. Tumor volume was approximated according to the formula for ellipsoid volume (width/2 × depth/2 × length/2 × π4/3). Data collection and analysis were performed blinded, T cell preparation and labeling were done by a different person than the person performing adoptive transfer to mice, and the tumor measurements and decoding of the treatment group were performed by separate individuals. For the in vivo assays, following subcutaneous injection of tumor into the mice, at 48 h after tumor xenograft, mice were randomly allocated into two groups, with one group injected intravenously with the different T cell clones. Age-matched male and female mice were used, using random allocation for both groups. According to PPL license PBA43A2E4 approved by the UK Home Office, the maximal tumor burden is 1.2 mm$^3$, and we confirm that within the duration of the experiment, the maximal tumor burden was not exceeded. No animals or data points were excluded from analysis. Data distribution was assumed to be normal, but this was not formally tested.

### Kaplan–Meier survival curve analysis

RNA expression datasets were downloaded from the TCGA database (https://portal.gdc.cancer.gov/) using RTCGA (version 1.18.0)[68]. Patient clinical metadata for associated datasets were downloaded from cBioPortal (https://www.cbioportal.org/)[69,70]. The datasets used were the SCM (TCGA, PanCancer Atlas), lung cancer (TCGA, PanCancer Atlas) and lung cancer (University of Cologne)[12,22–24,71,72]. Our analysis uses a publicly available coding analytical pipeline that facilitates the identification of *CD8A*-enriched and *CD3E*-enriched samples (enriched on T cells), which then were subjected to more granular analyses; although as TCGA dataset is not based on single-cell resolution, the analyses performed may not necessarily be representative of T cells. Using the surv_cutpoint function from the survminer R package, we objectively determined the optimal cutoff point for CD8 expression using the following arguments: time = 'Months.of.disease.specific. survival', event = 'Disease.specific.Survival.status'. We use the categorical variable of surv_categorize function, and samples with high *CD8A* and *CD3E* expression were classed as CD3$^+$CD8$^+$ samples. So, the CD3$^+$CD8$^+$ samples analyzed would be from datapoints that show evidence of a high T cell proportion. We then filtered out CD1$^{67}$03$^+$CD8$^+$ cells using the *ITGAE* (CD103) gene marker. Lastly, patients were segregated based on high or low expression of *ITGB3* (CD61), specifically to identify two groups of patients, having either CD61$^{hi}$CD103$^+$CD8$^+$CD3$^+$ samples or CD61$^{lo}$CD103$^+$CD8$^+$CD3$^+$ samples. Optimal cutoff points were calculated to distinguish between high or low expression of each of these four genes for each dataset (survival version 3.1-12)[73]. Only patients that showed high expression of *CD8A, CD3E* and *ITGAE* were deemed as *CD8A$^+$CD3E$^+$ITGAE$^+$* and kept in the analysis. These patients were used to plot Kaplan–Meier survival curves between *ITGB3$^+$* and *ITGB3$^-$* patients (survminer version 0.4.9, publicly available coding). For lung cancer datasets, the optimal cutoff points were calculated for stage I patients from each dataset separately, before being combined into a single survival plot. *P* values were determined by log-rank test.

### Statistical analysis

Unless stated otherwise, all graph generation and statistical analyses were conducted using GraphPad Prism software, and data are summarized as the median ± s.e.m. The number of patients and biological and technical repeats is as shown in the figure legends. Statistically significant differences between two groups were assessed using two-tailed paired *t*-test, with Wilcoxon adjustments for non-parametrically distributed variables. One-way ANOVA with Tukey's multiple-comparison test or two-way ANOVA with Tukey's multiple-comparison test was performed to compare two or more groups. Correlation analyses were performed using non-parametric Spearman rank correlation. Statistical significance was set as *$P$ < 0.05, **$P$ < 0.01 and ***$P$ < 0.001.

### Reporting summary

Further information on research design is available in the Nature Portfolio Reporting Summary linked to this article.

### Data availability

Raw proteomics data have been deposited on Mendeley Data (https:// doi.org/10.17632/b2xdk4h5xm.1)[74] and at the Proteome Xchange through PRoteomics IDEntifications Database (PRIDE) accession numbers PDX031794 and PDX045989. RNA and clinical data used for survival analysis are publicly available from TCGA (https://portal.gdc.cancer.gov/), including the SCM (TCGA, PanCancer Atlas), lung cancer (TCGA, PanCancer Atlas) and lung cancer (University of Cologne)[12,22–24,71,72]. Patient clinical metadata for associated datasets were downloaded from cBioPortal (https://www.cbioportal.org/)[69,70]. Raw data and reagents from all main and supplementary figures, beyond the mandatory dataset deposited on the public repository, are available on request from T.D. Source data are provided with this paper.

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

## Acknowledgements

We thank patients who volunteered to participate in this study. We express our gratitude to the late S. Ling Felce (CAMS Oxford Institute, University of Oxford) for the bioinformatics contribution to this study. We thank G. Zavalani (ORB, Nuffield Department of Surgical Sciences, University of Oxford) for assisting in patient acquisition and sampling during the early stage of this study, and A. Hayes and D. Royston (John Radcliffe Hospital, NHS Oxford University Hospitals) for assisting in the initial tumor sampling, preparation and immunohistochemistry staining. We thank X. Yang (CAMS Oxford Institute, University of Oxford) for the assistance with the functional assay work on the seven paired T cell lines. We thank J. Forbester (Cardiff University) for advice on knock-down system. We thank V. Junghans (CAMS Oxford Institute, University of Oxford) for providing initial guidance to R.C. for the plasmids making and preparation. Figure 3h was created with BioRender.com. We acknowledge the contribution to this study made by the Oxford Centre for Histopathology Research and the ORB, which are supported by the University of Oxford, the Oxford CRUK Cancer Centre and the NIHR Oxford Biomedical Research Centre (Molecular Diagnostics Theme/Multimodal Pathology Subtheme), and the NIHR CRN Thames Valley Network. The views expressed are those of the author(s) and not necessarily those of the NHS, the NIHR or the UK Department of Health. This work is supported by the Chinese Academy of Medical Sciences (CAMS) Innovation Fund for Medical Sciences (CFMS), China (grant no. 2018-I2M-002 to T.D., M.H.B.A.H., P.F.C., C.J., Z.L., E.A., F.G., R.T., DM-P., J-L.C., R.C., A.P.F., I.V., S.D., G.L., X.Y., N.M., N.K., D.C.C.D-F., Y-L.C., N.M.R., B.M.K., C.V., G.O., R.A.F., R.F., Y.P. and M.L.D.), UK MRC (T.D., Y.P., G.O. and J.-L.C.), CAMS Oxford Institute (COI) Career Development Fellowship (P.F.C., A.P.F., N.I.K. and Y.-L.C.). A.P.F. and S.D. are supported by Pfizer funding. Further support was provided from the NIHR Biomedical Research Centre (N.K.-A, C.V. and G.O.), Senior Investigator Award (G.O.) and Clinical Research Network (G.O.). M.L.D. was additionally supported by an ERC Advanced Grant (SYNECT AdG 670930) and the Kennedy Trust for Rheumatology Research. P.F.C. was supported by EMBO Long-Term Fellowship ALTF 1420-2015, in conjunction with the European Commission (LTFCOFUND2013 GA-2013-609409) Marie Sklodowska-Curie Actions and a Oxford-Bristol Myers Squibb Fellowship.

## Author contributions

T.D. conceptualized the study; T.D., Y.P. and M.H.B.A.H. designed the experiments. Y.P. and X.Y. generated the SSX-2-specific T cells, J.-L.C. generated NY-ESO-1, tyrosinase, melan-A cancer-specific T cell clones or lines. M.H.B.A.H. performed the majority of the T cell experiments assisted by X.Y., C.J. and N.M. F.G. and M.H.B.A.H. generated NY-ESO-1 TCR-T cells. P.F.C. performed TIRFM synapse experiments. Z.L. and C.J. performed Co-IP assays (C.N. provided basic support for the experiment). A.P.F., I.V. and S.D. performed mass spectrometry analysis of Co-IP lysates. R.F. and S.S.H. performed experiments with LC–MS/MS. U.G. did in vivo mouse work, with support from D.D.F. and Y.-L.C. R.T. and D.M.-P. collected clinical samples and data. L.C. performed immunohistochemistry staining. N.K.-A. analyzed IHC data. P.F.C., N.M. and G.L. generated CRISPR KO T cells. N.M., D.D.F., Y.-L.C., C.W., S.A.-C., K.C., P.S., J-.L.C., N.K., C.V., N.M.R., R.A.F. and R.C. provided technical assistance and critical reagents. M.H.B.A.H., P.F.C., E.A., N.K.-A. and A.K. analyzed data. T.D., M.L.D., Y.P., R.F. and M.H.B.A.H. supervised the data analysis. M.H.B.A.H. and P.F.C. wrote the original draft. T.D., M.L.D., Y.P., R.F., B.M.K., G.O. and R.A.F. reviewed and edited the manuscript and figures.

## Competing interests

The authors declare no competing interests.

## Additional information

**Extended data** is available for this paper at https://doi.org/10.1038/s41590-024-01802-3.

**Correspondence and requests for materials** should be addressed to Tao Dong.

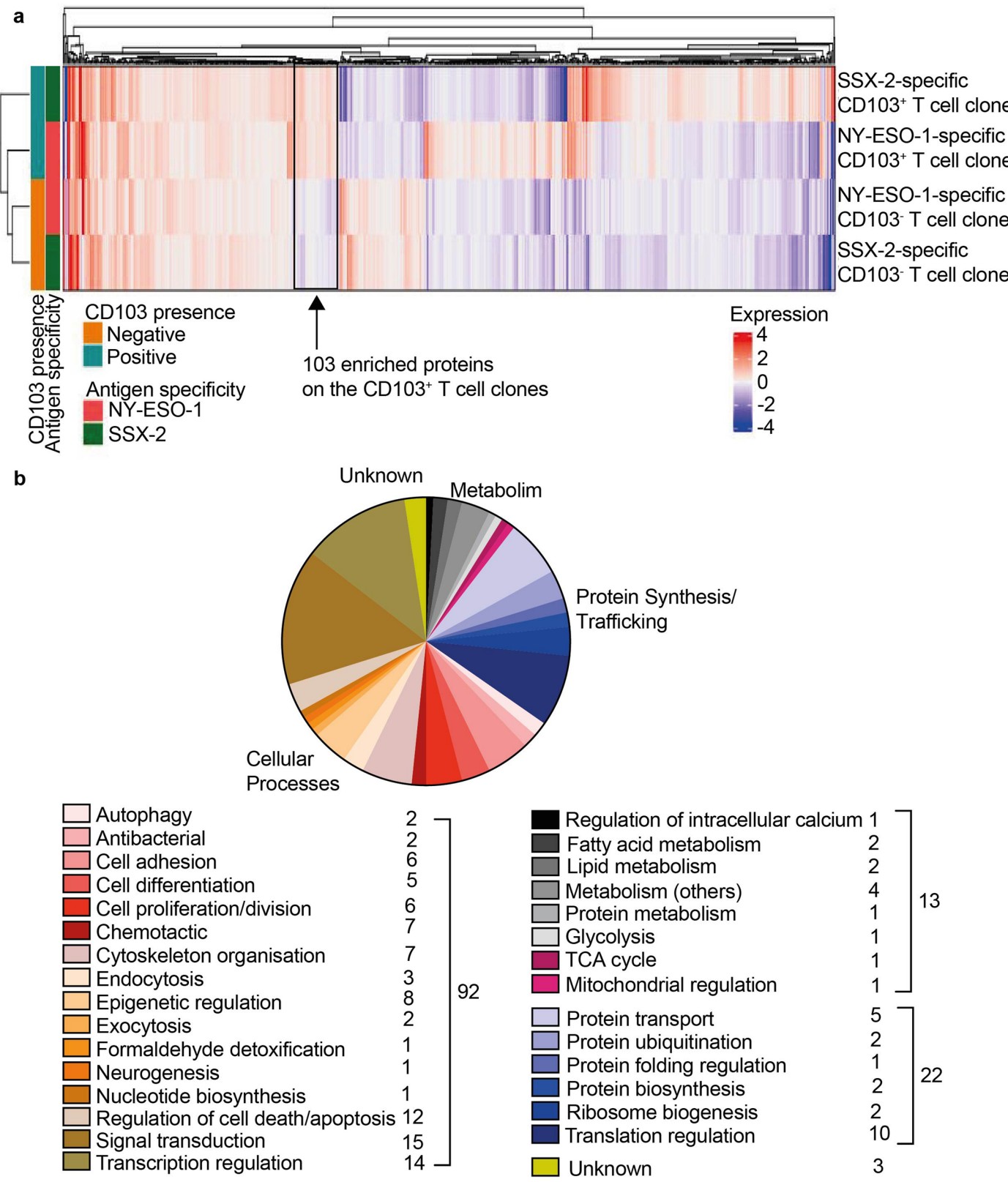

**Extended Data Fig. 1 | Enriched proteins of cancer-specific CD103⁺ CD8⁺ T cells. a**, Heatmap of 890 proteins by log$_2$ fold-change values, on CD103⁺ and CD103⁻ cancer-specific T cell clones from 2 different cancer patients. Arrow denotes the 103 proteins enriched on the CD103⁺ T cell clones. n = 2 patients' paired T cell clones. **b**, Pie chart on proteins classification by Gene Ontology

NCBI annotation, on four major groups of cellular processes, metabolism, protein synthesis and trafficking and unknown. Proteins were also subdivided into specific biological roles. Numbers on the side of each legend represent the number of proteins annotated to that subgroup. The number represents the number of proteins annotated to the major group.

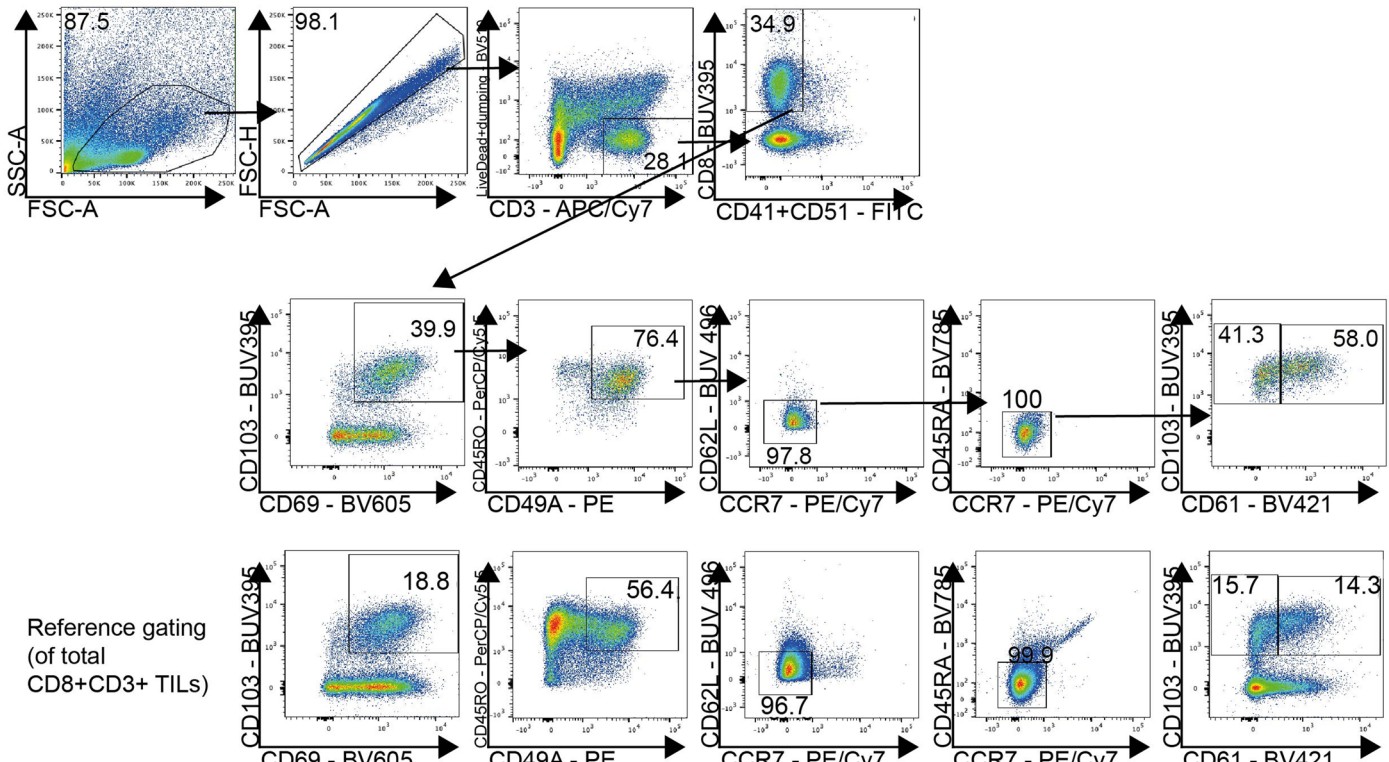

**Extended Data Fig. 2 | Gating strategy to identify CD61⁺ TILs ex vivo.** Gating strategy illustrating identification of CD61⁺CD103⁺ and CD61⁻CD103⁺ TILs based on initial gating of: lymphocytes (FSC-A vs SSC-A), single cells (FSC-H vs FSC-A), Live CD3⁺ cells (Live/Dead + dumping (CD56, CD11c) vs CD3), CD41 and CD51 negative CD8⁺ T cells (CD8 vs CD41 and CD51), T$_{rm}$ cells (CD103 vs CD69, CD45RO vs CD49a, CD62L vs CCR7, CD45RA vs CCR7). Gatings were performed based on the reference cell population of total CD8⁺CD3⁺ TILs. Subsequent analysis was then performed to analyse selected immunophenotype.

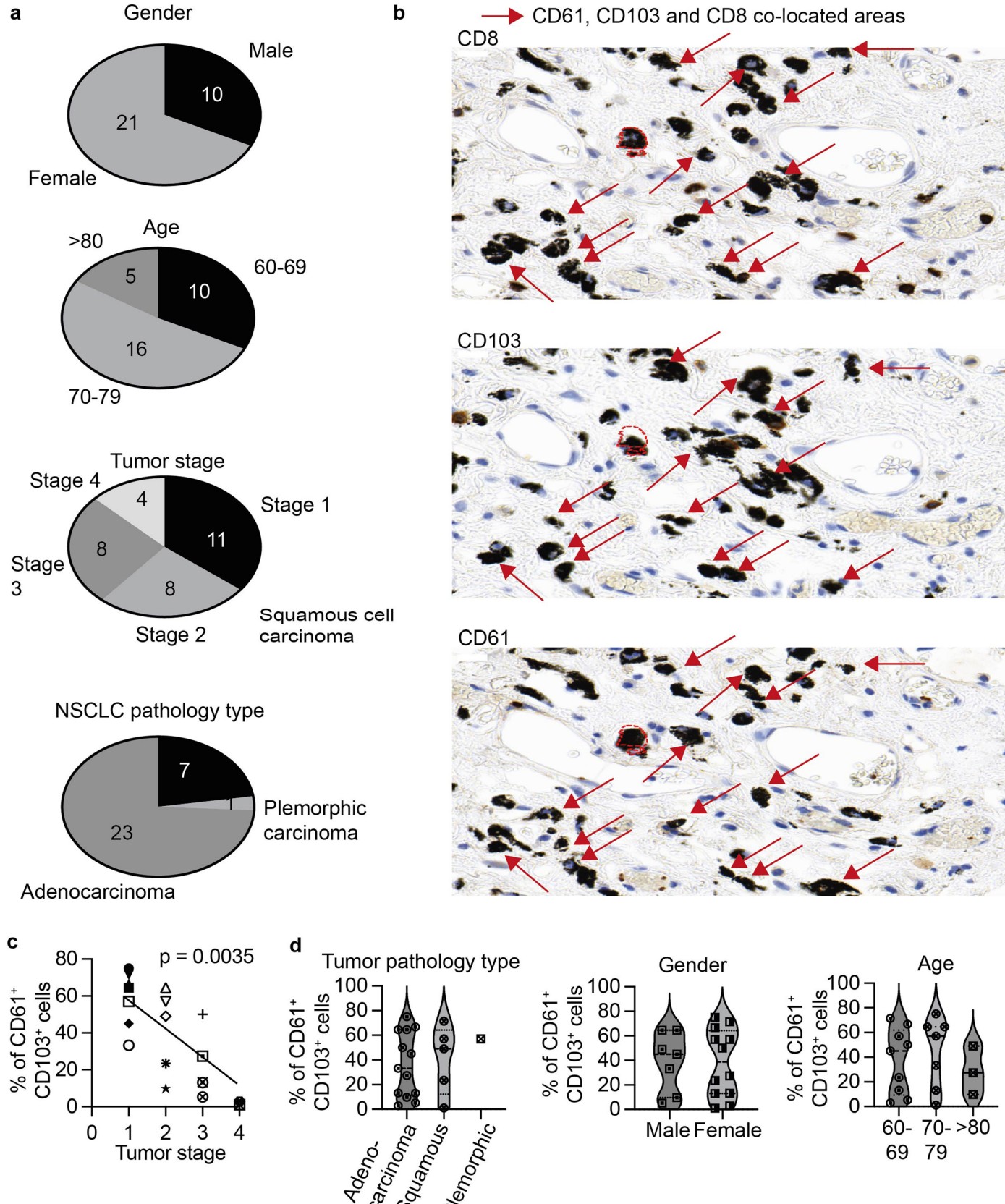

**Extended Data Fig. 3 | See next page for caption.**

**Extended Data Fig. 3 | Summary of clinical parameters and CD61 expression on TILs ex vivo. a**, Pie charts on the summary clinical parameters of NSCLC patients, according to gender, age, tumour stage and NSCLC pathology type, used in this study. **b**, Representative IHC images of CD8, CD103 and CD61 on three serial tumour resections, from one NSCLC patient. The red arrow represents areas of co-localisation between CD8, CD103 and CD61. Red dotted lines indicate the alignment of the serial tumour resections performed by Visiopharm. Scale white bar: 5μm. Representative image from Patient 7. Similar APP algorithm applied to all 19 patients IHC analysis, with consistent observations. **c**, Plot showing frequency of CD61$^+$ cells (out of the total CD103$^+$ TILs) and the tumour stage of each patient. Each variety of dots represents an individual cancer patient. p-value = 0.0035. Correlation analyses were performed using non-parametric Spearman rank correlation. **d**, Violin plots showing the frequency of CD61$^+$ cells (out of total CD103$^+$ TILs), according to tumour pathology type, gender and age. **a,c-d**. n = 19 patients.

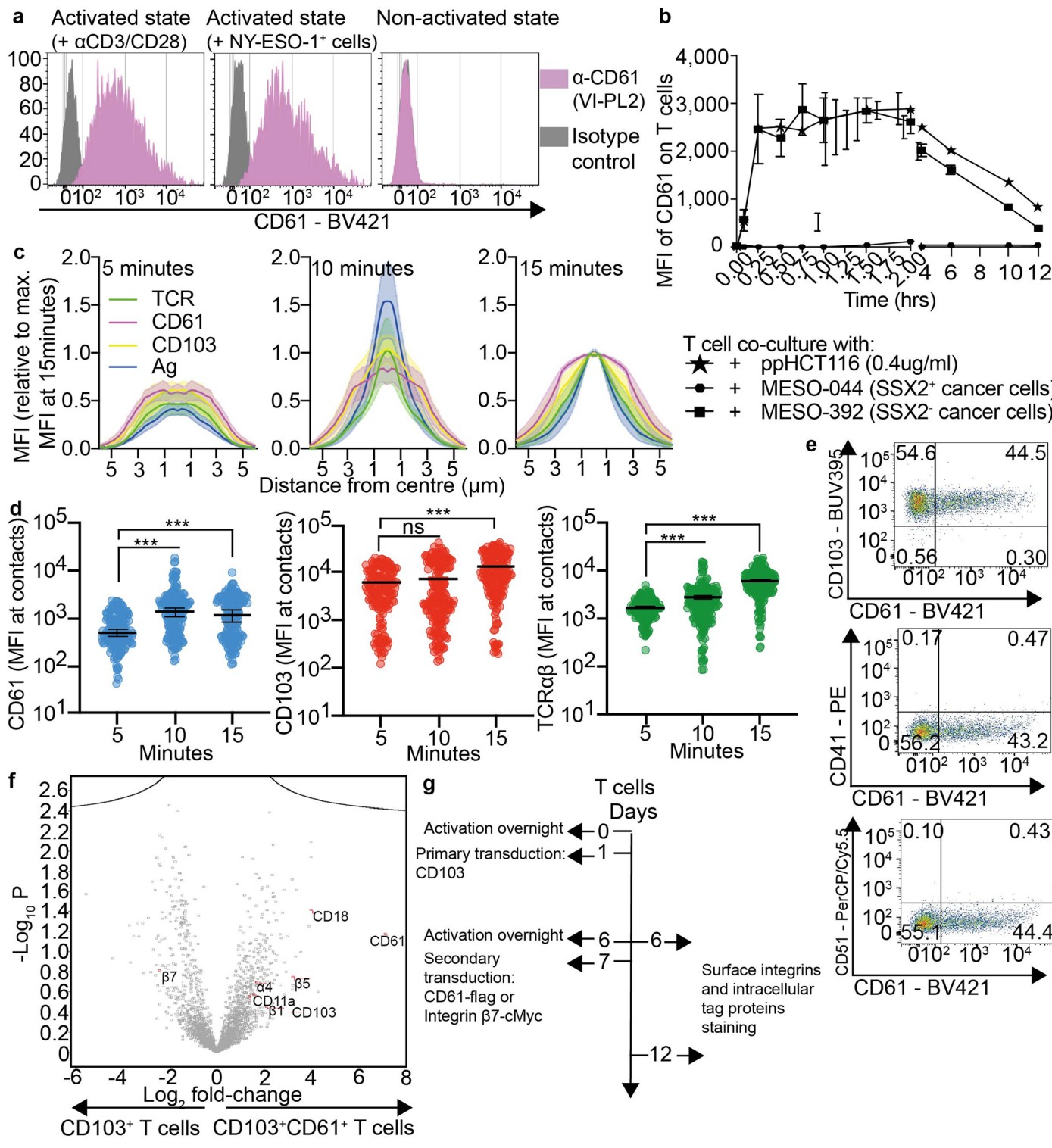

**Extended Data Fig. 4 | See next page for caption.**

**Extended Data Fig. 4 | CD61 co-localises with CD103. a**, Representative histogram showing CD61 expression on CD103[+] cancer-specific T cell clone (from patient 1), following activation either by αCD3/CD28 or by NY-ESO-1[+] cancer cells activation, or no activation. Grey represents isotype control, pink represents CD61 staining. **b**, Representative kinetic plot of CD61 expression by median fluorescence intensity (MFI) on cancer-specific CD103[+] T cell clone (from patient 1), following activation for either 0, 15 minutes, 30 minutes, 45 minutes, 1 hour, 1.25 hours, 1.5 hours, 1.75 hours, 2 hours and 6-, 8-, 10- and 12-hours. n = 3 independent experiments. **c**, Histograms showing the radially averaged MFI of antigen (denoted by HLA-A2$_{NY-ESO-1}$, in blue), CD103 (yellow), CD61 (magenta) and TCR (green) plotted as relative to maximum MFI at 15 minutes, according to distance from the synapse centre at 5, 10, 15 minutes post synaptic formation. n = 3 independent experiments. **d**, Dot-plots showing MFI of CD61, CD103 and TCRαβ at the point of synapse contacts, at 5-, 10-, and 15-minutes post synaptic formation. Each dot represents one synaptic contact per cell. n = 150 cells

examined over 3 independent experiments. *** p < 0.001, ns = not significant, one-way ANOVA with Tukey's multiple-comparison test. **e**, Flow cytometry plots showing CD61 expression against CD41, CD51 or CD103 on CD103[+] cancer-specific T cell clone (from patient 5). **f**, Volcano plot showing enrichment of CD61 and CD103 but downregulation of integrin β7 on the primary CD103[+]CD61[+] T cell line lysate compared to primary CD103[+] T cell line lysate, n = 3 independent experiments. Statistical test used involve one-way ANOVA, with Tukey multiple comparison test, converted to -log$_{10}$ p values for each datapoint. Raw fold-change values was normalised using log$_2$. **g**, Schematic showing integrins cell surface rescue workflow. Primary T cells were activated overnight (Day 0) before primary transduction with CD103. Cells were stained for CD103-HA and CD61-flag or integrin β7-cMyc surface and intracellular expression on Day 6. Cells were re-activated overnight before secondary transduction with CD61-flag or integrin β7-cMyc. Staining was repeated on Day 12. n = 3 independent experiments. **b-d**. Error bar and highlight presented as median± SEM.

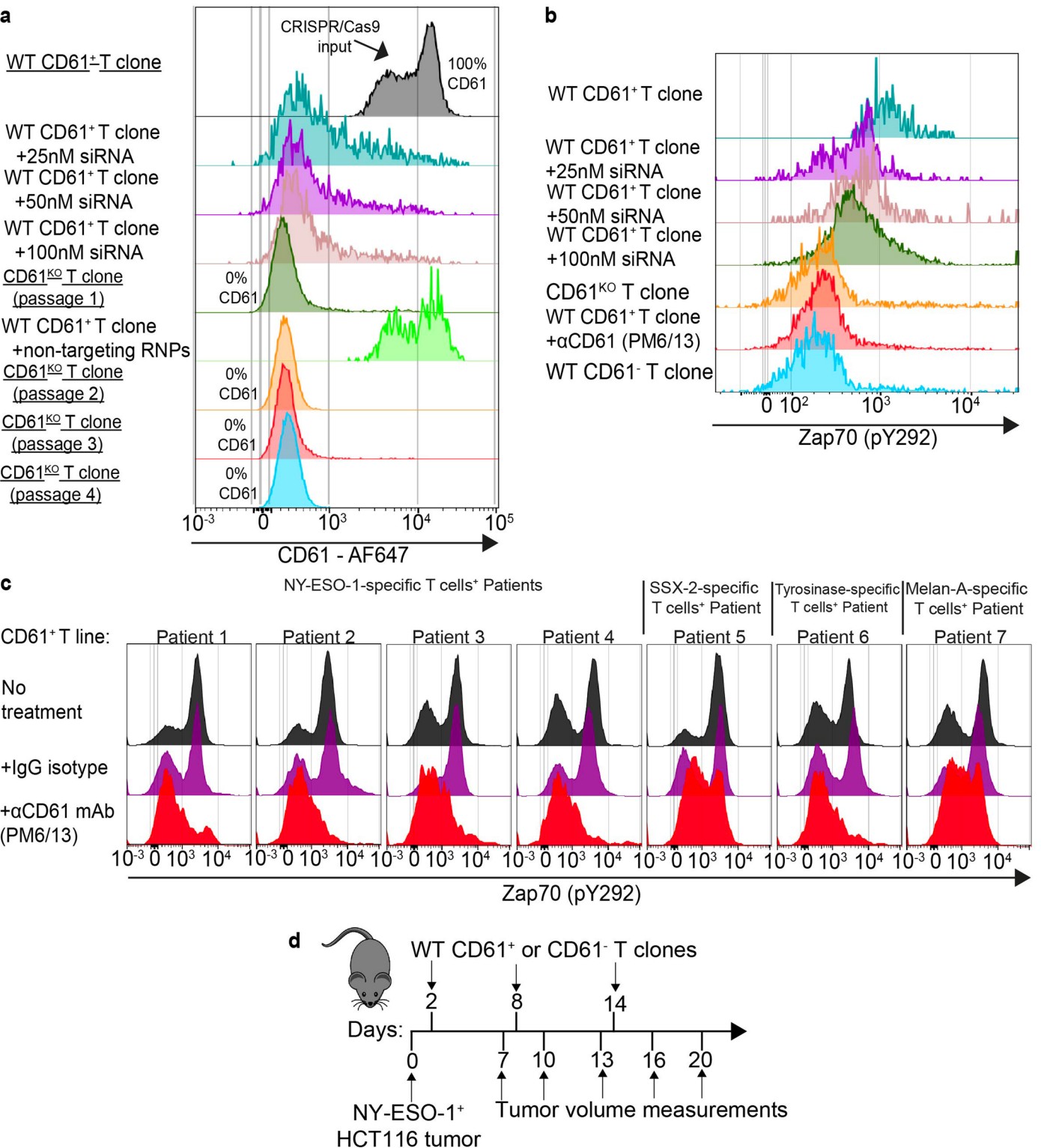

**Extended Data Fig. 5 | See next page for caption.**

**Extended Data Fig. 5 | Manipulation of CD61 affects T cell functions.**
**a**, Overlaid flow cytometry histograms showing downregulation of CD61 expression following serial CD61 siRNA treatment (+25 nM, or 50 nM, or 100 nM treatment) on WT CD61$^+$ T cell (of patient 1). The fifth histogram shows the abrogation of CD61 expression on CD61$^{KO}$ T cell (dark red) following CRISPR-Cas9 editing of the CD61 gene on WT CD61$^+$ T cell clone (grey). Note the CD61$^{KO}$ T cell derived from a starting population of 100% CD61 positive cells. The sixth histogram shows WT CD61$^+$ T cell transfected with non-targeting RNPs as control, showing no changes in the CD61 expression. CRISPR-Cas9-mediated CD61$^{KO}$ T cell demonstrated consistent CD61 abrogation across 4 passages of T cell expansion. **b**, Overlaid flow cytometry histograms showing phosphorylation level of Zap70 (pY292) on WT CD61$^+$ T cell clone, WT CD61$^+$ T clone treated with 25 nM, 50 nM or 100 nM siRNA, CD61$^{KO}$ T cell clone, WT CD61$^+$ T cell clone treated with anti-CD61 blocking antibody (PM6/13) and WT CD61$^-$ T cell clone (of patient 1). **c**, Histogram plots showing Zap70 (pY292) phosphorylation level on CD61$^+$ T cell lines, from 7 different cancer patients following either αCD61 neutralising antibody treatment, IgG isotype control treatment, or no treatment (four patients with NY-ESO-1-specific, and one patient each with SSX-2-specific, Tyrosinase-specific and Melan-A-specific). **d**, Schematic diagram of in vitro tumour growth assay, with NOD.SCID mice xenografted with NY-ESO-1$^+$ HCT116 tumour at day 0 before adoptive transferred with WT CD61$^+$ or CD61$^-$ T clones (derived from patient 1) at day 2, day 8 and day 14. Tumor volume measurements were taken at intervals, at Day 7, 10, 13, 16 and 20 post xenografts.

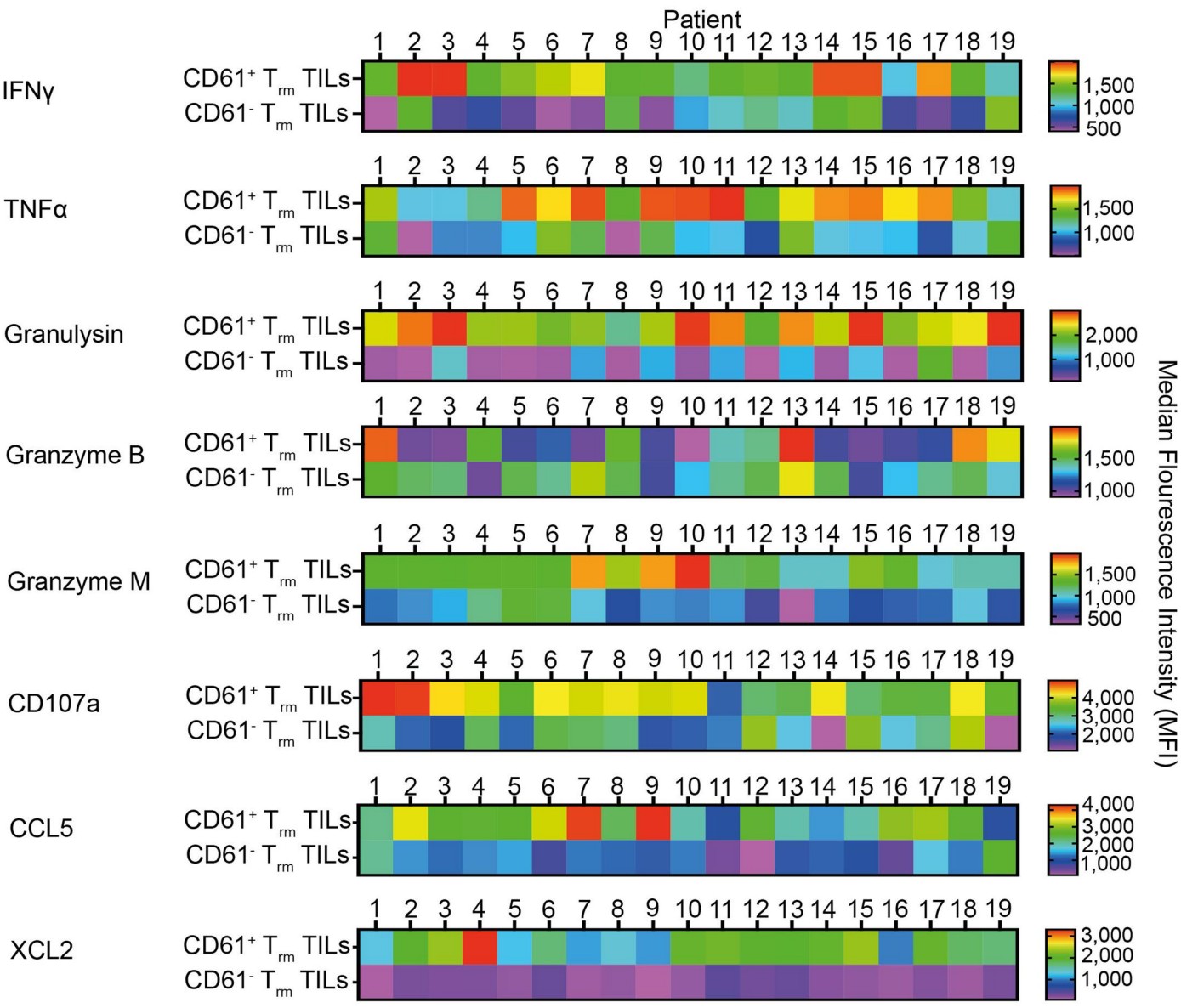

**Extended Data Fig. 6 | Effector immunophenotypes of NSCLC patients.** A set of heatmaps illustrating the MFI of IFNγ, TNFγ, granulysin, granzyme B, granzyme M, CD107a, CCL5 and XCL2, on CD61⁺CD103⁺ and CD61⁻CD103⁺ TILs, of each patient. n = 19 patients. Values represent MFI, with a colour gradient from red to blue. n = 19 patients.

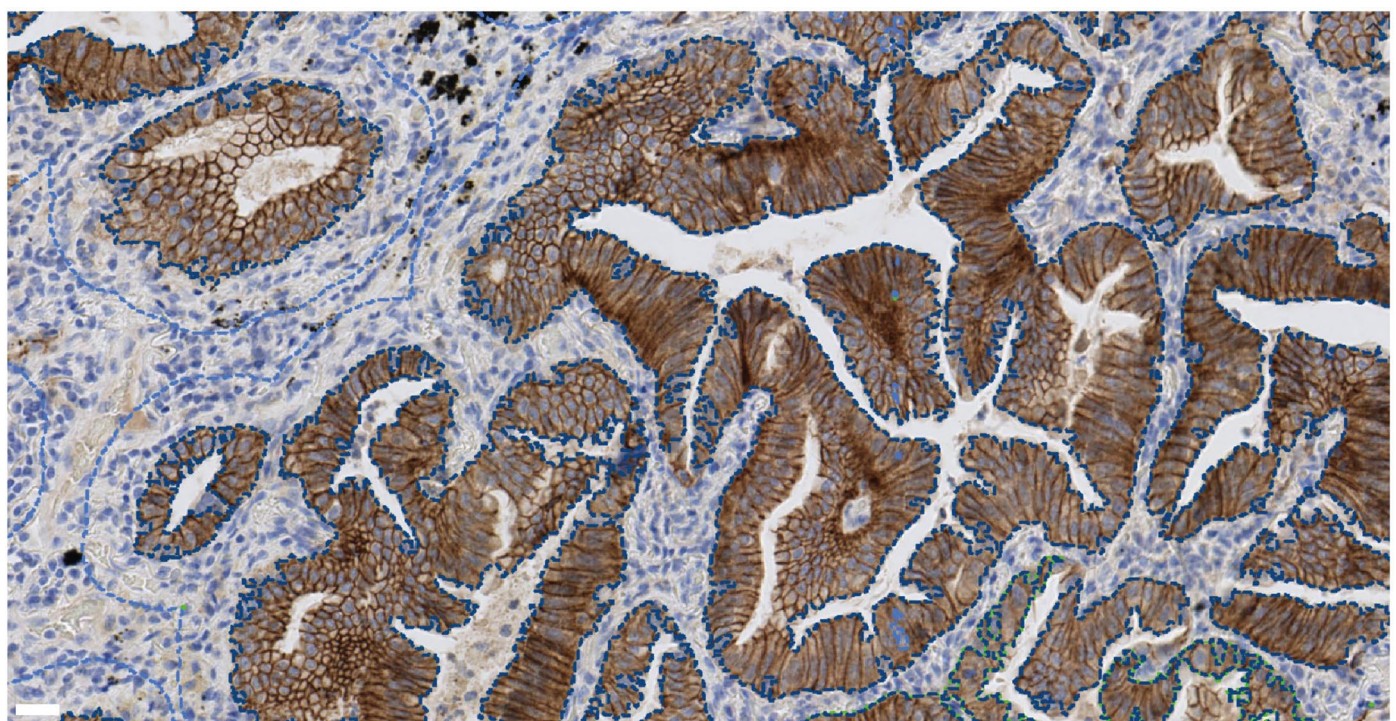

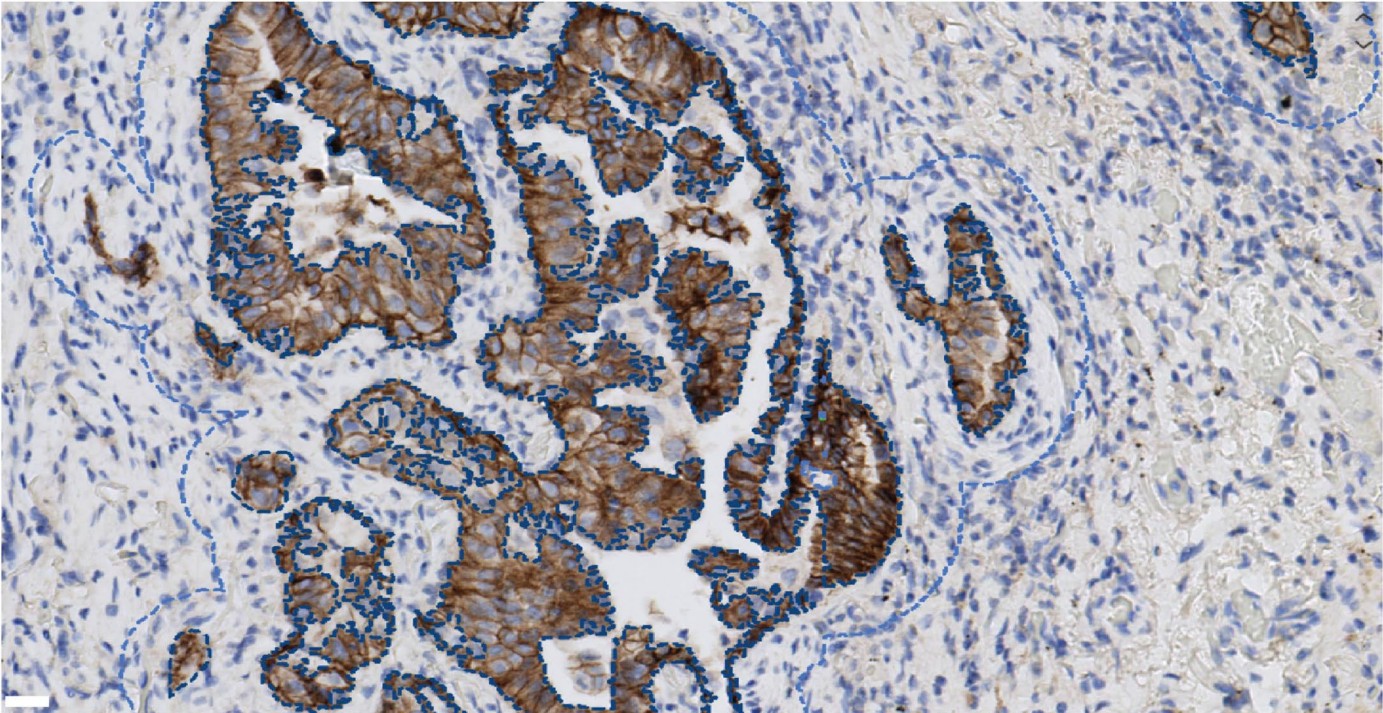

**Extended Data Fig. 7 | Width lining of 'clustering' areas of E-Cadherin**[over-expressed] **tumors.** Brown areas showing positive E-Cadherin[over-expressing] cells, with blue staining denoting nuclear staining. Blue dashed lining denoting area margin of 1.5 cm from the E-Cadherin[over-expressing] cells, determined using Visiopharm IHC APP algorithm (further detailed in Methods section). Scale white bar: 10 μm. Representative image from Patient 7. Similar APP algorithm applied to all 19 patients IHC analysis, with consistent observations.

# Reporting Summary

## Statistics

For all statistical analyses, confirm that the following items are present in the figure legend, table legend, main text, or Methods section.

| n/a | Confirmed | |
|---|---|---|
| ☐ | ☒ | The exact sample size (*n*) for each experimental group/condition, given as a discrete number and unit of measurement |
| ☐ | ☒ | A statement on whether measurements were taken from distinct samples or whether the same sample was measured repeatedly |
| ☐ | ☒ | The statistical test(s) used AND whether they are one- or two-sided<br>*Only common tests should be described solely by name; describe more complex techniques in the Methods section.* |
| ☐ | ☒ | A description of all covariates tested |
| ☐ | ☒ | A description of any assumptions or corrections, such as tests of normality and adjustment for multiple comparisons |
| ☐ | ☒ | A full description of the statistical parameters including central tendency (e.g. means) or other basic estimates (e.g. regression coefficient) AND variation (e.g. standard deviation) or associated estimates of uncertainty (e.g. confidence intervals) |
| ☐ | ☒ | For null hypothesis testing, the test statistic (e.g. *F*, *t*, *r*) with confidence intervals, effect sizes, degrees of freedom and *P* value noted<br>*Give P values as exact values whenever suitable.* |
| ☒ | ☐ | For Bayesian analysis, information on the choice of priors and Markov chain Monte Carlo settings |
| ☒ | ☐ | For hierarchical and complex designs, identification of the appropriate level for tests and full reporting of outcomes |
| ☐ | ☒ | Estimates of effect sizes (e.g. Cohen's *d*, Pearson's *r*), indicating how they were calculated |

*Our web collection on statistics for biologists contains articles on many of the points above.*

## Software and code

Policy information about availability of computer code

| Data collection | Flow cytometry data were collected by BD FACSDiva V9.0 on BD LSR Symphony, or Attune NxT software V3.2.1 on ThermoFisher Attune Next flow cytometer. Proteomics data were collected with either Orbitrap Fusion Lumos Tribrid platform (instrument control software v3.3) or Orbitrap Acend tribrid instrument. Miscroscopal images were collected with Olympus IX83 inverted microscope equipped with a 4-line (405 nm, 488 nm, 561 nm, and 640 nm laser) illumination system. Immunoblotting images were collected with Li-COR Odessy DLx, Li-COR Acqustion software v2.0. Immunohistochemistry images were collected using Phillips IntelliSite Pathology Solution and Visiopharm Integrator System (VIS) platform version 2020.09.0.8195. |
|---|---|

| Data analysis | Flow cytometry data were analysed with FlowJo V10.5.3 software for Mac OS, with Phenograph and tSNE plug-ins installed. Proteomics raw data were normalised to non-activated data, with STRING (https://string-db.org), NCBI Gene Ontology (https://geneontology.org) and Reactome online databases used to categorise proteins and pathways. Data was processed ion Perseus (v1.3.2.3. For ReactomePA analysis v1.32.0, Protein to gene names were converted to Entrez IDs (org.Hs.eg.db version 3.11.4) before using as input for overrepresentation analysis (clusterProfiler version 3.18.0, ReactomePA version 1.32.0) to find REACTOME pathways with enriched genes (with P value cut off 0.01 and P adjusted value cut off 0.05). The resulting output was used to create a gene concept network. Barplots were constructed for selected pathways using log2 fold change values for specific genes found in pathways (ggplot2 version 3.3.2).Volcano plots were generated using the processed data and plotted using VOlcaNOseR v1.0.3 Quantification of fluorescence intensity for microscopy experiments was performed with Fiji/ImageJ (LifeLine Java version 8) (National Institute of Health). Immunohistological analyses were done with Visiopharm v2020.09.0.8195, by first aligning adjacent slides and creating regions of overlapping positive staining to identify T cells. For survival curve Kaplan-Meier analysis, patient survival data was merged with RNA expression data for ITGB3, ITGAE, CD8A and CD3E. Optimal cutpoints were calculated to distinguish between high or low expression of each of these four genes for each dataset (survival version 3.1-12). A patient was deemed to have CD61+CD103+CD8+CD3+ cells if they showed high expression of all four genes. This was used to plot Kaplein-Meier survival curves between CD61+CD103+CD8+CD3+ patients and other remaining patients (survminer version 0.4.9, publicly-available coding). |
|---|---|

For manuscripts utilizing custom algorithms or software that are central to the research but not yet described in published literature, software must be made available to editors and reviewers. We strongly encourage code deposition in a community repository (e.g. GitHub). See the Nature Portfolio guidelines for submitting code & software for further information.

# Data

Policy information about availability of data

All manuscripts must include a data availability statement. This statement should provide the following information, where applicable:
- Accession codes, unique identifiers, or web links for publicly available datasets
- A description of any restrictions on data availability
- For clinical datasets or third party data, please ensure that the statement adheres to our policy

Raw proteomics data was deposited on Mendeley Data (dx.doi.org/10.17632/b2xdk4h5xm.1) and at the Proteome Xchange through PRoteomics IDEntifications Database (PRIDE) accession number PDX031794 and PDX045989. RNA and clinical data used for survival analysis were sourced directly from publicly available database, The Cancer Genome Atlas (TCGA) (https://portal.gdc.cancer.gov). The TCGA datasets used were the skin cutaneous melanoma (TCGA, PanCancer Atlas), lung cancer (TCGA, PanCancer Atlas) and lung cancer (University of Cologne). Patient clinical metadata for associated datasets was downloaded from cBioPortal (https://www.cbioportal.org). Raw data and reagents from all main and supplementary figures, beyond the mandatory dataset deposited on public repository, are available on request from T.D.

# Human research participants

Policy information about studies involving human research participants and Sex and Gender in Research.

| Reporting on sex and gender | Gender information was collected but not used for analysis in this study. The potential bias, such as the gender of patients, is unlikely to impact the final results as the samples collection and the analysis for each patient were done without any prior knowledge of this parameter, except that they are confirmed to have lung cancer but no metastasis. |
|---|---|
| Population characteristics | A total of 31 NSCLC patients were recruited and analysed for this study, ranging in age between 63 to 80 years old. Clinical parameters of patients are as informed in Extended Data Fig. 3 and Extended Data Table 2. Briefly, 7 patients are diagnosed with squamous carcinoma, 1 for plemorphic carcinoma and 23 for adenocarcinoma. 11 patients were diagnosed with TNM stage 1, 8 patients with TNM stage 2, 8 patients with TNM stage 3 and 4 patients with TNM stage 4. All patients were confirmed not to have any prior treatments and confirmed not to have metastatic cancer. |
| Recruitment | Confirmed non-metastatic NSCLC patients were recruited from the John Radcliffe Hospital, Oxford, United Kingdom, between December 2020 and April 2021. Subjects included both females and males who were between 63 to 80 years old. Ethics was approved by the NHS South Central – Oxford C Research Ethics Committee (REC: 19/SC/0173) under the Oxford Radcliffe Biobank (ORB) Tissue Access Committee ethics reference number 18/A026 and 20/A081. All procedures were performed according to the Declaration of Helsinki guidelines. Clinical parameters of individual patients are described in Extended Data Figure 3 and Source File 2. Patients provided voluntary informed written consent before surgery, and no compensation was provided. |
| Ethics oversight | Ethics oversight is carried out by the NHS South Central – Oxford C Research Ethics Committee (REC: 19/SC/0173) under the Oxford Radcliffe Biobank (ORB) Tissue Access Committee ethics reference number 18/A026 and 20/A081. |

Note that full information on the approval of the study protocol must also be provided in the manuscript.

# Field-specific reporting

Please select the one below that is the best fit for your research. If you are not sure, read the appropriate sections before making your selection.

☒ Life sciences  ☐ Behavioural & social sciences  ☐ Ecological, evolutionary & environmental sciences

For a reference copy of the document with all sections, see nature.com/documents/nr-reporting-summary-flat.pdf

# Life sciences study design

All studies must disclose on these points even when the disclosure is negative.

| Sample size | For the clinical cohort, a total of 31 patients with confirmed NSCLC (without prior treatment or metastasis) were included in this study. Sample sizes were based on maximal available sample sets, where detailed clinical data were also available from the biobank. For in vitro assays, each T cell assay was set up for 3 independent experiments, with sample size chosen based on the consistency between three independent replicates observed for similar functional T cell assays described previously (Peng et al., Nat Immunol, 2022, Peng et al., Nat Immuol 2020, Abd Hamid et al., Cancer Immunol Res, 2020, Abd Hamid et al., Cancer Immunol Res, 2019). For in vivo adoptive transfer model, the optimal sample size used were based on previous studies (Prota et al, Cancer Immunol Res, 2020; Shanderov et al., J Immuno Ther, 2021). |
|---|---|
| Data exclusions | For in vitro and in vivo T cells assay, the antigen-specificity are confirmed by tetramer staining prior to the assays performed, assays were carried out only if tetramer staining is at purity >90%. Additionally, T cell assays will only be carried out if the CD103 and/or CD61 expression on the T cell clones are at purity >80% (in comparison to the double negative T cell clones). For co-immunoprecipitation assay, initial flow cytometry staining of transduced primary CD8 T cells and U937 cells were performed before every assay. Co-IP will only take place if the CD103 and/or CD61 expression are at purity >80% (in comparison to the WT cells). For clinical flow cytometry analysis, no exclusion were carried out, in order to understand the full variability between patients. |
| Replication | For T cells mechanism and functional assays, replications of assays were performed, with 3 replicates per experiment and 3 independent repeats of the same assay, to ensure consistency between experiment of the same assay. All results gave similar successful outcome. For clinical analysis, samples were analysed on individual patients. Clinical analysis of each patients was performed once, without replication due to limited number of cells procured from tissue samples of each patient. 19 of the patients data was confirmed using two orthogonal approaches (flow cytometry and IHC), and all results gave similar successful outcome. |
| Randomization | Randomisation for the in vitro experiments were not relevant as input information are vital for identification of samples ran on machine such as flow cytometer. Randomisation for clinical cancer patients samples experiments were not relevant as clinical background of each patients was blinded to researcher by the Biobank before passing samples to researchers. For in vivo assay, following subcutaneous injection of tumor into the mice, at 48hours post tumor xenograft, mice were randomly allocated into two groups, with one group injected intravenously with the different T cell clones. Age-matched male and female mice were used, and randomly allocated for both groups. |
| Blinding | For in vivo adoptive transfer assay, T cell clones were prepared by one individual and clones were labelled with random alphabet. Allocation of mice into different groups, subcutaneous and intravenous cells injection into mice and the tumor volume measurement were carried out by another individual (from outside the main research group, ie collaborator) without prior knowledge of the association between the specific alphabet and specific T cell clones. For most clinical analysis, samples collection for each patient were done without any prior knowledge of the clinical parameters of each patient, except that they are confirmed to have lung cancer but no metastasis. Clinical parameters were referenced to and only included in the analysis when performing comparison between early stage and late stage patients. |

# Reporting for specific materials, systems and methods

We require information from authors about some types of materials, experimental systems and methods used in many studies. Here, indicate whether each material, system or method listed is relevant to your study. If you are not sure if a list item applies to your research, read the appropriate section before selecting a response.

## Materials & experimental systems

| n/a | Involved in the study |
|---|---|
| ☐ | ☒ Antibodies |
| ☐ | ☒ Eukaryotic cell lines |
| ☒ | ☐ Palaeontology and archaeology |
| ☐ | ☒ Animals and other organisms |
| ☒ | ☐ Clinical data |
| ☒ | ☐ Dual use research of concern |

## Methods

| n/a | Involved in the study |
|---|---|
| ☒ | ☐ ChIP-seq |
| ☐ | ☒ Flow cytometry |
| ☒ | ☐ MRI-based neuroimaging |

## Antibodies

| Antibodies used | | | | | | |
|---|---|---|---|---|---|---|
| Flourophore | Marker | Clone ID | Supplier | Catalogue number/RRID ID | Dilution |
| Purified | anti-CD3/CD28 | NA | StemCell Technologies | 10791/RRID:AB_2827806 | titer: 10ul |
| BUV395 | anti-CD103 | Ber-ACT8 | BD Biosciences | 564346/RRID:AB_2738759 | 1:33 |
| PerCP/Cy5.5 | anti-CD41 | HIP8 | Biolegend | 303719/RRID:AB_2561732 | 1:33 |
| PE | anti-CD51 | NKI-M9 | Biolegend | 327910/RRID:AB_940564 | 1:33 |
| PE/Cy7 | anti-CD27 | M-T271 | Biolegend | 256411/RRID:AB_2562258 | 1:33 |
| BUV496 | anti-CD28 | CD28.2 | BD Biosciences | 741168/RRID:AB_2870741 | 1:33 |
| BUV737 | anti-PD-1 | EH12.1 | BD Biosciences | 612791/RRID:AB_2870118 | 1:33 |
| BB515 | anti-Tim3 | 7D3 | BD Biosciences | 565568/RRID:AB_2744368 | 1:33 |
| APC/Cy7 | anti-CD39 | A1 | Biolegend | 328226/RRID:AB_2571981 | 1:25 |
| PerCP/Cy5.5 | anti-CD45RO | UCHL1 | Biolegend | 304222/RRID:AB_2174124 | 1:33 |

| BV650 | anti-CD3 | UCHT1 | BD Biosciences | 563851/RRID:AB_2744391 | 1:33 |
|---|---|---|---|---|---|
| BUV805 | anti-CD8 | SK1 | BD Biosciences | 612889/ RRID:AB_2833078 | 1:50 |
| BV605 | anti-CD69 | FN50 | Biolegend | 310938/RRID:AB_2562307 | 1:50 |
| Purified | anti-CD103 | EP206 | Leica Biosystem | PA0374/NA | 1:1000 |
| Purified | anti-CD8 | 4B11 | Leica Biosystem | PA0183/NA | 1:1500 |
| Purified | anti-CD61 | VI-PL2 | Abcam | AB_1086-711/NA | 1:250 |
| AF647 | anti-ZAP70 (pY292) | J34-602 | BD Biosciences | 558515/RRID:AB_647148 | 1:33 |
| AF647 | anti-PLCg1 (pY783) | 27/PLC | BD Biosciences | 557883/RRID:AB_396921 | 1:25 |
| PE | anti-Lck (pY505) | 4/LCK-Y505 | BD Biosciences | 558552/RRID:AB_397084 | 1:25 |
| AF647 | anti-Vav1 pY174 | EP510Y | Abcam | ab76225/RRID:AB_1524546 | 1:25 |
| PE/Cy7 | anti-CD107a | H4A3 | Biolegend | 328617/RRID:AB_11147761 | 1:33 |
| BV421 | anti-E-Cadherin | 67A4 | BD Biosciences | 743712/RRID:AB_2741690 | 1:33 |
| PE/Cy7 | anti-CD8 | SK1 | Biolegend | 344712/RRID:AB_2044008 | 1:25 |
| Purified | anti-CD61 | PM6/13 | Novus Biotechnology | NBP1-28398/ RRID:AB_1853098 | 1:10 |
| BV 510 | anti-CD56 | 5.1H11 | Biolegend | 362534/RRID:AB_2565633 | 1:33 |
| BV510 | anti-CD11b | ICRF44 | Biolegend | 301334/RRID:AB_2562112 | 1:50 |
| APC/Cy7 | anti-CD3 | UCHT1 | BD Biosciences | 557832/RRID:AB_396890 | 1:33 |
| BV421 | anti-CD61 | VI-PL2 | BD Biosciences | 744381/RRID:AB_2742194 | 1:25 |
| FITC | anti-CD41 | HIP8 | Biolegend | 303704/RRID:AB_314374 | 1:50 |
| FITC | anti-CD51 | NKI-M9 | Biolegend | 327908/RRID:AB_940560 | 1:33 |
| PE | anti-CD49a | T2/S7 | BD Biosciences | 568716/NA | 1:33 |
| BUV496 | anti-CD49a | T2/S7 | BD Biosciences | 755215/NA | 1:50 |
| PE/Cy7 | anti-CD69 | FN50 | Biolegend | 310912/RRID:AB_314847 | 1:33 |
| BV785 | anti-CD39 | A1 | Biolegend | 328240/ RRID:AB_2814191 | 1:25 |
| BV421 | anti-Tim-3 | 7D3 | BD Biosciences | 565562/RRID:AB_2744369 | 1:33 |
| PE | anti-TIGIT | TgMab-2 | BD Biosciences | 568672/NA | 1:33 |
| APC/Cy7 | anti-IFNg | B27 | Biolegend | 506524/RRID:AB_2566136 | 1:33 |
| BV785 | anti-TNFa | Mab11 | Biolegend | 502948/ RRID:AB_2565858 | 1:33 |
| APC | anti-integrin beta 7 | FIB504 | Biolegend | 321208/RRID:AB_571965 | 1:50 |
| PE | anti-granulysin | DH2 | Biolegend | 348004/RRID:AB_2263307 | 1:33 |
| AF488 | anti-granzyme M | 4B2G4 | ThermoFisher | 53-9774-42 RRID:AB_2848451 | 1:25 |
| AF647 | anti-granzyme B | GB11 | BD Biosciences | 560212/ RRID:AB_11154033 | 1:50 |
| BUV737 | anti-CCL5 | 2D7/CCR5 | Biolegend | 612808/RRID:AB_2870133 | 1:33 |
| Purified | anti-E-Cadherin | 36B5 | Leica Biosystems | PA0387/NA | 1:500 |
| Purified | anti-E-Cadherin | DECMA-1 | BioLegend | 147302/RRID: AB_2563038 | 1:33 |
| Purified | anti-CD49d | 9F10 | BioLegend | 304302/RRID: AB_314428 | 1:50 |
| Purified | anti-CD61 | VI-PL2 | BioLegend | 336402/RRID: AB_1227584 | 1:33 (FC) |
| Purified | anti-Granzyme B | GB11 | Invitrogen | MA1-80734/RRID: AB_931084 | 1:33 |
| AF488 | anti-CD103 | Ber-ACT8 | BioLegend | 350208/RRID: AB_10641844 | 1:33 |
| AF488 | anti-TCR α/β | IP26 | BioLegend | 306712/RRID: AB_528967 | 1:33 |
| AF647 | anti-Integrin β7 | FIB504 | BioLegend | 321222/RRID: AB_2715979 | 1:50 |
| AF647 | anti-CD61 | VI-PL2 | BioLegend | 336408/RRID: AB_2128750 | 1:33 |
| AF647 | anti-CD103 | EPR4166(2) | Abcam | ab225153/RRID:AB_2884945 | 1:200 |
| Purified | anti-CD3 | OKT3 | Biolegend | 317302/RRID:AB_571927 | 1:10 |
| BV421 | anti-CD103 | Ber-ACT8 | Biolegend | 350213/RRID:AB_2563514 | 1:33 |
| PE/Cy7 | anti-CD61 | VI-PL2 | Biolegend | 336416/RRID:AB_2566692 | 1:33 |
| FITC | anti-integrin beta 7 | FIB504 | Biolegend | 321212/RRID:AB_830856 | 1:50 |
| AF647 | anti-cMyc | 9E10 | Biolegend | 626808/RRID:AB_2888732 | 1:100 |
| PE | anti-DYKDDDDK | L5 | Biolegend | 637310/RRID:AB_2563148 | 1:100 |
| BV421 | anti-integrin beta 7 | FIB504 | BD Biosciences | 564283/RRID:AB_2738728 | 1:50 |
| PE | anti-CD61 | VI-PL2 | Biolegend | 336406/RRID:AB_2128752 | 1:33 |
| Purified | anti-FLAG | M2 | Sigma-Aldrich | F3165/RRID:AB_259529 | 1:500 |
| Purified | anti-b-actin | AC-74 | Sigma-Aldrich | A5316/RRID:AB_476743 | 1:1000 |
| Purified | anti-CD103 | EPR4166(2) | Abcam | 129202/RRID:AB_11142856 | 1:500 |
| PE | anti-CD107a | H4A3 | Biolegend | 328618/RRID:AB_11147761 | 1:20 |
| Purified | anti-cMyc | 9E10 | Sigma-Aldrich | M4439/RRID:AB_439694 | 1:500 |
| BUV496 | anti-CD62L | DREG-56 | BD Biosciences | 741155/RRID:AB_2870731 | 1:50 |
| PE/Cy7 | anti-CCR7 | G043H7 | Biolegend | 353226/RRID:AB_11126145 | 1:50 |
| FITC | anti-XCL2 | 06 | Novus Biological | NBP3-06177F/NA | 1:25 |

| Validation | All antibodies used in this study are commercially available. Antibodies used in a specific species or application have been appropriately validated by manufacturers for that application and this information is provided on their website and product information datasheets. Within manufacturers website, details of verified reactivity, application used are verified by quality-testing (for flow cytometry, QC will be by flow cytometric staining of selected cell lines), with species application tested and verified are evidenced by product citations on the product webpage of the manufacturer. All antibodies described here have been further optimised for an appropriate concentration by testing several dilutions. |
|---|---|

# Eukaryotic cell lines

Policy information about cell lines and Sex and Gender in Research

| Cell line source(s) | Wild-type HEK293T and HCT116 are from ATCC (CRL-3216, CCL-247). CD103 and CD61 transduced primary CD8 T cell lines |
|---|---|

| Cell line source(s) | was established in the lab. CD103+ SSX2-specific CD8 T cell clone, CD103- SSX2-specific CD8 T cell clone, CD103+ NYESO-1-specific CD8 T cell clone, CD103- NYESO-1-specific CD8 T cell clone were established in the lab. |
|---|---|
| Authentication | Cell lines from ATCC were used at the earliest passage, they were no further authentication. CD103 and CD61 transduced cell lines were verified by flow cytometry staining of CD103 and CD61. |
| Mycoplasma contamination | All cell lines were tested negative for mycoplasma. |
| Commonly misidentified lines (See ICLAC register) | No misidentified cell lines were used according to the version 11 of register of misidentified cell lines. |

# Animals and other research organisms

Policy information about studies involving animals; ARRIVE guidelines recommended for reporting animal research, and Sex and Gender in Research

| Laboratory animals | Immunodeficient NSG mice (strain NOD.Cq-Prkdc scid Il2rgtm1Wjl/SzJ), all mice were housed in ventilated cages, maintained under specific pathogen-free conditions, with 12hrs dark/light cycle, ambient room temperature daily at 18-23 celsius, 40-60% humidity and used at 8-10 weeks old of either sex. |
|---|---|
| Wild animals | The study did not involve wild animals. |
| Reporting on sex | Sex information was not used in the analysis of the in vivo model, because the model used are based on SCID mice system and the human tumor xenograft (that express the specific HLA and tumor antigen) recognised specifically by the human T cells used for adoptive transfer. Mice hormonal and its immune system would not affect the outcome of the data, because the cancer cell killing will be directed specifically by human TCR-pMHC interaction (plus action of co-stimulatory/inhibitory receptors). Therefore, age-matched male and female mice were used for both treatment groups.. |
| Field-collected samples | The study did not involves samples collected from the field. |
| Ethics oversight | All mice experiments were performed in accordance with Animals (Scientific Procedures) Act 1986 and according to the University of Oxford Animal Welfare and Ethical Review Body (AWERB) guidelines, and operating under the UK Home Office license PBA43A2E4. |

Note that full information on the approval of the study protocol must also be provided in the manuscript.

# Flow Cytometry

## Plots

Confirm that:

☒ The axis labels state the marker and fluorochrome used (e.g. CD4-FITC).

☒ The axis scales are clearly visible. Include numbers along axes only for bottom left plot of group (a 'group' is an analysis of identical markers).

☒ All plots are contour plots with outliers or pseudocolor plots.

☒ A numerical value for number of cells or percentage (with statistics) is provided.

## Methodology

| Sample preparation | For ex vivo samples, For the ex vivo multicolour flow cytometry, cell suspension used for the assay was isolated from tissue as previously described20. Briefly, tissues were cut into small pieces using a pair of scissor and forceps before enzymatically dissociated into cell suspensions in RPMI-1640 using human tumor dissociation kit (Miltenyi Biotech), following protocol provided by the supplier. Following the enzymatic dissociation, cells were filtered through 100um strainer to remove indigestible parts of the tissue, with dead cells or debris were removed by centrifugation at 1,500 rpm for 10 minutes. Cells were then resuspended in RPMI-1640 supplemented with 10% FCS (Sigma Aldrich), 2 mM L-glutamine (Sigma Aldrich) and 1% v/v (500U/ml) penicillin/streptomycin (Sigma Aldrich). For peripheral blood, the PBMC were isolated using Ficoll-Hypaque gradient isolation, followed by centrifugation to pellet the PBMC and resuspended in the same complete media as above.For each immunophenotyping staining, 1M cells of paratumor tissue, tumor and peripheral blood were first stained with Live/Dead Fixable Aqua Stain Kit (ThermoFisher) for 20 minutes at 4C. For surface staining, cells were washed and then stained with dumping markers: BV510 anti-CD56 (Biolegend) and BV510 anti-CD11b (Biolegend), T cells markers: BUV805 anti-CD8 (BD Biosciences) and BV650 or APC/Cy7 anti-CD3 (BD Biosciences), integrins: BUV395 anti-CD103 (BD Biosciences), BV421 or AF647 anti-CD61 (BD Biosciences), PerCP/Cy5.5 or FITC anti-CD41 (Biolegend), APC anti- integrin B7 (Biolegend) and PE or FITC anti-CD51 (Biolegend), tissue-resident T cell markers: PerCP/Cy5.5 anti-CD45RO (Biolegend), PE or BUV496 anti-CD49a (BD Biosciences) and PE/Cy7 or BV605 anti-CD69 (Biolegend), tumor-reactive TILs marker: APC/Cy7 or BV785 anti-CD39 (BD Biosciences and Biolegend), T cell differentiation markers: PE/Cy7 anti-CD27 (BD Biosciences) and BUV496 anti-CD28 (BD Biosciences), inhibitory markers: BUV737 anti-PD-1 (BD Biosciences), BV421 or BB515 anti-Tim-3 (BD Biosciences) and PE anti-TIGIT (BD Biosciences) for another 20 minutes at 4C. For intracellular cytokine staining, cells were T cells were treated with 0.7ug/ml Monensin and 1ug/ml Brefeldin A (BD Biosciences), washed and permeabilised with BD CytoFix/CytoPerm Solution for 20 minutes at 4C, before stained with cytokines: APC/Cy7 anti-IFNg (Biolegend) and BV785 anti-TNFa (Biolegend), cytolytic molecules: PE anti-granulysin (Biolegend), AF488 anti-granzyme M (ThermoFisher), AF647 anti-granzyme B (BD Biosciences), chemokines: BUV737 anti-CCL5 (Biolegend) and FITC anti-XCL2 for another 20 minutes at 4C. Following antibodies staining, cells were fixed with 1X CellFix (BD Biosciences) and acquired on BD LSR Symphony (BD |
|---|---|

Biosciences) and analysed on FlowJo V.10 (BD Biosciences).

For in vitro assays, T cell lines and clones were T cells were co-culture with cancer cells at an E:T ratio of 1:10 at 37C for 15 minutes, 30 minutes or 2 hours. Following T cells activation, cells were stained with Live/Dead Fixable Aqua Dead Cell Stain Kit (Thermo Fisher) before fixed with BD Cytofix Fixation Buffer for 10 minutes at 37⬚C. Cells were then permeabilised using BD Phosflow Perm Buffer III for 30 minutes at 4C before stained with FITC anti-ZAP70 (BD Biosciences), AF647 anti-ZAP70 (pY292) (BD Biosciences), AF488 anti-PCLg1 (BD Biosciences), AF647 anti-PLCg1 (pY783) (BD Biosciences),  PE anti-Lck (pY505) (BD Biosciences), AF647 anti-Lck (BD Biosciences) and AF647 anti-Vav1 (pY174) (Abcam). Cells were then acquired immediately on Attune Nxt flow cytometer (ThermoFisher) and analysed on FlowJo V.10 (BD Biosciences). To assess contribution of CD61 towards TCR signalling proteins activities, T cells were treated with 10nM aminogenistein (Lck inhibitor, Santa Cruz Biotechnology) before the T cells activation and the Phosflow assay were performed (Phosflow assay). For T cell cytotoxicity assay, Cancer cells were initially stained with 0.5ug/ml CFSE (ThermoFisher) prior to co-culture with T cells at an E:T ratio of 1:10 at 37C for either 2, 4, 6 and 8 hours. Cells were then stained with 7-AAD (BD Biosciences) and BV421 anti-E-Cadherin (Biolegend) and PE/Cy7 anti-CD8 (BD Biosciences) before acquiring on the Attune Nxt flow cytometer (ThermoFisher) and analysed on FlowJo V.10 (BD Biosciences). To evaluate the T cell cytotoxic efficacy, the WT T cells were treated with anti-CD61 (10ug/ml, PM6/13, Novus Biotechnology), in parallel to the T cells activation. For T cell proliferation assay, 1M cells of paratumor tissue or of tumor tissue were stained with 0.5ug/ml CFSE prior to activation with 10ul anti-CD3/CD28 (StemCell Technologies). The cells were incubated at 37C for 72 hours. After, the cells were stained with Live/Dead Fixable Aqua Cell Stain Kit (Thermo Fisher) for 20 minutes at 4C before being stained with conjugated antibodies against BV650 anti-CD3 (BD Biosciences), BUV805 anti-CD8 (BD Biosciences), BUV395 anti-CD103 (BD Biosciences), BV421 anti-CD61 (BD Biosciences), PerCP/Cy5.5 anti-CD45RO (Biolegend), PE anti-CD49a (BD Biosciences and Biolegend) and PE/Cy7 anti-CD69 (Biolegend). Following antibodies staining, cells were fixed with 1X CellFix (BD Biosciences) and acquired on BD LSR Symphony (BD Biosciences) and analysed on FlowJo V.10 (TreeStar Inc.). Cells were considered proliferative based on decrease in CFSE fluorescence, within the 1st downward peaks of CFSE onwards.

| | |
|---|---|
| Instrument | Samples were acquired on BD LSR Symphony or ThermoFisher Attune Nxt Flow Cytometer. |
| Software | Data were collected using FACS Diva v9.0.1 or Attune NXt software V3.2.1 and analysed using FlowJo v10 software for Mac OS. |
| Cell population abundance | For the clinical samples, cells were not sorted, all analysis were done with multicolour flow cytometry. The specific population of interest ex vivo ranges from 2% to 78%. For the cancer-specific T cell clones, the process of identification, processing, sorting and in vitro culture is as previously described (Abd Hamid et al., Cancer Immunol Res, 2020). CD103 and/or CD61 transduced cell lines were not sorted before performing co-IP, as the purity is high (>80%). |
| Gating strategy | For all flow cytometry-based experiments, cells were first gated on single lymphocytes by a forward side scatter gate. Identification of live T cells were done based on excluding dead cells (by Fixable Live/Dead staining), CD3 and CD8 staining, before identification for CD103+ CD61+ cells. For intracellular cytokine staining (ICS), phosflow staining and integrins validation on transduced U937 cell lines, positive/negative population were gated according to corresponding negative controls, either unstimulated samples or wild-type unstimulated/non-transduced samples. |

☒ Tick this box to confirm that a figure exemplifying the gating strategy is provided in the Supplementary Information.

