## [Peer Review File · Nature Immunology]

Peer Review Information

Journal: Nature Immunology

Manuscript Title: Unconventional human CD61 pairing with CD103 promotes TCR signalling and antigen-specific T cell cytotoxicity

Corresponding author name(s): Professor Tao Dong

Reviewer Comments & Decisions:

Decision Letter, initial version:

20th Jul 2023

Dear Professor Dong,

Your Article, "An unconventional integrin promotes human immunity" has now been seen by 3 referees. You will see from their comments copied below that while they find your work of considerable potential interest, reviewer in particular has raised substantial concerns that must be addressed. In light of these comments, we cannot accept the manuscript for publication, but would be very interested in considering a revised version that addresses these serious concerns.

Also, we have looked over your Author Response to these criticisms, and I am happy to say that we are pleased by your revision plan. Our only concern is with regards to your reply to ref3 comment 7. Our interpretation of this comment is that the reviewer is probably less keen on merely seeing some controls added to validate this experiment with pinocembrin. We think the reviewer is probably expecting an alternative experiment entirely to support the conclusions here.

If you wish to submit a substantially revised manuscript, please bear in mind that we will be reluctant to approach the referees again in the absence of major revisions.

If you choose to revise your manuscript taking into account all reviewer and editor comments, please highlight all changes in the manuscript text file in Microsoft Word format].

* Include a "Response to referees" document detailing, point-by-point, how you addressed each

referee comment. If no action was taken to address a point, you must provide a compelling argument. This response will be sent back to the referees along with the revised manuscript.

* If you have not done so already please begin to revise your manuscript so that it conforms to our Article format instructions at <http://www.nature.com/ni/authors/index.html>. Refer also to any guidelines provided in this letter.

The Reporting Summary can be found here:

When submitting the revised version of your manuscript, please pay close attention to our [href="https://www.nature.com/nature-portfolio/editorial-policies/image-integrity">Digital Image Integrity Guidelines](https://www.nature.com/nature-portfolio/editorial-policies/image-integrity). and to the following points below:

[REDACTED]

If you wish to submit a suitably revised manuscript we would hope to receive it within 6 months. If you cannot send it within this time, please let us know. We will be happy to consider your revision so long as nothing similar has been accepted for publication at Nature Immunology or published elsewhere.

Nature Immunology is committed to improving transparency in authorship. As part of our efforts in this direction, we are now requesting that all authors identified as 'corresponding author' on published papers create and link their Open Researcher and Contributor Identifier (ORCID) with their account on the Manuscript Tracking System (MTS), prior to acceptance. ORCID helps the scientific community achieve unambiguous attribution of all scholarly contributions. You can create and link your ORCID from the home page of the MTS by clicking on 'Modify my Springer Nature account'. For more information please visit www.springernature.com/orcid.

Thank you for the opportunity to review your work.

Sincerely,

Nick Bernard, PhD
Senior Editor
Nature Immunology

Reviewers' Comments:

Reviewer #1:

Remarks to the Author:

This is an interesting and potentially important study that identifies the expression of CD61 on CD103+ T cells.

CD61 appears to be seen on early differentiated effector cells and associates with higher levels of functional activity and reduced features of cellular exhaustion.

The model of promiscuous CD103 engagement is important and novel.

The work benefits from the collaboration of T cell immunology and excellence in synapse analysis - and is of the highest technical quality

-I am surprised that CD61 is not mentioned in the title. Indeed, in my view the title is too broad ('human immunity') and should be more specific and accurate

-I would caution calling all CD39+ T cells as cancer-specific. Although they are enhanced in this specificity within the tumour microenvironment, there are many T cells that express this marker that are not tumour-specific.

-A limitation is that the cellular work is limited to clones derived from two patients, but this stated as such within the manuscript. The findings on the larger cohort of NSCLC patients are valuable.

-The team could potentially have mined the very large published gene datasets to look at how CD61 associates with cancer outcome, potentially in relation to T cell infiltrate, but I do not see this as obligatory at this stage.

-some minor proof reading is needed - e.g. missed word at line 588 and 590

Reviewer #2:

Remarks to the Author:

The manuscript by Hamid and colleagues shows a novel and interesting interaction between CD61 and CD103, which appears to promote T cell function. This is a solid mechanistic study that contributes to our understanding of the promiscuous nature of CD103, which has broad importance for T cell biology, as well as a previously unknown function for CD61. One of my major criticisms is the suggestion that CD103-expressing TIL are tissue resident. CD103 is expressed by effector/exhausted TILs (Duhon 2018 - cited #17) and could equally be considered a biomarker for any CD8+ T cell that has experienced TGF β signaling. Residency is incredibly hard to prove in humans and use of this marker is not sufficient. Further, given that the authors show that CD61 is upregulated upon T cell activation,

and that this is transient, it is likely that CD61+ TIL are activated T cells that have recently encountered antigen.

Major Comments:

- Fig 4e. I am struck by the differential protective capacity of CD61+ and CD61- T cells, and that these cells may be epigenetically programmed to be less functional. CD61 may not be the sole cause for this difference in tumor growth. The overexpression of CD61 in the CD61- NY-ESO T cells or deletion of CD61 in CD61+ NY-ESO T cells would greatly strengthen the authors conclusions, as would the demonstration that CD61 blockade prevents tumour clearance in vivo.
- Is CD61:CD103 directly binding to Lck to bring it to the TCR within the synapse? Does this compensate for CD8?
- Figure 5d – This requires some representative images to show how these regions were determined.

Minor Comments:

- Fig2c – the authors note that CD103-CD61 co-localisation increases within 5-10min of T cell activation and then decreases by 10-15min. Are these differences meaningful and are there statistics to support this (it is difficult to assess this observation)? can the authors show CD61 expression at t=0 and 5min post activation (only 2h is shown).
- Line 383-386 – Be careful with this interpretation. As I understand it, the TCGA data used is not sorted on CD3+ CD8+ T cells but represents expression of 4 genes within the excised tumour. This doesn't necessarily mean the CD61 (or the CD103/ITGAE) detected was expressed by the T cells within this transcriptional data.
- The title of the paper is vague and uninformative.

Reviewer #3:

Remarks to the Author:

This paper describes expression of CD61 on CD103+ T cells and suggests that these proteins interact to form a novel integrin combination in T cells. The manuscript suggests that CD61 and CD103 co-localise transiently in microclusters at the synapse and that CD61 enhances TCR signaling.

There are several concerns about the approaches and results that are summarized:

(i) The basic premise starts from comparing the proteome of TCR-matched CD103+ and CD103- T cell clones. Using 1 million T cells per sample cells with CD103+/- were compared. I did not see notes on biological or technical replicates and wondered how reproducible results were? What were the n numbers for proteomics? What was the variability?

(ii) The authors report increased CD61 expression in T cells after culture with antigenic cancer cells. As cancer cells have been reported to express CD61, is it possible that CD61 has been acquired from cancer cells?

(iii) The pull downs to show an interaction between CD61 and CD103 were all carried out using over-expression of both integrin chains. Was there a reason for this? Ideally co-immunoprecipitations would be shown from endogenously expressed integrin chains. Even then co-precipitation does not necessarily show a direct interaction as other cellular components could be contributing. This data needs to be strengthened.

(iv) The FACS data shown in 2a indicates a significant surface expression of CD61 after activation (assuming that the x axis is logarithmic; x and y values need to be added to almost all FACS plots including this one). The authors suggest that this is a transient upregulation during which CD61 and CD103 interact and use TIRF imaging to support this idea. In Figure 2b some overlap in the signals for CD61 and CD103 in areas up to 1 micron across are used to suggest that CD61 and CD103 in microclusters (usually thought to be ~0.2microns diameter). The signals for CD61 and CD103 are not coincident in the way that would suggest colocalization but might rather be the result of TIRF showing a 2D image of a 200nm 3D depth. This would occur if the T cell membrane were not completely flat on the bilayer. What is needed is a membrane marker that defines where the cell membrane is relative to the bilayer.

Extended data figure 4b shows a time course by TIRF showing cells after 5, 10 and 15 minutes on bilayers labelled for TCR, CD61 and CD103 with images that differ from Figure 2b (also a 10-minute time point). Although the middle panel from 2b and Ext data 4b should look the same, they do not, with many more microcluster-like dots showing little overlap at 5 minutes and increasing clusters at 10 and 15 minutes. The precise overlap between individual clusters that you would expect to see if CD61 and CD103 were colocalized is not evident.

A simpler interpretation of the data might be that the clusters are simply the points of contact as the T cell comes into contact with the bilayer. This is supported by the time course in Ext data 4b. in which more clusters accumulate over time.

(v) The manuscript suggests that CD61 enhances TCR signaling as there are differing levels of signaling proteins in CD61+ and CD61- T cell clones. These data include a "complete CD61 knock-out", that the methods reveal was generated using pinocembrin treatment (lines 892-4). It is not clear how pinocembrin works, but it is misleading to refer to it as a "complete knock-out" as this drug treatment will undoubtedly produce many different changes in T cell clones. The data provided does not demonstrate a direct link between CD61 and the changes observed in signaling proteins.

(vi) Genistein is a very broad inhibitor of tyrosine kinases and is not specific to Lck. There will be many off-target effects.

Author Rebuttal to Initial comments

Response to Editor:

Dear Professor Dong,

Your Article, "An unconventional integrin promotes human immunity" has now been seen by 3 referees. You will see from their comments copied below that while they find your work of considerable potential interest, reviewer in particular has raised substantial concerns that must be addressed. In light of these comments, we cannot accept the manuscript for publication, but would be very interested in considering a revised version that addresses these serious concerns.

Also, we have looked over your Author Response to these criticisms, and I am happy to say that we are pleased by your revision plan. Our only concern is with regards to your reply to ref3 comment 7. Our interpretation of this comment is that the reviewer is probably less keen on merely seeing some controls added to validate this experiment with pinocembrin. We think the reviewer is probably expecting an alternative experiment entirely to support the conclusions here.

R: We thank the editor for the opportunity to revise our manuscript. We acknowledge the concerns raised by the reviewers and have made substantial revision that address these concerns as per the summary revision and the point-to-point response below.

We agree with the editor interpretation of the reviewer 3 comment 7, and has now revised our response. We have now generated alternative data and experiments in replacement of pinocembrin. This alternative data involves using different T cell conditions, namely generating CD61 knock-out T cells using CRISPR-Cas9 method, serial dilution (knock-down) of CD61 siRNA treatments and the use of anti-CD61 blocking antibody treatment, on the WT CD61+ T cells. These new T cell conditions are now used the revised functional assay. Using these new T cell conditions, we provided new evidences that further support our manuscript conclusion on the important CD61 role in regulating TCR signalling and therefore T cell cytotoxicity and effector functions, as per new Figure 3c-g, 4a-d and Extended Data Figure 7a-c. Detailed rebuttal for this comment is now on our R7 to reviewer 3 comment 7 below.

If you wish to submit a substantially revised manuscript, please bear in mind that we will be reluctant to approach the referees again in the absence of major revisions.

If you choose to revise your manuscript taking into account all reviewer and editor comments, please highlight all changes in the manuscript text file in Microsoft Word format].

R: We thank the editor for allowing for the revision and all changes has now been highlighted as track-changes in the text file.

* Include a "Response to referees" document detailing, point-by-point, how you addressed each referee comment. If no action was taken to address a point, you must provide a compelling argument. This response will be sent back to the referees along with the revised

manuscript.

* If you have not done so already please begin to revise your manuscript so that it conforms to our Article format instructions at <http://www.nature.com/ni/authors/index.html>. Refer also to any guidelines provided in this letter.

R: The manuscript has been revised to conform to the format instructions.

The Reporting Summary can be found here:
<https://www.nature.com/documents/nr-reporting-summary.pdf>

R: A revised checklist has now been completed and included in this resubmission.

When submitting the revised version of your manuscript, please pay close attention to our [href="https://www.nature.com/nature-portfolio/editorial-policies/image-integrity">Digital Image Integrity Guidelines](https://www.nature.com/nature-portfolio/editorial-policies/image-integrity). and to the following points below:

R: Scans and images of gels and western blots has now been revised to conform to the guideline.

R: New mass spectrometry data and other metadata files has now been archived in the Proteomics Identification Database (PRIDE), Project accession number PDX045989, as well as on Mendeley Data depository at <http://dx.doi.org/10.17632/b2xdk4h5xm.1>. Reviewer account details can be provided upon request.

Dear Dr Bernard,

We greatly appreciate the opportunity to revise our manuscript for publication within Nature Immunology and address the concerns raised by you and the reviewers. We have now revised our manuscript, and highlighted and track-changes all revision. The detailed responses follow, but for ease of reference, we have summarised below the changes:

1. As requested, the manuscript has been carefully reviewed and clarified throughout.
2. Revision of the manuscript title, 'An unconventional CD61 integrin expression on human antigen-specific CD8+ T cells promotes anti-tumor effector and cytotoxic immunity'.
3. New data generated as requested by the reviewers have now been included in the revised manuscript;
 - a. New T cell conditions as per request by Reviewer 2 Comment 4 and Reviewer 3 Comment 7 (Extended Data Figure 7a):
 - i. CD61 CRISPR knock-out (KO) T cell clones, generated from the original WT CD61+ T cell clones,
 - ii. Gradient/serial dilution of CD61 siRNA treatments on the original WT CD61+ T cell clones,
 - iii. Anti-CD61 blocking antibody, treated on the original WT CD61+ T cell clones.
 - b. These new T cell conditions are now used for revised functional Phosflow assay and T cell killing assay (to show neutralising/eliminating CD61 impairs T cell cytotoxic function) (Figure 3c-f, Figure 4a-d, Figure 5c, Extended Data Figure 7b).
 - c. Mass spectrometry analysis of the integrins cell lysates (Figure 2g and Extended Data Figure 4g) to validate potential interaction.

- d. TRIF showing unchanged membrane topography when visualising synapse (Extended Data Figure 10a-b)
 - e. Histogram plot of Zap70 phosphorylation level using Lck specific inhibitor, A770041 (Figure 3g)
 - f. Extended Data Figure 9. Visiopharm APP's tumor islets 'clustering' margin areas width lining on IHC images
4. Additional revisions of figures and extended data
- a. We have updated figures and extended data labelling
 - i. Figure 5f % of cells with 'tumor-reactive' phenotype
 - ii. Extended Data Figure 2, to include additional markers for Trm identification (CD62L, CD45RA, CCR7)
 - iii. Figure 2c-d to include p value statistics
 - iv. Extended Data Figure 4a to include more datapoints prior to 2 hours CD61 expression
 - v. FACS plot labelling (Figure 1d, 2a, 2h, 3g, 5a, 6a, 6e, Extended Data Figure 4e, 7a-b)
5. We have added five new authors in recognition of their contribution to this work, Adan Pinto Fernandes, Simeon Draganov and Iolanda Vendrell (mass spectrometry analysis of cell lysates), Paul Supp (for sorting of CD61 CRISPR KO T cells) and Guihai Liu (for identification, culturing and cloning of CD61 CRISPR KO T cell clones).

Response to Referees

Reviewer #1

(Remarks to the Author)

C1: *This is an interesting and potentially important study that identifies the expression of CD61 on CD103+ T cells. CD61 appears to be seen on early differentiated effector cells and associates with higher levels of functional activity and reduced features of cellular*

exhaustion. The model of promiscuous CD103 engagement is important and novel.

The work benefits from the collaboration of T cell immunology and excellence in synapse analysis - and is of the highest technical quality

R1: We are grateful to the reviewer for the complimentary remarks and positive feedbacks on our manuscript.

C2: *-I am surprised that CD61 is not mentioned in the title. Indeed, in my view the title is too broad ('human immunity') and should be more specific and accurate*

R2: We agree with the comment by the reviewer. We have now revised the title to include CD61 and to be more specific.

We would like to propose the title: 'An unconventional CD61 integrin expression on human antigen-specific CD8+ T cells promotes anti-tumor effector and cytotoxic immunity', at line 1-2.

C3: *-I would caution calling all CD39+ T cells as cancer-specific. Although they are enhanced in this specificity within the tumour microenvironment, there are many T cells that express this marker that are not tumour-specific.*

R3: We thank the reviewer for the cautionary advice. We have revised the relevant text, and defined them as "potential tumour-reactive T cells", as per line 513-523. and revised Figure 5f y-axis labelling.

C4: *-A limitation is that the cellular work is limited to clones derived from two patients, but this stated as such within the manuscript. The findings on the larger cohort of NSCLC patients are valuable.*

R4: We thank the reviewer for appreciating our effort in utilising the valuable larger cohort of NSCLC patients to validate and overcome the limitation of the in vitro cellular work.

C5: *-The team could potentially have mined the very large published gene datasets to look at how CD61 associates with cancer outcome, potentially in relation to T cell infiltrate, but I do not see this as obligatory at this stage.*

R5: We appreciate the reviewer's comment. We agree that this is a valid point that merits further investigation in the future as it is not obligatory at this stage. We now have added a sentence on the reviewer suggestion in the discussion section, on text line 686-688.

C6: *-some minor proof reading is needed - e.g. missed word at line 588 and 590*

R6: We thank the reviewer for the comment, and additional proofreading has been carried out.

Reviewer #2

(Remarks to the Author)

C1: *The manuscript by Hamid and colleagues shows a novel and interesting interaction between CD61 and CD103, which appears to promote T cell function. This is a solid mechanistic study that contributes to our understanding of the promiscuous nature of CD103, which has broad importance for T cell biology, as well as a previously unknown function for CD61.*

R1: We appreciate Reviewer 2's complimentary remarks and positive feedbacks on our study.

C2: *One of my major criticisms is the suggestion that CD103-expressing TIL are tissue resident. CD103 is expressed by effector/exhausted TILs (Duchen 2018 – cited #17) and could equally be considered a biomarker for any CD8+ T cell that has experienced TGFβ signaling. Residency is incredibly hard to prove in humans and use of this marker is not sufficient.*

R2: We appreciate the comment by the reviewer. We agree with the reviewer that the CD103 marker alone is not enough to show residency. Previous studies by Trm developmental research groups of Donna Farber (Columbia University), Laura McKay (University of Melbourne), Fathia Mami-Chouaib (Pasteur Institute) and others have utilised multiple markers to denote Trm cells, including CD103, CD69, CD45RO and CD49a (for identifying tissue-resident T cells) and the lack of CD62L, CX3CR1 and S1PR1 (of exit cue markers) (Kumar et al., cell Rep, 2017; Savas et al., Nat Med, 2018; Ganesan et al., Nat Immunol., 2017; Webb et al., Clin Cancer Res, 2014; Park et al., Nature, 2019; Malik et al., Sci Immunol, 2017; El-Asady, J Exp Med, 2005; Sathaliyawala et al., Immunity, 2013).

In our current manuscript, we therefore included these set of markers to denote Trm TILs in our cohort of NSCLC patients, as shown in Extended Data Figure 2. We use the term 'Trm TILs' to represent CD103+CD69+CD49a+CD45RO+ T cells. In addition to this, we confirmed the absence of exit cues markers (CD62L, CCR7) and non-memory cell marker (CD45RA) on the population of interest. We have therefore revised Extended Data Figure 2 to also show gating strategy involving CD62L, CCR7 and CD45RA. We further revise text line 492-495 to emphasize the use of these additional markers for identification of Trm TILs and Methods text line 1271-1272, 1730 for the new information on the flow cytometric antibodies used to detect CD62L, CCR7 and CD45RA.

For the above reasons and in agreement with the reviewer, we avoided the use of the term 'Trm T cells' to denote the CD103+ T cell clones. We instead chose to use the term 'CD103+ T cells' when referencing the T cell clones in our in vitro cellular work throughout our study. Additional proof-reading has now been carried out, to ensure the correct use of terms throughout the study.

C3: *Further, given that the authors show that CD61 is upregulated upon T cell activation, and that this is transient, it is likely that CD61+ TIL are activated T cells that have recently encountered antigen.*

R3: We appreciate the reviewer's agreement with our argument on Figure 2a on the transiency of CD61 expression on T cells, where CD61 is likely only expressed by activated T cells that have recent antigen encounter/activation.

Major Comments:

C4: *Fig 4e. I am struck by the differential protective capacity of CD61+ and CD61- T cells, and that these cells may be epigenetically programmed to be less functional. CD61 may not be the sole cause for this difference in tumor growth.*

R4: We agree CD61 may not be the sole cause for this difference in tumor growth and appreciate the suggestion by the reviewer on the potential epigenetic differences between the T cell clones, which merits further investigation. We have now included several sentences within the discussion section to speculate on the potential contribution of epigenetic differences between cancer-specific CD8+ T cell clones, on text line 698-710.

C5: The overexpression of CD61 in the CD61- NY-ESO T cells or deletion of CD61 in CD61+ NY-ESO T cells would greatly strengthen the authors conclusions, as would the demonstration that CD61 blockade prevents tumour clearance in vivo.

R5: We thank the reviewer for the suggestions of works to further strengthen our conclusions on the role of CD61 towards T cell function. In revised manuscript, we have included new data, by generating: (i) CD61 CRISPR KO T cell clones from the WT CD61+ T cell clones, and (ii) CD61 KD T cell clones using gradient/serial dilution of siRNA treatment, and performed revision of the functional assays to emphasis the role of CD61. In summary, with these new cells generated, we now able to show that CD61 KO T cell clones have impaired TCR signalling (as per new Figure 3c-f, text line 332-341), decreased cytotoxic degranulation (new Figure 4a, text line 393-396) and limited in vitro killing of cancer cells (new Figure 4c, text line 398-403). We further treated the WT CD61+ T cell clones with anti-CD61 blocking antibodies and showed similar impairment of T cell functions in the same figures and text lines (new Figure 4b, line 396-398, Figure 4d, line 403-407. Examples of new added figures for Figure 3c, 3d and 4c is as below:

Figure 3c-d:

Figure 4c:

To further reaffirm this result at ex vivo level, we treated TILs derived from lung cancer patients (n = 19) with the anti-CD61 blocking antibody and found impaired expression of cytotoxic granulysin and granzyme M expression (new Figure 5c, text line 505-508).

C6: • *Is CD61:CD103 directly binding to Lck to bring it to the TCR within the synapse? Does this compensate for CD8?*

R6: We thank the reviewer for the comment. With regards to the compensation for CD8, We think it is unlikely that the CD61:CD103 with Lck can compensate for CD8. This is because it is well-established that TCRs of CD8+ T cells benefit from CD8 binding to MHC complexes to directly promote TCR signalling initiation and TCR/pMHC complex stability.

For the CD61 binder, the identification of the specific intracellular binding partners is beyond the scope of this study, though we have highlighted Lck as a potential adapter protein of CD61 as it is known to interact with ZAP-70 and sustain its phosphorylation. We agree that a full and in-depth CD61 downstream signalling on T cells evaluation merits further investigation, and added a sentence in the discussion session to highlight this point (text line 672-674).

C7: • *Figure 5d – This requires some representative images to show how these regions were determined.*

R7: We thank the reviewer for the comment of now Figure 5e, and now further elaborated in the Methods section, subsection ‘Immunohistochemistry analysis’, sub ‘Quantitative Output Variables and Calculations’ on the methodology text line 1328-1340, as well as an example IHC Visiopharm image to denote width lining of the ‘clustering’ margin areas near the tumor islets on new Extended Data Fig. 9, line 1812-1817. Briefly, the Visiopharm IHC Analysis Protocol Packages (APP) algorithm lining was used to define the ‘clustering’ width denoted by blue dashed, in which the APP set ‘clustering’ as the half average of the distance between one E-Cadherin^{over-expressed} stained region with another.

CD61 co-localisation increases within 5-10min of

T cell activation and then decreases by 10-15min. Are these differences meaningful and are there statistics to support this (it is difficult to assess this observation)? can the authors show CD61 expression at t=0 and 5min post activation (only 2h is shown).

R8: We thank the reviewer for the comment and apologise for the confusion. The difference in co-localisation is qualitatively different. Figure 2c is used to demonstrate consistently stable co-localisation between CD103 and CD61 throughout the synaptic formation, and revised text line 244-245, 326, 329, 1748-1750 has now been made to clarify this. We also now added the statistics to show no changes in the PCC co-localisation, of Figure 2c.

With regards to the reviewer question on *can the authors show CD61 expression at t=0 and 5min post activation (only 2h is shown)*, we are assuming that the reviewer is referring to the Extended Data Figure 4a, where the peak CD61 expression is highest at 2hours post T cell activation. We now have revised the Extended Data Figure 4a to include t=0 and several other timepoints between the t=0 and t=2hr in the revision, to show consistent upregulation of CD61 within the first 2 hours post T cells activation.

C9: • *Line 383-386 – Be careful with this interpretation. As I understand it, the TCGA data used is not sorted on CD3+ CD8+ T cells but represents expression of 4 genes within the excised tumour. This doesn't necessarily mean the CD61 (or the CD103/ITGAE) detected was expressed by the T cells within this transcriptional data.*

R9: We thank the reviewer for this cautionary advice. We agree that TCGA dataset includes genes from various cells types present in the collected tissues, which includes T cells, B cells, cancer cells and many others. However, we used a strategy that allowed us to focus the analysis on T cells specifically, by restricting the analysis to cells expressing *CD3E* and *CD8A* markers. This first step allowed us to identify CD8+ T cells prior to narrow down to populations of interest using *ITGAE* (CD103) and *ITGB3* (CD61). This approach ensures that the analysis of CD103 and CD61 is limited to T cells, and does not include any other cell types. We now clarified the text line 443-448 as well as in the Methods section text line 1543-1548 according to our explanation above.

C10: • *The title of the paper is vague and uninformative.*

R10 We agree with the comment by the reviewer. We have now revised the title to include CD61 and to be more specific.

We would like to propose the title: 'An unconventional CD61 integrin expression on human antigen-specific CD8+ T cells promotes anti-tumor effector and cytotoxic immunity', at line 1-2.

Reviewer #3

(Remarks to the Author)

C1: *This paper describes the expression of CD61 on CD103+ T cells and suggests that these proteins interact to form a novel integrin combination in T cells. The manuscript suggests that CD61 and CD103 co-localise transiently in microclusters at the synapse and that CD61 enhances TCR signaling.*

R1: We greatly appreciate the reviewer for highlighting many of the key elements of the manuscript.

There are several concerns about the approaches and results that are summarized:

C2: *(i) The basic premise starts from comparing the proteome of TCR-matched CD103+ and CD103- T cell clones. Using 1 million T cells per sample cells with CD103+/- were compared. I did not see notes on biological or technical replicates and wondered how reproducible results were? What were the n numbers for proteomics? What was the variability?*

R2: We thank the reviewer for this comment and apologise for leaving out the notes on reproducibility. In this manuscript, we have shown the biological reproducibility of the result by using two pairs of CD103+/- T cell clones from two different cancer patients. The two pairs of CD103+/- T cell clones are highly distinguishable from each other, as each pair (i) were derived from two separate cancer patients, (ii) having different TCR repertoire and CDR3 usages and (iii) having different antigen specificity, which we have extensively described in our previous work- Abd Hamid et al, Cancer Immunol. Res., 2020 (Reference number 12).

In the previous study, and here briefly described in the first sentences of the Results section (text line 140-157), one T cell clone pair was isolated from a gastric cancer patient with the TCR: TRAV8-6 TRAJ 30, TRBV 6-1 TRBJ 2-7 recognising the SSX2 tumor antigen (HLA-A201-restricted, KASEKIFYV peptide). The second pair was isolated from a melanoma patient with distinct TCR repertoire: TRAV 12-2 TRAJ 31, TRBV 12-4 TRBJ 1-2 recognising

the NY-ESO-1 tumor antigen (HLA-A201-restricted, SLLMQITQC peptide). We now revise this text line to emphasize the diversity in biological source.

The proteomics analysis involves two technical duplicates for each clone, with low variability between replicates, and revision to the text line 209 has now been made. We further validated the expression of CD61 on the same pairs of T cell clones using flow cytometry assay with three technical replicates. Surface staining was performed regularly at the start of all in vitro functional assays, the proteomic assay, in vivo and ex vivo assays, to ensure consistent CD61 upregulation following antigenic or anti-CD3/CD28 stimulation. An example of this data is shown in Figure 2a., and the emphasis of the regular staining for experimental consistencies is now added to line 1060-1063, 1100-1104.

C3: *(ii) The authors report increased CD61 expression in T cells after culture with antigenic cancer cells. As cancer cells have been reported to express CD61, is it possible that CD61 has been acquired from cancer cells?*

R3: We thank the reviewer for the comment. While we understand the reviewer's concerns, we would like to point out that we also find that CD61 to be upregulated on T cells stimulated with anti-CD3/CD28 antibodies, in the absence of cancer cells (as per Figure 2a left panel). As such, we believe that the CD61 detected is intrinsic to the T cells.

C4: *(iii) The pull downs to show an interaction between CD61 and CD103 were all carried out using over-expression of both integrin chains. Was there a reason for this? Ideally co-immunoprecipitations would be shown from endogenously expressed integrin chains. Even then co-precipitation does not necessarily show a direct interaction as other cellular components could be contributing. This data needs to be strengthened.*

R4: We thank the reviewer for the comment. The reason why we chose over-expression system to do the pull down is because the transient co-expression between CD61 and CD103 on the T cell clones makes it difficult to perform stable pulldown assay. In order to overcome this limitation, we thus relied on over-expression system to have a better control on the expression of single or doubly transduced chains. In addition, we inserted well characterised tags (ie. HA, Flag and c-Myc), to ensure we used validated antibodies for the IPs and Western Blotting detection. We agree with the reviewer that co-IP does not mean direct interaction and indeed in this manuscript, we have avoided using the term 'direct' interaction, instead, we have used the term 'potential interaction', such as on line 271.

The reviewer has mentioned that co-IP alone is not a sufficient approach, and we agree with this comment. We therefore performed an integrins chain ‘rescue’ assay, as per current Figure 2h, Extended Data Figure 4b and line 297-305. This is based on the well-established notion that individual integrin chain is not able to be presented on the cell surface, but require an alpha and beta chain to paired up for cell surface expression and inducing function. In line with this notion, we demonstrated in Figure 2h that CD103 transduction on primary T cells is not sufficient to allow CD103 cell surface expression. However, if CD61 was subsequently transduced on the same T cells, we readily detected the surface expression of both chains, suggesting that CD61 is likely pairing up with CD103 in this over-expression system in order to express this heterodimer on the cell surface. We included this figure as below:

Additionally, to further address the insufficient approach of using co-IP/WB and in strengthening the potential interaction data, for this revised manuscript, we have carried out mass spectrometry analysis of the pull-down lysate samples as mass spectrometry is a highly sensitive method that can detect/identify multiple proteins that may form part of the CD61:CD103 complex. Using this approach, we confirmed detection and enrichment of the integrins CD61 and CD103 only on CD61+CD103+ T cells, as well as detection of other proteins commonly associated with integrin heterodimer complex- indicating the lysate do comprise of CD61:CD103 integrins protein complex (new Fig. 2h), as figure below:

We now added a new Methods section subsection ‘Mass spectrometry analysis of integrins pulldown lysates’. We now also include new Figure 2g and Extended Data Figure 4g on the

mass spectrometry data as well as new description on text line 285-295, 336-337, 1764-1767 to emphasize the potential interaction via mass spectrometry approach.

C5: (iv) The FACS data shown in 2a indicates a significant surface expression of CD61 after activation (assuming that the x axis is logarithmic; x and y values need to be added to almost all FACS plots including this one). The authors suggest that this is a transient upregulation during which CD61 and CD103 interact and use TIRF imaging to support this idea. In Figure 2b some overlap in the signals for CD61 and CD103 in areas up to 1 micron across are used to suggest that CD61 and CD103 in microclusters (usually thought to be ~0.2microns diameter). The signals for CD61 and CD103 are not coincident in the way that would suggest colocalization but might rather be the result of TIRF showing a 2D image of a 200nm 3D depth. This would occur if the T cell membrane were not completely flat on the bilayer. What is needed is a membrane marker that defines where the cell membrane is relative to the bilayer.

R5: We apologise for not including the x- and y- values for the axes of the FACS plot on Figure 2a (and other FACS plots). We have now corrected this error to all FACS plots (Figure 1d, 2a, 2h, 3g, 5a, 6a, 6e as well as Extended Data Figure 4e, 7a-b).

With regards to reviewer's comment on the flatness of the contact between T cell membrane and the bilayer, we have shown previously in Varma et al., Immunity 2006 that the T cell membrane flattens out via ICAM1: LFA-1 interactions on the substrate restricting the formation of significant membrane protrusions. We have used the same ICAM1: LFA-1 system in this study to ensure flatness of the T cell membrane relative to the bilayer. Additional evidences showing flatness of contact using this system has further been established in other studies, including by Choudhuri et al., Nature, 2014 (Figure 3) and Saliba, Cespedes-Donoso et al., eLife, 2019 (Figure 1c-d).

New Extended Data Figure 10c (as further shown below, text line 1421-1425, 1851-1857), further demonstrated super-res images demonstrating CD103 and CD61 are not enriched in membrane protrusions, rather, bound pairs of CD61 and CD103 co-locate within specific membrane clusters localising in the cSMAC and synaptic cleft at 200nm size. Images of ii and iii shown different orthogonal views demonstrating most CD103 and CD61 signal distributes in the cells-SLB interface and in clusters localised in cSMAC (ii) and synaptic cleft (iii).

To further define the cell membrane relative to the bilayer (apart from above ICAM1: LFA-1 system used), we used a combination of glycoalyx imaging, via wheat germ hemagglutinin (WGA) as membrane marker staining with surface internal reflection contrast microscopy (denoted as IRM). The WGA is used to label plasma membrane-associated glycans on the contact zone, alongside antibody staining of CD103 and CD61. To demonstrate the flatness, shown below are two new image panels on CD103^{high}CD61^{high}-sorted T cell clone (each image panel was derived from two separate independent experiments), where further magnification of both IRM and glycan images demonstrate no increases in the contact reflection (of IRM shades) and glycan fluorescence intensity (plasma membrane signal) indicative of no topological changes in the plasma membrane, such as 3D protrusions being projected in 2D, in our TIRF imaging. Apart from these two independent experiments, the flatness behaviour is also consistent on other clones and TCR-T tested in this assay. These two new image panels are now included in Extended Data Figure 10a, with technicalities also included in the Methods section, line 1392-1408, 1819-1839. Arrow represents areas of CD61 enrichment, no observation of yellow intense indicative of no accumulation of plasma membrane 3D protrusions.

Top panel: scale bar 5μm. Bottom panel: 500nm.

In parallel, we further performed CD61 and CD103 staining on one CD103^{low}CD61^{low}-sorted T cell clone and one CD103^{high}CD61^{high}-sorted T cell clone, in order to emphasise their co-localisation at the contact zone, depending on the T cell's expression of surface CD61 and CD103. As shown below and on Extended Data Figure 10b, note the different levels of expression and co-localisation among the CD61 and CD103 in the different clones expressing different levels of both integrins, with more co-localisation areas observed on the CD103^{high}CD61^{high}-sorted T cell clone. The central enrichment of antigenic HLA-A2_{NY-ESO-1} indicative of the central clustering of TCR bound HLA-A2_{NY-ESO-1}, where CD61 and CD103 are present in the TCR-pMHC microclusters (see also manuscript Extended Data Figure 4b, and Extended Data Figure 10a). Technicalities also included in the Methods section, line 1410-1419, 1938-1851.

C6: Extended data figure 4b shows a time course by TIRF showing cells after 5, 10 and 15 minutes on bilayers labelled for TCR, CD61 and CD103 with images that differ from Figure 2b (also a 10-minute time point). Although the middle panel from 2b and Ext data 4b should look the same, they do not, with many more microcluster-like dots showing little overlap at 5 minutes and increasing clusters at 10 and 15 minutes. The precise overlap between individual clusters that you would expect to see if CD61 and CD103 were colocalized is not evident.

A simpler interpretation of the data might be that the clusters are simply the points of contact as the T cell comes into contact with the bilayer. This is supported by the time course in Ext data 4b. in which more clusters accumulate over time.

R6: As we mentioned in the current manuscript, we do not expect a complete co-localisation of CD61 and CD103 in these three snapshots of Extended Data Figure 4b, as we hypothesised that the pairing is transient and malleable (due to CD61 expression itself being transient) and in the presence (and potential influence) of integrin beta 7 dynamics on the T cells as well. To avoid biases in the co-localisation interpretation, we agree with the reviewer on the use of term 'points of contact' to denote microclusters, and have now included the explanation on text line 234-237 and 243. Additionally, new images of Extended Data Figure 10c demonstrated orthogonal views of bound pairs of CD61 and CD103 co-locate within specific membrane clusters localising in the cSMAC and synaptic cleft at 200nm size, which suggest these integrins might be sorted within nascent synaptic ectosomes and thus why at later time points we seen an increased co-localisation with the TCR (new text line 1421-1425).

C7: (v) The manuscript suggests that CD61 enhances TCR signaling as there are differing levels of signaling proteins in CD61+ and CD61- T cell clones. These data include a “complete CD61 knock-out”, that the methods reveal was generated using pinocembrin treatment (lines 892-4). It is not clear how pinocembrin works, but it is misleading to refer to it as a “complete knock-out” as this drug treatment will undoubtedly produce many different changes in T cell clones. The data provided does not demonstrate a direct link between CD61 and the changes observed in signaling proteins.

R7: We agree with the reviewer that referring to pinocembrin treatment on the T cells as complete KO is misleading as the drug treatment may produce different confounding changes in the T cells. Therefore, we have now generated following T cell conditions to provide more convincing data to demonstrate a direct link between CD61 and TCR signalling proteins;

- A. CD61 CRISPR knock-out (KO) T cell clones, generated from the original WT CD61+ T cell clones,
- B. Gradient/serial dilution of CD61 siRNA treatments on the original WT CD61+ T cell clones,
- C. Anti-CD61 blocking antibody, treated on the original WT CD61+ T cell clones.

The CD61 expression in the parental WT, and abovementioned conditions are now included in revised Extended Data Figure 7a, and as shown below.

We now have repeated the T cell functional assays using these new conditions. In summary, we saw that serial siRNA treatments demonstrated gradual decrease in phosphorylation of the signalling proteins of pZap70, pPLC γ 1 and Lck, and that the CD61 CRISPR KO T cell have limited activity of the signalling proteins, showing similar levels as that of WT CD61⁺ T cell clones (new Figure 3c, d, f, Extended Data Figure 7b).

This further mimic the effect when using anti-CD61 blocking antibody treatment on the original WT (new Figure 3e, Extended Data Figure 7b).

Indeed, the use of these conditions (KO T cells, siRNA treatments and anti-CD61 blocking antibody) demonstrated diminished degranulation marker expression CD107a and killing efficacy of T cells (new Figure 4a-d).

We now clarified the text line 354-363, 368-370, 396-402, 413-427, 467-470, 1792-1804 as well as in the Methods section text line 1046-1107 according to our explanation above.

C8: (vi) *Genistein is a very broad inhibitor of tyrosine kinases and is not specific to Lck. There will be many off-target effects.*

R8: We thank the reviewer for the advice and we have taken this into consideration. We have generated additional data after the manuscript was submitted, using a different inhibitor, A770041, which is known to be more specific towards Lck inhibition. We now showed that cells treated with the A770041 inhibitor also demonstrated downregulation of phosphorylated Zap70, implying A770041 elicits an effect similar to Genistein, and support the notion that the inhibition is specific to Lck. This new information has now been added to text line 370-375, 402-404, Methods section line 1496-1499 to mentioned source of the drug, and revised Figure 3g.

Decision Letter, first revision:

4th Dec 2023

Dear Professor Dong,

As you know, your Article, "An unconventional CD61 integrin expression on human antigen-specific CD8+ T cells promotes anti-tumor effector and cytotoxic immunity" has been seen again by reviewer 2 and 3 and issues were raised with powering and colocalization data.

We have now looked over your author response and plan for revision, and we would be OK in consulting the reviewers again if you carry out these revisions. Regarding the reliance on 2 clones, we appreciate that you have used these clones in an exploratory sense and that there are other data presented, but we do agree with the reviewer 3 that a lot of data here are reliant on these clones. As such, some mediation might be needed by asking other reviewers (present or new) to comment on this debate.

We therefore invite you to revise your manuscript taking into account all reviewer and editor comments. Please highlight all changes in the manuscript text file in Microsoft Word format.

- * Include a "Response to referees" document detailing, point-by-point, how you addressed each referee comment. If no action was taken to address a point, you must provide a compelling argument. This response will be sent back to the referees along with the revised manuscript.
- * If you have not done so already please begin to revise your manuscript so that it conforms to our Article format instructions at <http://www.nature.com/ni/authors/index.html>. Refer also to any guidelines provided in this letter.
- * Please include a revised version of any required reporting checklist. It will be available to referees to aid in their evaluation of the manuscript goes back for peer review. They are available here:

Reporting summary:

When submitting the revised version of your manuscript, please pay close attention to our [href="https://www.nature.com/nature-portfolio/editorial-policies/image-integrity">Digital Image Integrity Guidelines](https://www.nature.com/nature-portfolio/editorial-policies/image-integrity). and to the following points below:

-- that unprocessed scans are clearly labelled and match the gels and western blots presented in

figures.

-- that control panels for gels and western blots are appropriately described as loading on sample processing controls

-- all images in the paper are checked for duplication of panels and for splicing of gel lanes.

[REDACTED]

We hope to receive your revised manuscript within two weeks. If you cannot send it within this time, please let us know. We will be happy to consider your revision so long as nothing similar has been accepted for publication at Nature Immunology or published elsewhere.

Nature Immunology is committed to improving transparency in authorship. As part of our efforts in this direction, we are now requesting that all authors identified as 'corresponding author' on published papers create and link their Open Researcher and Contributor Identifier (ORCID) with their account on the Manuscript Tracking System (MTS), prior to acceptance. ORCID helps the scientific community achieve unambiguous attribution of all scholarly contributions. You can create and link your ORCID from the home page of the MTS by clicking on 'Modify my Springer Nature account'. For more information please visit please visit www.springernature.com/orcid.

Sincerely,

Nick Bernard, PhD
Senior Editor
Nature Immunology

Reviewers' Comments:

Reviewer #2:

Remarks to the Author:
 Authors:

The revised manuscript has been much improved since the initial submission – and the addition of CD161 knockout/knockdown data greatly strengthens the central message of the study.

Minor points:

Line 398 (previous comment 9). Regarding the TGCA data, the authors response was not clear. How can CD8 T cells be 'extracted' from this data set – this would make sense for single cell data but I am unaware that the TGCA dataset was at this resolution. If this is not single cell gene expression, the authors should reiterate that they cannot be sure that the expression they are seeing is on T cells, which does not detract from the associations observed in 4H.

The MS title should read "An unconventional CD61 integrin expressed on" rather than "An unconventional CD61 integrin expression on"
 Check grammar throughout e.g. line 486 "CD61+ Trm TILs do no exhibit" should read "not exhibit".

Reviewer #3:
 Remarks to the Author:
 Continuing concerns:

1. Much of the data in this paper is based on 2 pairs of clones derived from 2 different cancer patients. One of each pair, expressing CD103 the other not. As they are derived from different patients, there are many other differences between these cells. These differences far exceed CD103, as shown in their previous publication (citation 20).
2. The proteomics relies on only 2 technical duplicates for each clone. This is massively under-powered. It is well documented that small differences in cell culture conditions can impact proteomes.
3. Other results also report only two repeats (eg Extended Figure 10). This is insufficient to demonstrate reproducibility.
4. I am puzzled by the authors use of WGA to distinguish CD61 and CD103 localisation from areas of membrane contact. WGA sees glycosylated proteins (that can be both on the plasma membrane and intracellular) not the plasma membrane per se. In Extended data Figure 10, in which the authors describe "Not all areas of membrane-SLB contact shown enrichment of CD61 and CD103 (grey arrows)"; "Arrow represent areas of CD61 enrichment, not yellow intense (WGA) indicative of no accumulation of plasma membrane 3D protrusions." As WGA detects glycosylation and both CD61 and CD103 are glycosylated, there should be complete overlap with WGA signals whether they are on the cell surface or not. Or are the authors suggesting that the CD61 that fails to overlap is not glycosylated?
5. R7—It's good that the authors have produced cells that are CRISPR knocked out. Additional controls showing that the procedure alone did not impact CD61 expression would have been good as CD61 surface appearance is thought to be transient. In addition, information on when (ie how long after KO and KD) was confirmed by FACS needs to be included. Extended figure 7 still shows pinocembrin "knock-out". Was this left in by mistake?
6. A770041 is used in this study at 7.5nM while it is described as showing some selectivity towards inhibiting Lck (over Fyn) at 147nM. How was the 7.5nM dose derived?

Author Rebuttal, first revision:

Dear Dr Bernard,

We greatly appreciate the opportunity to revise our manuscript for publication within Nature Immunology and address the concerns raised by you and the reviewers.

We appreciate the powering concern based on the reliance of data on the 2 clones. In response, we have now **generated additional functional data from 7 additional paired cancer-specific CD103:CD61⁺ T cell lines from seven new cancer patients** to support our results by performing T cell cytotoxicity, CD107a expression, TCR phosphorylation Phosflow assay and CD61 kinetics. Using these additional T cell lines from additional cancer patients, we provided supportive validation evidence on the role of CD61 in regulating T cell functions, as per new Figure 1f, 2a, 3d, 3f, 4b, 4d, Extended Data Figure 7c. Detailed rebuttal for this concern is now on our R1 to reviewer 3 comment 1 below.

We have now revised our manuscript and highlighted and tracked changes for this revision. The detailed responses are as follows, but for ease of reference, we have summarised below the changes:

1. As requested, the manuscript (main text, figure legends and Materials Methods) has been carefully reviewed and clarified throughout.
2. New supportive validation data on the CD61 functional mechanism was generated as requested by the editor and reviewers and included in the revised manuscript. Using additional 7 cancer-specific CD61⁺ T cell lines from seven new cancer patients, we now included the following data:
 - a. CD61 expression only on CD103⁺ T cell lines, but not on CD103⁻ T cell lines (Figure 1f), text lines 169-173 and 193-196.
 - b. CD103⁺ T cell lines upregulate CD61 only following activation, either by agonistic α CD3/CD28 activation or by co-culture with antigenic cancer cells. No upregulation was observed in the resting state (no-activation state) (Figure 2a), text lines 200-201, 204-205, 283-287.
 - c. The seven T cell lines were treated with α CD61 neutralising antibody, isotype control or no treatment, and we showed:
 - i. Phosphorylated Zap70 (pY292) levels diminished following α CD61 neutralising antibody treatment (Figure 3d, Extended Data Figure 7c), text line 327-331, 359-361, 1769-1772, 1764-1767.
 - ii. Lck level diminished following α CD61 neutralising antibody treatment (Figure 3f), text lines 337-341 and 363-365.
 - iii. Limited degranulation marker, CD107a expression following α CD61 neutralising antibody treatment (Figure 4b), text lines 383 and 422-424

- iv. Reduced frequency of cancer cell deaths following prior α CD61 neutralising antibody treatment on the CD61⁺ T cell lines (Figure 4d), text line 389-393 and 426-429.
3. Co-localisation data interpretation has been revised, on text lines 1794-1797.
4. Reviewer's 2 concern on TCGA dataset: rephrasing of wording used has been undertaken, on text lines 407-411, 1489-1500.

Response to Referees

Reviewer #2

(Remarks to the Author)

C1: The revised manuscript has been much improved since the initial submission – and the addition of CD161 knockout/knockdown data greatly strengthens the central message of the study.

R1: We thank the reviewer for the complimentary remark on the past revision of our manuscript.

Minor points:

C2: Line 398 (previous comment 9). Regarding the TCGA data, the authors response was not clear. How can CD8 T cells be 'extracted' from this data set – this would make sense for single cell data but I am unaware that the TCGA dataset was at this resolution. If this is not single cell gene expression, the authors should reiterate that they cannot be sure that the expression they are seeing is on T cells, which does not detract from the associations observed in 4H.

R2: We appreciate the helpful comments by this reviewer, and have now revised the manuscript as follows:

- A. within the Methods section as per text line 1489-1500: Our analysis of TCGA dataset utilises an analytical pipeline that facilitates the identification of *CD8A* and *CD3E* - enriched samples (known to be enriched on T cells), which then were subjected to more granular analyses. As TCGA dataset is not single-cell resolution, the analyses performed may not necessarily be of T cells. Using the *surv_cutpoint* function from the *survminer* R package, we objectively determined the optimal cutpoint for CD8 expression using the following arguments: `time=" Months.of.disease.specific.survival"`, `event=" Disease.specific.Survival.status"`. This was categorised into a categorical variable using

the *surv_categorize* function and samples with high *CD8A* and *CD3E* expression as categorised above were classed as CD3+CD8+ samples. So, the CD3+CD8+ samples analysed would be from data points that show evidence of a high T cell proportion.

5. Reprashing the terminology from the TCGA dataset analysed as per text lines 407-411, 1489-1500.

C3: The MS title should read “An unconventional CD61 integrin expressed on” rather than “An unconventional CD61 integrin expression on”.

R3: We have now revised the title to account for the grammar and be more specific. The new title is: Unconventional human CD61 pairing with CD103 promotes TCR signalling and antigen-specific T cell cytotoxicity, as per text lines 1-2.

C4: Check grammar throughout e.g. line 486 “CD61+ Trm TILs do no exhibit” should read “not exhibit”.

R4: We thank the reviewer for the comment, and additional proofreading and checking of grammatical errors have now been carried out throughout the manuscript.

Reviewer #3

(Remarks to the Author)

C1: Much of the data in this paper is based on 2 pairs of clones derived from 2 different cancer patients. One of each pair, expressing CD103 the other not. As they are derived from different patients, there are many other differences between these cells. These differences far exceed CD103, as shown in their previous publication (citation 20).

R1: We appreciate the reviewer’s concern and we acknowledge other candidate proteins identified from the proteomic analysis might also have functional relevance to CD103+ T cells. Therefore, we have added new sentences in the discussion section, text lines 647-650 to highlight this avenue for further future studies.

We would also like to emphasize our study approach uses exploratory-to-validation workflow to demonstrate that CD61 was one of the candidates, which we subsequently used for the downstream mechanistic work as per Figure 2-4, to understand the functional difference between CD103+ and CD103- T cells, concerning CD61.

We appreciate this reviewer’s concerns about the power of the study using only two pairs of clones from two cancer patients. We therefore performed additional experiments using newly generated 7 paired T cell lines from seven additional cancer patients, confirming that :

- a. CD61 expression is only found on CD103⁺ T cell lines, but not on CD103⁻ T cell lines (Figure 1f), text lines 169-173 and 193-196.
- b. The T cell lines upregulate CD61 only following activation either by agonistic α CD3/CD28 activation or by co-culture with antigenic cancer cells, but no upregulation is observed during resting-state (no-activation state) (Figure 2a), text line 200-201, 204-205, 283-287.

- c. The seven T cell lines were treated with α CD61 neutralising antibody, isotype control or no treatment, and we have shown:
 - i. Phosphorylated Zap70 (pY292) level diminished following α CD61 neutralising antibody treatment (Figure 3d, Extended Data Figure 7c), text lines 327-331, 359-361, 1769-1772, 1764-1767.

- ii. Lck level diminished following αCD61 neutralising antibody treatment (Figure 3f), text lines 337-341 and 363-365.
- i. Limited degranulation marker, CD107a expression following αCD61 neutralising antibody treatment (Figure 4b), text line 383 and 422-424
- iii. Reduced frequency of cancer cell death following prior αCD61 neutralising antibody treatment on the CD61⁺ T cell lines (Figure 4d), text lines 389-393 and 426-429.

C2: The proteomics relies on only 2 technical duplicates for each clone. This is massively under-powered. It is well documented that small differences in cell culture conditions can impact proteomes.

R2: We appreciate the reviewer's concerns about the proteomics assay. However, we would like to emphasize that the technical replication is used to only address potential variability in the

mass spectrometry-based measurement and to minimize potential missing values in the data of either replicate. We would like to emphasize that to minimize differences/major confounding effects which might affect our proteomic analysis such as individual patients, antigen specificity and T cell receptor/pMHC affinity, we have used paired CD103+ and CD103- T cell clones isolated from the SAME patients, with SAME cancer antigen specificity and having the SAME T cell receptor.

This initial exploratory experiment was subsequently followed with exhaustively detailed and orthogonal approaches (as detailed in R1 above) that validate the protein of interest for this mechanistic study. The consistent observations from the validation approaches on our proteomics exploratory experiment demonstrated that our observations are in line with our conclusions.

Further, the additional large NSCLC cohort clinical data strengthened and corroborated our CD61 finding, as supported by reviewer 1: *'the findings in larger NSCLC cohort are valuable'*. Additionally, this comprehensive exploratory-to-validation workflow enabled detailed functional validation and mechanistic study as per Fig 2-6, as reviewer 2 highlighted: *"This is a solid mechanistic study that contributes to our understanding of the promiscuous nature of CD103, which has broad importance for T cell biology, as well as a previously unknown function for CD61"*. As per C1, we acknowledge the heavy reliance on the 2 clones from the 2 patients. Therefore, as further described in R1, we have now generated consistently similar data on the validation part of mechanisms using CD61+ T cell lines from 7 additional new patients. Having 7 additional patients showing consistently similar mechanistic and functional observations to that of the 2 clones reaffirm our hypothesis and importantly, significantly improves the powering of this study.

C3: Other results also report only two repeats (eg Extended Figure 10). This is insufficient to demonstrate reproducibility.

R3: We respectfully disagree, as already shown in the legends of each Figure and Extended Figures, we mentioned the 'number of experimental repeats' - for most in vitro data (n=3) (Fig 3c-g, 4a-d). For ex vivo clinical patient data, we have shown n=19 patients (Fig 1c, 1e, 5, 6) or n=31 patients (Fig 1c, 5e). For in vivo work, we use groups of n=8-10 mice (Fig 4e-g). Some figures such as microscopy images and histograms are representative of the replicates as described in the legends, where the total replicates are given. We acknowledge that replication of the proteomics results is limited, but we only deduce qualitative observation and not statistical conclusions from that data and we followed up the qualitative proteomics results appropriately and comprehensively as discussed above in R1 and R2.

In general, apart from the proteomics, we emphasized in this new revision that all clinical data points are 1 dot point = 1 patient, in vivo work: 1 data point = 1 mouse, in vitro work (apart from proteomics and lysates), typically comprise 3 biological replicates per experiment, for a total of 3 experimental repeats.

There may have been some unclear sentences, therefore, in this revision, we have ensured better clarification in the legends of each Figure and Extended Figure, including on text lines: 180, 187, 190-191, 286-287, 302-303, 305, 372-373, 436-438, 491, 544, 1711, 1716-1717, 1720-1721, 1733, 1758-1761, 1764-1767.

In most TIRFM experiments, images were acquired across a minimum of three independent experiments involving the acquisition of a minimum of 50 synapses per experiment. We have amended the manuscript accordingly to clarify this further (please refer to text lines: 308-310, 371-372, 1725, 1727-1728, 1813-1815.

and Extended Figure 10 legend). We would like to clarify that the images in Extended Figure 10 (evaluating the relationship between WGA and integrins) are representative of two and three independent experiments for SoRa and TIRFM imaging, respectively. We have clarified that SoRa experiments are representative observations of imaging experiments studying both T-cell clones and T-cell lines.

C4: I am puzzled by the authors use of WGA to distinguish CD61 and CD103 localisation from areas of membrane contact. WGA sees glycosylated proteins (that can be both on the plasma membrane and intracellular) not the plasma membrane per se. In Extended data Figure 10, in which the authors describe “Not all areas of membrane-SLB contact shown enrichment of CD61 and CD103 (grey arrows)”; “Arrow represent areas of CD61 enrichment, not yellow intense (WGA) indicative of no accumulation of plasma membrane 3D protrusions.” As WGA detects glycosylation and both CD61 and CD103 are glycosylated, there should be complete overlap with WGA signals whether they are on the cell surface or not. Or are the authors suggesting that the CD61 that fails to overlap is not glycosylated?

R4: WGA binds to N-acetylglucosamine and sialic acid, both of which are abundant monosaccharides found in glycosylated proteins (N-linked), lipids, and small RNAs (PMID: 10468292, 332681663, 4004145). These glycoconjugates compose the glycocalyx, the outermost layer of the cell where most cell surface receptor-ligand interactions, including adhesions, occur. Because lipophilic dyes label bystander SLB and produce various degrees of toxicity, we used WGA labelling after immune synapse formation and fixation to identify global areas of glycocalyx enrichment both on the cell surface and its associated contact points. As we demonstrated in Ext Fig 10 c I (the super-resolution (Sora)), WGA efficiently labels cell surface protrusions, such as microvilli, and is therefore a good probe for the detection of changes in cell surface topography. Because we did not observe the formation of annular structures representing the collapse in a 2D projection (Sora Ext Figure 10 c ii-iii) of 3D cylindrical structures like microvilli, our data suggest that integrins are likely assembling flat microclusters within peripheral and specialised cell-SLB contact points.

We understand the confusion that was made in the figure legends, as described by the reviewer: ‘In Extended data Figure 10, in which the authors describe “Not all areas of membrane-SLB contact shown enrichment of CD61 and CD103 (grey arrows)”; “Arrow represents areas of CD61 enrichment, not yellow intense (WGA) indicative of no accumulation

of plasma membrane 3D protrusions.” We apologise for this confusion. A New revision of the wording has now been made on text lines 1794-1797: ‘White arrows represent areas of CD61 enrichment, where CD61 and CD103 signals concentrate within the cell-bilayer interface at the synapse centre (white arrow, see also Extended Data Figure 10c ii and iii), and poorly localise in microvilli (the non-interface area). Scale bar:5µm.’ Our data do not support the absence of CD61 glycosylation and understanding its impact on function merits further investigation, which is not within the scope of this study.

C5: R7—It’s good that the authors have produced cells that are CRISPR knocked out. Additional controls showing that the procedure alone did not impact CD61 expression would have been good as CD61 surface appearance is thought to be transient.

C5: The CRISPR-Cas9 system that we used in this study removes the CD61 (*ITGB3*) gene itself. Therefore, in the CD61 KO T cells, there would not be any CD61 gene transcription occurring because there is no gene to be transcribed, and thus no protein will be produced by the KO T cells. As Extended Data Figure 7a shows, the KO T cell is completely abrogated of CD61 expression- to the same level as the WT CD61⁻ T cell control, following a minimum of five days of CRISPR-Cas9 CD61 gRNA treatment, to ensure full degradation of pre-synthesized and stored CD61. We have now further clarified this in the Methods section, text lines 1043-1050. Additionally, we have now revised the Extended Data Figure 7a, to further show additional controls we have performed, of i) CD61⁺ T cell with empty vector CRISPR treatment still maintaining positive CD61 expression as that of the WT CD61⁺ T cell, and ii) CD61 consistent depletion on the CD61^{KO} T cells in the first 4 passages of T cell culture/expansion (and this may further assuage the concern of C6 below).

C6: In addition, information on when (ie how long after KO and KD) was confirmed by FACs needs to be included. Extended figure 7 still shows pinocembrin “knock-out”. Was this left in by mistake?

R6: Information on when (ie how long after KO and KD) was confirmed by FACs has now been included in the Methods section, text lines 1043-1050. Similar to R5, we have revised Extended Data Figure 7a to show consistent CD61 abrogation on the CD61^{KO} T cells in the first four passages of T cell culture/expansion used for the functional assays, text lines 1043-1050.

We are a bit confused by the comments regarding pinocembrin, as in the last revision, we believe all our references to pinocembrin and its associated data were already removed. We apologize if there is any reference to Pinocembrin still present. We now have ensured the removal of all information relating to the pinocembrin in this revision. In the current revision, all KO data (ie Ext Data Figure 7, Figure 3-4) refers to CRISPR-Cas9-method CD61^{KO} T cells, not by pinocembrin.

C7: A770041 is used in this study at 7.5nM while it is described as showing some selectivity towards inhibiting Lck (over Fyn) at 147nM. How was the 7.5nM dose derived?

R7: We believe the reviewer's identification of the value of 147nM comes from the first-ever study on the discovery of A770041 inhibitory action by the Hirst lab (Stachlewitz et al., J. Pharmacol. & Exp. Ther., 2005), which demonstrated Ec50 of the drug at 147nM in presence of 1mM ATP, which was performed in mice models, and heparinised human whole blood. The value of 147nM is relevant to the more complex physiology of the systems used in that study. It would be improper/incorrect to equate the same dosage used for the mice model and whole blood to *in vitro* T cell functional assay.

Our system of *in vitro* T cell functional assay is a more controlled assay where T cells are treated directly with the drug- thus the effect would be quite direct in comparison to using the more physiologically complex model such as whole blood and mice models. Using high dosages in *in vitro* assays is well-known to cause cell death, especially when using only a small number of cells in these *in vitro* assays. Therefore, we have originally performed initial optimisation of the drug in the context of our *in vitro* functional assay to determine the best Ec50 concentration: that allows the strongest Lck inhibition on the antigen-specific T cells, while minimising drug toxicity on the T cells. Based on our *in vitro* drug optimisation experiment, we determined that 7.5nM is the optimised concentration for suitable viability of T cells to be used in our functional assays (without the drug killing the T cells), as shown below:

Dosage of A770041 drug treatment:

Thank you very much for the opportunity to perform additional experimentation and to revise the manuscript accordingly. We are most grateful as it has undoubtedly enhanced the strength and validity of the findings.

Yours sincerely

Professor Tao Dong

Decision Letter, second revision:

24th Jan 2024

Dear Dr. Dong,

Thank you for submitting your revised manuscript "Unconventional human CD61 pairing with CD103 promotes TCR signalling and antigen-specific T cell cytotoxicity" (NI-A35975B). It has now been seen again by original referee 3

Although this reviewer is still fairly negative we think that we can now proceed in principle to publish your paper in Nature Immunology if you can address reviewer 3 points 4 and 5. We believe that this will only require textual modification. For point 4, the figure can be removed or at a minimum de-emphasized and conclusions from it watered down. For point 5, we are hopeful that a better explanation of the methods can address this concern. The reviewer is worried that you have not CRISPRd CD61 out, but rather have sorted for CD61 expression, as you show a single peak of CD61-negative cells, and describe this as a "clone". We agree this is problematic if correct. Please clarify and if the reviewer is correct then please ensure that the text makes this clear and avoids over-interpretation. The reviewer says that really you should show the gene deletion and notes that IDT sells a relevant kit for this.

However, we will now perform detailed checks on your paper and will send you a checklist detailing our editorial and formatting requirements in about a week. Please do not upload the final materials and make any revisions until you receive this additional information from us.

If you had not uploaded a Word file for the current version of the manuscript, we will need one before beginning the editing process; please email that to immunology@us.nature.com at your earliest convenience.

Thank you again for your interest in Nature Immunology Please do not hesitate to contact me if you have any questions.

Sincerely,

Nick Bernard, PhD
Senior Editor
Nature Immunology

Reviewer #3 (Remarks to the Author):

1. Additional text acknowledges the fact that other differences exist.
2. The authors acknowledge that the proteomics is underpowered and indicate that it was simply used as an exploratory data set. I recommend that they include additional text explaining that while the proteomics studies were underpowered, they chose to pursue CD61 from this preliminary data.
3. Reproducibility descriptions in figure legends are improved.
4. The authors seem not to have followed my last query, so I will try to clarify that in Ext data 10 the panel (a) the WGA and CD61 panels show an arrow pointing at a dark area in which WGA signal is not seen (lower row, 3rd from left). In the CD61 panel there is a CD61 signal that would correspond to the point in the cell that does not label with WGA. This was what I was querying as it implies that CD61 in this location does not label with WGA. As CD61 has carbohydrate modifications it should label. Overall, Ext Figure 10 is highly confusing, does not add to the main message and would be better removed together with the text describing these results, particularly as these results are only

discussed in the methods section.

5. The CRISPR KO data still needs further clarification. The additional methods indicate that the cells were treated for 5 days with gRNA (I assume that they mean RNPs). However, the protocol also indicates that the cells were sorted and cloned by serial dilution. For what marker were these cells sorted? Was it CD61? If so how do the authors know that these are gene deleted and any different from the "WT CD61- clones"?

6. OK

Additional comment: additional and existing data in figures show bar graphs with MFI plotted. It is not always clear whether this was an MFI established from FACS data or imaging data. This needs to be clear and primary data would need to be available.

Author Rebuttal, Second Revision:

Response to Referees

Dear Dr Bernard,

We have made the changes according to reviewer 3 Point 4 and 5, as below:

Reviewer #3:

Remarks to the Author:

4. The authors seem not to have followed my last query, so I will try to clarify that in Ext data 10 the panel (a) the WGA and CD61 panels show an arrow pointing at a dark area in which WGA signal is not seen (lower row, 3rd from left). In the CD61 panel there is a CD61 signal that would correspond to the point in the cell that does not label with WGA. This was what I was querying as it implies that CD61 in this location does not label with WGA. As CD61 has carbohydrate modifications it should label.

Overall, Ext Figure 10 is highly confusing, does not add to the main message and would be better removed together with the text describing these results, particularly as these results are only discussed in the methods section.

R: We fully removed Extended Data Figure 10 from the supplemental figure and text mentions in the Methods section, per reviewer and Editor recommendation.

5. The CRISPR KO data still needs further clarification. The additional methods indicate that the cells were treated for 5 days with gRNA (I assume that they mean RNPs). However, the protocol also indicates that the cells were sorted and cloned by serial dilution. For what marker were these cells sorted? Was it CD61? If so how do the authors know that these are gene deleted and any different from the "WT CD61- clones"?

R: We made textual modifications to clarify the method as per:

1. line 929-930 (Ablation of the gene of interest, *ITGB3*, was achieved by transfection with Cas9-gRNA ribonucleoprotein (RNP) complexes.
2. line 927-929 (We used the WT CD61⁺ T cell clone which is 100% positive for CD61 expression as the cell input for treatment (Extended Data Figure 5a).
3. line 947-955 (The 5-day culture post CRISPR/Cas9 edition was made to ensure full degradation of pre-existing and synthesised CD61. Cell sorting was then performed using CD61 marker, to enable the selection of truly CD61 negative (CD61^{KO}) cells from the initial 100% positive WT CD61⁺ T cell input. On the foundational basis that a gene deletion leads to the absence of protein, we carried out flow cytometry staining on the sorted cells at 4 different passages. Based on Extended Data Figure 5a, while the WT CD61⁺ T cell (input cell) was 100% positive for CD61, the sorted cell stained at each passage was absent of CD61, confirming the KO effect.)

Final Decision Letter:

Dear Dr. Dong,

I am delighted to accept your manuscript entitled "Unconventional human CD61 pairing with CD103 promotes TCR signalling and antigen-specific T cell cytotoxicity" for publication in an upcoming issue of Nature Immunology.

Over the next few weeks, your paper will be copyedited to ensure that it conforms to Nature Immunology style. Once your paper is typeset, you will receive an email with a link to choose the appropriate publishing options for your paper and our Author Services team will be in touch regarding any additional information that may be required.

Acceptance is conditional on the data in the manuscript not being published elsewhere, or announced in the print or electronic media, until the embargo/publication date. These restrictions are not

intended to deter you from presenting your data at academic meetings and conferences, but any enquiries from the media about papers not yet scheduled for publication should be referred to us.

Please note that *Nature Immunology* is a Transformative Journal (TJ). Authors may publish their research with us through the traditional subscription access route or make their paper immediately open access through payment of an article-processing charge (APC). Authors will not be required to make a final decision about access to their article until it has been accepted. Find out more about Transformative Journals.

Your paper will be published online soon after we receive your corrections and will appear in print in the next available issue.

Also, if you have any spectacular or outstanding figures or graphics associated with your manuscript - though not necessarily included with your submission - we'd be delighted to consider them as candidates for our cover. Simply send an electronic version (accompanied by a hard copy) to us with a possible cover caption enclosed.

If you have not already done so, we strongly recommend that you upload the step-by-step protocols

used in this manuscript to the Protocol Exchange. Protocol Exchange is an open online resource that allows researchers to share their detailed experimental know-how. All uploaded protocols are made freely available, assigned DOIs for ease of citation and fully searchable through nature.com. Protocols can be linked to any publications in which they are used and will be linked to from your article. You can also establish a dedicated page to collect all your lab Protocols. By uploading your Protocols to Protocol Exchange, you are enabling researchers to more readily reproduce or adapt the methodology you use, as well as increasing the visibility of your protocols and papers. Upload your Protocols at www.nature.com/protocolexchange/. Further information can be found at www.nature.com/protocolexchange/about .

Please note that we encourage the authors to self-archive their manuscript (the accepted version before copy editing) in their institutional repository, and in their funders' archives, six months after publication. Nature Portfolio recognizes the efforts of funding bodies to increase access of the research they fund, and strongly encourages authors to participate in such efforts. For information about our editorial policy, including license agreement and author copyright, please visit www.nature.com/ni/about/ed_policies/index.html

Sincerely,

Nick Bernard, PhD
Senior Editor
Nature Immunology